# TO AUGMENT OR NOT TO AUGMENT? DIAGNOSING DISTRIBUTIONAL SYMMETRY BREAKING

**Hannah Lawrence**[*1], **Elyssa Hofgard**[*1], **Vasco Portilheiro**[2], **Yuxuan Chen**[3], **Tess Smidt**[1],
**Robin Walters**[3]
[1]MIT, [2]University College London, [3]Northeastern University
[*]Equal contribution.

## ABSTRACT

Symmetry-aware methods for machine learning, such as data augmentation and equivariant architectures, encourage correct model behavior on all transformations (e.g. rotations or permutations) of the original dataset. These methods can improve generalization and sample efficiency, under the assumption that the transformed datapoints are highly probable, or "important", under the test distribution. In this work, we develop a method for critically evaluating this assumption. In particular, we propose a metric to quantify the amount of symmetry breaking in a dataset, via a two-sample classifier test that distinguishes between the original dataset and its randomly augmented equivalent. We validate our metric on synthetic datasets, and then use it to uncover surprisingly high degrees of symmetry-breaking in several benchmark point cloud datasets, constituting a severe form of dataset bias. We show theoretically that distributional symmetry-breaking can prevent invariant methods from performing optimally even when the underlying labels are truly invariant, for invariant ridge regression in the infinite feature limit. Empirically, the implication for symmetry-aware methods is dataset-dependent: equivariant methods still impart benefits on some symmetry-biased datasets, but not others, particularly when the symmetry bias is predictive of the labels. Overall, these findings suggest that understanding equivariance — both when it works, and why — may require rethinking symmetry biases in the data.

## 1 INTRODUCTION

By integrating physical symmetries into model architectures as group invariances, equivariant neural networks often achieve superior performance across materials science (Liao et al., 2023), robotics (Wang et al., 2024a), drug discovery (Igashov et al., 2024), fluid dynamics (Wang et al., 2021), computer vision (Esteves et al., 2019), and beyond. Given a group $G$ of symmetries, like rotations or permutations, a function or neural network $f$ is equivariant if $f(gx) = gf(x) \ \forall g \in G$. Their success is typically explained in terms of improved sample efficiency and generalizability, resulting from the ability to relate data sample $x$ and transformed data sample $gx$ (Cohen & Welling, 2016b). Alternatively, data augmentation may be used to enforce equivariance by applying a random $g$ to each input $x$ in the training set and its corresponding label. For all of these equivariance methods, it is an *explicit assumption* that the ground truth function $f$ is equivariant (Thomas et al., 2018; Cohen & Welling, 2016a; Cohen et al., 2018). However, there is also often an *implicit assumption* that transformed samples $gx$ occur relatively uniformly in distribution, i.e. $p(x) \approx p(gx)$. Theoretical results on the benefits of equivariance and data augmentation almost always assume that $x$ and $gx$ are equally likely under the data distribution (Elesedy & Zaidi, 2021; Chen et al., 2020; Mei et al., 2021; Lyle et al., 2020; Tahmasebi & Jegelka, 2023).

In this paper, we study *distributional symmetry breaking* (Wang et al., 2024b;d), or equivalently *symmetry bias*—when a datapoint $x$ and its transform $gx$ are *not* equally likely under the data distribution. Formally, this means that $p_X(x) \neq p_X(gx)$ for some $x \in X$ and $g \in G$, where $p_X : X \to \mathbb{R}$ is the data distribution, or equivalently that all points in the *orbit* $\{gx\}_{g \in G}$ of $x$ are not equally likely. This paper takes a step towards the goal of understanding how distributional symmetry breaking affects the performance of equivariant methods, including the ubiquitous practice of data augmentation. Intuitively, although equivariance can help performance by providing the correct inductive bias on all

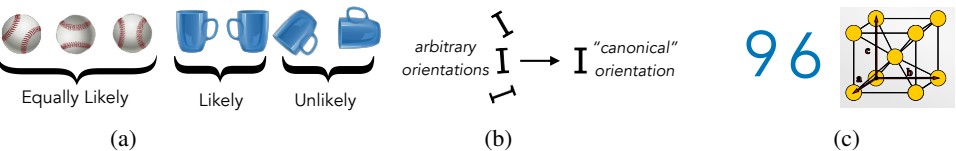

(a)  (b)  (c)

Figure 1: (a) **Distributional symmetry breaking**: Baseballs are likely to occur in any orientation in photos, and are therefore uniform across orbits. In contrast, coffee mugs are more likely to appear with the handle on the side. The latter is an example of distributional symmetry breaking. (b) **Canonicalization**: Canonicalization is when an object only ever appears in one, "canonical", orientation. This is the strongest form of distributional symmetry breaking. (c) **Inherent vs. user-defined canonicalization**: Datapoints can be canonicalized for reasons that are inherent, such as the orientation of a digit determining whether it is a 6 or a 9. However, it can also be user-defined, such as the orientation of a crystal lattice, without any deeper connection to the data-generating process.

transformations of the input data, it may also discard useful information. For example, consider the oft-discussed example of classifying "6"s and "9"s in the MNIST dataset. The two digits look very similar[1] when rotationally aligned, but are easily distinguishable under their naturally occurring orientations. Thus under rotational augmentation, this task becomes much more difficult. In general, this discarded orientation information may be *inherent*, such as the previous MNIST example, or *user-defined*, such as the conventions used to orient crystal structures (Figure 1). In practice, Cohen et al. (2018) demonstrated that rotational equivariance only improves performance on MNIST when the dataset is artificially rotated. Thus at a high level, distributional symmetry breaking can impact how non-equivariant methods perform relative to equivariant methods in-distribution.

Yet, quantifying the amount of symmetry breaking in a distribution remains challenging, particularly in the absence of domain knowledge (Wang et al., 2024b; 2023; 2024d). We thus propose a metric to measure the degree of distributional symmetry breaking, which can place a distribution on the spectrum between fully symmetrized (or "isotropic")—where all points in each orbit $\{gx\}_{g \in G}$ are equally likely—on one side, and fully canonicalized—where only a single sample $x$ in each orbit $\{gx\}_{g \in G}$ is in-distribution—on the other (Figure 1). We aim for this metric to prove useful both as a practical tool for data exploration, and as a lens for rethinking the more fundamental questions of why, and when, equivariant methods succeed.

Concretely, we propose a two-sample classifier test (Lopez-Paz & Oquab, 2017), in which a model is trained to distinguish between samples from $p_X$ (the original data distribution) and $\bar{p}_X$ (the augmented data distribution) (Figure 2). The accuracy of this classifier on a held-out test set is a natural, *interpretable* measure, between 0 and 1, of distance between $p_X$ and $\bar{p}_X$. This (1) allows for interpretability methods (applied to the classifier itself), and (2) sidesteps the kernel selection required by Chiu & Bloem-Reddy (2023) in their tests for distributional symmetry, offloading it to the less impactful choice of architecture. Applying this metric to a variety of datasets, including QM9 (Wu et al., 2017), revised MD17 (Christensen & von Lilienfeld, 2020), OC20 (Chanussot* et al., 2021), and ModelNet40 (Wu et al., 2015), we find that all are highly non-uniform under 3D rotations.

Complementing these empirical findings, we provide nuanced theory on the trade-offs between different equivariant methods under distributional symmetry-breaking, and show equivariant methods can be harmful depending on properties of the data distribution. We use ridge(less) regression as a model, which captures some of the behavior of neural networks when applied in the neural tangent kernel space (D'Ascoli et al., 2020; Atanasov et al., 2023; Jacot et al., 2018). We show that even when the ground-truth function is invariant, data augmentation can be harmful when invariant and non-invariant features are strongly correlated. As our main contributions, we:

- Define a flexible metric for measuring distributional symmetry breaking in a dataset (Section 2). This is a tool for probing datasets' symmetry biases without *a priori* knowledge of their creation.

---

[1] Distributional symmetry breaking differs from *functional symmetry breaking* (Wang et al., 2024d), where the mapping between inputs and outputs is not fully equivariant (e.g. during a phase transition in a material). For the purposes of this paper, we treat "6" and "9" as distinct digits that simply look similar — distributional, not functional, symmetry breaking. This is supported by the observation in e.g. Wang et al. (2024b) that they can be correctly classified with high accuracy.

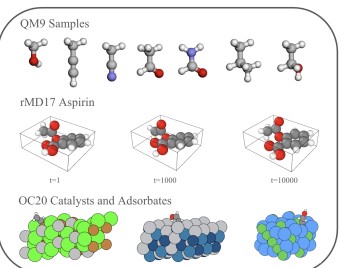 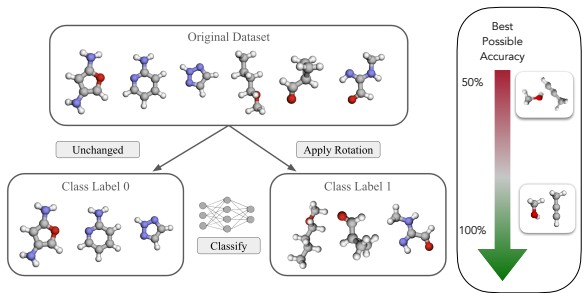

Figure 2: (left) Visualizations of unrotated samples from several materials datasets, with their canonicalization visible. (right) A classifier test for determining if a sample is from the original dataset, or rotated. With no distributional symmetry breaking, no classifier can achieve better than 50% test accuracy. However, if the original dataset was fully canonicalized, the classifier can theoretically achieve perfect accuracy (for an infinite group; otherwise, $1 - \frac{1}{2|G|}$).

- Provide a novel theoretical analysis of invariant ridge regression under distributional asymmetry, showing that data augmentation sometimes hurts (Section 4).

- Use our metric to discover that point cloud benchmarks, including QM9 and ModelNet40, are highly canonicalized (Section 5). We correspondingly evaluate the impact of equivariant methods (augmentation, constrained architecture, and stochastic averaging), using an auxiliary task-dependent metric for finer-grained analysis. We find surprising results on the relation to distributional symmetry breaking, as summarized in Figure 6.

## 2 PROPOSED METRICS

Consider data points $x \in \mathcal{X}$ drawn from a distribution $p_X$, acted on by a compact group $G$. We assume that there is a ground truth labeling function $f : \mathcal{X} \to \mathcal{Y}$ that is equivariant, i.e. $f(gx) = gf(x)$. We do *not* assume that $p_X(x) = p_X(gx)$; instead, we wish to quantify the degree to which $p_X$ breaks distributional symmetry by failing to satisfy this equality, i.e., **to define a metric $m(p_X)$ which measures how close $p_X$ is to symmetric.** To this end, define the symmetrized density $\bar{p}_X(x) := \int_{g \in G} p_X(gx)dg$. The density $\bar{p}_X$ is the closest invariant distribution to $p_X$: for any $G$-invariant measure $\mu$ on $\mathcal{X}$ it minimizes $\int_x (i(x) - p_X(x))^2 d\mu(x)$ over all invariant densities $i$.

We assume a dataset of samples from $p_X$, and obtain samples from $\bar{p}_X$ by applying random $G$-augmentations. As our metric of distributional symmetry breaking, we now wish to approximate some notion of distance $d$ between $p_X$ and $\bar{p}_X$ based on a finite number of samples — but this is not straightforward to choose or compute.

Chiu & Bloem-Reddy (2023) set $d$ to be the maximum mean discrepancy (MMD) with respect to some choice of kernel, corresponding to a non-parametric two sample statistical test. However, there is not always a clear choice of kernel. For example, for materials datasets of geometric graphs, Chiu & Bloem-Reddy (2023) do not provide an applicable kernel that includes chemical information. Rectifying this requires choosing a kernel suitable for $\mathcal{X}$, which may be non-trivial, and as noted in Lopez-Paz & Oquab (2017), may not return values in units that are directly interpretable.

We propose instead applying a two sample classifier test, a common tool for detecting and quantifying distribution shift in machine learning (Lopez-Paz & Oquab, 2017). We train a small neural network NN to distinguish between distributions as a binary classification task, and define the distance $d$ between distributions as the *test* accuracy:

$$d_{class}(p_0, p_1) = \mathbb{E}_{c \sim \text{Bern}(\frac{1}{2})} \mathbb{E}_{x \sim p_c} \left[ \mathbb{1}(\text{NN}(x) = c) \right].$$

Our metric is then $m(p_X) := d_{class}(p_X, \bar{p}_X)$. Concretely, we construct a binary classification dataset from an original dataset as shown in Figure 2 and Algorithm 1, with half of the dataset transformed by random group elements (label 1), and the rest unchanged (label 0).

---

**Algorithm 1** Metric for Distributional Symmetry Breaking, $m(p_X)$

---

1: **Inputs:** Unlabeled train/test sets $\mathscr{D}_{train}$ and $\mathscr{D}_{test}$, group $G$, binary classifier network NN
2: **For** $split \in \{train, test\}$:
3:    Randomly divide $\mathscr{D}_{split}$ into equally sized $D_{split}$ and $\widetilde{D}_{split}$
4:    For each $x \in \widetilde{D}_{split}$, uniformly sample $g \sim G$ and apply $g$ to $x$
5:    Define classification dataset $D^*_{split} := \{(x,0) : x \in D_{split}\} \cup \{(x,1) : x \in \widetilde{D}_{split}\}$
6: **Train** binary classifier NN on the dataset $D^*_{train}$ with the standard BCE loss
7: **Return** NN's test accuracy, $\mathbb{E}_{(x,c) \in D^*_{test}} [\mathbb{1}(\text{NN}(x) = c)]$

---

**Interpretation of $m(p_X)$** The trained classifier's test accuracy is easily interpretable, reflecting how often it can distinguish between the original and symmetrized distributions. If $p_X$ is already group-invariant, then $p_X = \bar{p}_X$ and no network can reliably distinguish between samples from the two, so $m(p_X) \approx 1/2$. If in contrast $p_X$ is canonicalized in a discernable way, then $m(p_X) \approx 1$.

To build intuition for how $m(p_X)$ interpolates between these two extremes, let us also compute it for the case of a finite group, with a dataset consisting of a single orbit $\{gx_1 : g \in G\} := \{x_1, x_2, \ldots x_r\}$. Parametrize the data distribution as $p(x_i) = \theta_i, \sum_i \theta_i = 1$. What is the optimal classification accuracy between a uniform distribution over $x_1, x_2, \ldots x_r$ (class 1), and $p$ (class 0), under infinite samples? For each $i$, the optimal classifier assigns 1 if $\frac{1}{r} > \theta_i$, and 0 otherwise. The resulting optimal accuracy is $m(p_X) := 1 - \frac{1}{2} \sum_{i=1}^{r} \min(\frac{1}{r}, \theta_i)$. For example, for a multimodal distribution with probability mass equally distributed among $m$ modes, the best possible accuracy is $1 - \frac{m}{2r}$, which interpolates between $\frac{1}{2}$ when $m = r$ and $1 - \frac{1}{2r}$ for a perfectly canonicalized distribution. In this analysis, we have assumed infinite samples, an adequately expressive NN, and perfect optimization (although it is accurate for MNIST; see Table 1). In reality, these factors will affect $m(p_X)$ as discussed in Appendix B, although ablations (Appendix D.5) indicate little sensitivity to the size of NN. For an infinite group like $SO(3)$, we instead have $m(p_X) \leq 1$, and rely on a validation set to avoid overfitting.

## 2.1 TASK-DEPENDENT METRIC $t(p_{X,Y})$

The value $m(p_X)$ determines whether there is discernible lack of uniformity over group transformations in the unlabeled dataset. However, it does not capture whether that distributional symmetry breaking (e.g. preferred orientations) is correlated with the task labels, such as MNIST 6s/9s. If it does, then we hypothesize that augmenting is likely to be a poor choice, as it discards task-relevant information contained in the orientations (Appendix A.1 formalizes this information loss claim).

Towards this goal, we briefly introduce a metric $t(p_{X,Y})$ of *task-useful* distributional symmetry breaking (see Appendix C for full details). Let $c \colon \mathscr{X} \to G$ be a canonicalization function, denoting where on each orbit $x$ is. Since data augmentation destroys any information contained in $c(x)$, we wish to understand the dependence between orientations $c(x)$ and labels $f(x)$. A natural way to do this is to predict $f(x)$ directly from $c(x)$, where $c(x)$ is a randomly initialized, untrained equivariant neural network.[2] We then compare the test loss $\mathscr{L}(c(x) \to f(x))$ to that obtained when the inputs are randomly transformed by elements of the given group, $\mathscr{L}_{rot} = \mathscr{L}(c(gx) \to f(gx), g \sim G)$ which removes any task-relevant information in the orientations. Thus, $t(p_{X,Y}) := \frac{\mathscr{L}_{rot}}{\mathscr{L}}$ is large if $\mathscr{L} << \mathscr{L}_{rot}$, i.e. the symmetry bias is relevant to the specific task at hand.

## 3 RELATED WORK

**Learning symmetry breaking** Several works seek to discover *functional symmetry breaking*, where the task may be only partially, rather than fully equivariant, i.e. there are some $x$ and $g$ such that $f(gx) \neq gf(x)$ (Wang et al., 2024d; Finzi et al., 2021; McNeela, 2023; Hofgard et al., 2024; Smidt et al., 2021; Urbano & Romero, 2024a). We distinguish this (more common) notion from our focus, *distributional symmetry breaking* ($p(x) \neq p(gx)$), which Wang et al. (2023; 2024c) showed can harm the performance of equivariant models. Indeed, several works proposing equivariant methods

---

[2]This is closely related to the concepts of V-information (Xu & Raginsky, 2017) and the information bottleneck (via the canonicalization) (Tishby et al., 1999).

have noted that the improvement of their method relative to baselines relies on applying test-time augmentations (Cohen et al., 2018; Kaba et al., 2023). This motivates our method.

**Learning how to augment** Learning an augmentation distribution is one way to address either kind of symmetry breaking. Benton et al. (2020a) address functional symmetry breaking by learning an augmentation distribution. For example, Miao et al. (2023) encode an input using an invariant network, then use this encoding to sample from a learned distribution, feeding randomly transformed inputs into a classifier. Urbano & Romero (2024b) pursue a similar goal in a self-supervised setting, and show their method can be used to canonicalize data, or detect when an input is transformed out of distribution. Learning to predict transformations applied to data, which is possible only with distributional symmetry breaking, was proposed for representation learning by Gidaris et al. (2018). We explore how $c(p_X)$ can be used for selective augmentation in Appendix D.2.4.

**Detecting distributional symmetry** In the unsupervised setting, Desai et al. (2022) and Yang et al. (2023) train discriminative networks for symmetry discovery in a similar way to our binary classifier, but do not produce a quantitative measure of distributional asymmetry on benchmarks. Chiu & Bloem-Reddy (2023) consider non-parametric hypothesis tests for distributional symmetry, and use the distance between the group-averaged and original distributions as the test statistic. Soleymani et al. (2025) devise a robust kernel test for invariance, where a witness $g \in G$ must be provided to prove $p$ is non-invariant. Charvin et al. (2023) propose an information theoretic framework for detecting distributional *equivariance* (rather than invariance, as we consider here).

**Pros and cons of invariant methods** Our theoretical work follows up on Elesedy & Zaidi (2021); Chen et al. (2020), who show that when $p_x$ is invariant, symmetrization or data augmentation improve risk. Most existing work that studies the benefits of invariance in over-parameterized settings similar to ours also assumes invariant $p_x$ (Mei et al., 2021; Bietti et al., 2021). On the limitations of invariant methods, Shao et al. (2024) established that any equivariant algorithm applied to extrinsically equivariant data, under certain assumptions on the hypothesis class, cannot obtain optimal sample complexity in terms of PAC learnability. Lin et al. (2024); Huang et al. (2025) also study unexpected effects of data augmentation, although not focusing on the effects of symmetry.

## 4 THEORY: INVARIANT REGRESSION UNDER DATA ASYMMETRY

To exhibit the subtleties of distributional symmetry-breaking, we analyze high-dimensional ridge regression under non-symmetric covariance. We show that even when the ground-truth function is invariant, *data augmentation and symmetrization can be harmful when invariant and non-invariant features are strongly correlated.* This is intuitive: a non-invariant feature is useful for an invariant task if it correlates well with an invariant feature used by the ground truth function, and augmentation renders such a non-invariant feature unusable. Perhaps surprisingly, data augmentation is always helpful in the under-parameterized regime, while in the over-parameterized regime it can be harmful even when data is fully symmetric.

Suppose $G \leq O(d)$ acts linearly on $\mathbb{R}^d$, and let $y_i = x_i^\top \beta + \varepsilon_i$ for i.i.d. $x_i \sim \mathcal{N}(0, \Sigma)$, $\varepsilon_i \sim \mathcal{N}(0, \sigma^2)$, and invariant[3] ground truth $\beta$ (i.e. $g\beta = \beta$ for all $g \in G$). Importantly, we do not assume $g\Sigma g^\top = \Sigma$ (so $p_x$ may not be invariant). Given data $\{(x_i, y_i)\}_{i=1}^n$ and $\lambda > 0$, we consider the ridge regression problem, $\hat{\beta}_\lambda = \arg\min_\beta \frac{1}{n} \|y - X^\top \beta\|^2 + \lambda \|\beta\|^2 = (\hat{\Sigma} + \lambda I)^{-1} \hat{\Sigma}_{yx}$ where $\hat{\Sigma} = X^\top X / n$ and $\hat{\Sigma}_{yx} = X^\top y / n$ for $X \in \mathbb{R}^{n \times d}$ the matrix of samples and $y \in \mathbb{R}^n$ the label vector.

There are several natural approaches to enforcing invariance. Under the standard inner product, $\mathbb{R}^d$ decomposes into two orthogonal subspaces $V_0$ and $V_\perp$, where $V_0$ is the $d_0$-dimensional set of vectors invariant to $G$. In the first approach, Elesedy & Zaidi (2021) consider *test-time symmetrization*, $\mathbb{E}_g[g\hat{\beta}] = P_0\hat{\beta}$, where $P_0$ is the orthogonal projection onto $V_0$. The second approach is to use only the *invariant features* in the data, $\hat{\beta}_{\lambda, \text{inv}} = \arg\min_\beta \frac{1}{n} \|y - (XP_0)^\top \beta\|^2 + \lambda \|\beta\|^2 = (\hat{\Sigma}_{\text{inv}} + \lambda I)^{-1} \hat{\Sigma}_{yx, \text{inv}}$ where $\hat{\Sigma}_{\text{inv}} = (XP_0)^\top XP_0 / n = P_0 \hat{\Sigma} P_0$ and $\hat{\Sigma}_{yx, \text{inv}} = (XP_0)^\top y / n = P_0 \hat{\Sigma}_{yx}$. In the linear setting, this turns out to be the **same** as ridge regression when (1) restricting $\beta$ to be invariant, or (2) under *infinite data augmentation*, i.e. the model sees $(gx_i, y) \ \forall g \in G$ (Appendix A.3).

---

[3]While such a $\beta$ must be 0 if $G = O(d)$, the same is not true for $G \leq O(d)$, e.g. $G$ a subgroup of the permutation group that acts on only a subspace of $\mathbb{R}^d$.

For any estimator $\hat{\beta}$, we are interested in its generalization error (or risk) on unseen data. Conditioned on the input data $X$, it takes the form $R_X(\hat{\beta}) = \mathbb{E}_{x,\varepsilon}[(x^\top \beta - x^\top \hat{\beta})^2 | X] = \mathbb{E}_\varepsilon[\|\beta - \hat{\beta}\|_\Sigma^2 | X]$, where $\|\beta\|_\Sigma^2 = \beta^\top \Sigma \beta$. Elesedy & Zaidi (2021) prove that when $p_x$ is invariant, one can always do better by symmetrizing at test time: $\mathbb{E}_X[R_X(P_0\hat{\beta})] \leq \mathbb{E}_X[R_X(\hat{\beta})]$ (even for non-linear predictors). We study the behavior of non-invariant $p_x$ under two settings.

- **Under-parametrized ridgeless regime:** When $d < n-1$ and $\lambda \to 0$, correlations between invariant and non-invariant features can drive $\mathbb{E}_X[R_X(P_0\hat{\beta})]$, the risk of test-time symmetrization, to infinity. But surprisingly, data augmentation is always helpful, even if $p_x$ is not invariant.

- **Over-parametrized regime:** This setting captures some of the behavior of real-world neural networks (D'Ascoli et al., 2020; Atanasov et al., 2023; Jacot et al., 2018). When $d > n$, we use a minimal model to show data augmentation can be harmful when there are strong correlations between invariant and non-invariant features, particularly when they lie in a space of dimension significantly smaller than $d$.

## 4.1 THE UNDER-PARAMETERIZED RIDGELESS REGIME

Using straightforward expressions for the bias-variance decomposition (see Lemma 2, Appendix), we show that data augmentation always improves generalization when $d < n-1$ and $\lambda \to 0$.

**Theorem 1.** *In the under-parameterized ridgeless setting, assuming $\Sigma$ is full-rank, $\mathbb{E}[R_X(\hat{\beta})] = \frac{\sigma^2 d}{n-d-1} \geq \mathbb{E}[R_X(\hat{\beta}_{\text{inv}})] = \frac{\sigma^2 d_0}{n-d_0-1}$, so augmentation helps. In contrast, for test-time symmetrization we have $\mathbb{E}[R_X(P_0\hat{\beta})] = \frac{\sigma^2}{n-d-1}\text{Tr}(\Sigma^{-1}\Sigma_{\text{inv}}) \geq \frac{\sigma^2 d_0}{n-d-1}$, with equality when $p_x$ is invariant.*

While Elesedy & Zaidi (2021, Theorem 7) prove a non-negative gap for test-time symmetrization when $p_x$ is invariant, we see its risk can be much larger than that of regular (unconstrained) linear regression, when $\Sigma^{-1}$ does not "align" with $\Sigma_{\text{inv}}$. (This is illustrated in an example in Appendix A.6.)

## 4.2 THE OVER-PARAMETERIZED REGIME

We next consider $d > n$, taking the regime $n, d \to \infty$ and $d/n \to \gamma > 1$ to get deterministic estimates of the risk (assuming $\Sigma$ has bounded spectrum). It is known that as $\gamma \to 1$, the test risk of the usual ridgeless estimator $\hat{\beta}$ diverges. (Hastie et al., 2022). Similarly, one may easily show that *data augmentation can lead to a divergence in risk even when $p_x$ is perfectly symmetric*. Namely, at the interpolation threshold $d_0/n \to \gamma_0 = 1$, effective dimension (consisting of invariant features) equals sample size.

**Fact 1.** *For identity covariance $\Sigma = I$, in the $\lambda \to 0$ limit we have asymptotic risk $R(\hat{\beta}_{\text{inv}}) = (1 - \gamma_0^{-1}) + \sigma^2(\gamma_0^{-1}/(1 - \gamma_0^{-1}))$, which explodes as $\gamma_0 \to 1$.*

To try to isolate the effect of the interpolation threshold from that of symmetry-breaking, we suppose $\gamma_0 > 1$, i.e. there are many possible invariant features to choose from. For tractability, we consider a minimal model for the covariance. Letting $d_c < \min(d_0, d - d_0)$ be the number of strong "coupling modes," let $\Sigma = (\sigma_c - \sigma_w)\sum_{k=1}^{d_c} u_k u_k^\top + \sigma_w I$, where $\sigma_c > \sigma_w$ are the coupling and weak (or "white") eigenvalues, and $u_k = (v_{0,k} + v_{\perp,k})/\sqrt{2}$ are perfect superpositions of orthogonal basis elements of $V_0$ and $V_\perp$. We consider $\sigma_w \to 0$ as the limit of strong correlations. Using random matrix theory (Atanasov et al., 2024b; Bach, 2024b), we characterize the asymptotic risk for data augmentation (Appendix A.7).[4] We find that data augmentation is *guaranteed* to perform worse when # correlational modes $\ll$ ambient dimension (Figure 3), while in other settings the story is more complex.

**Theorem 2.** *Let $d_c/n \to \gamma_c$ and consider the ridgeless limit $\lambda \to 0$, and $n, d \to \infty$. In the limit of strong correlations: (i) if $\gamma_c < 1$, both methods are unbiased and data augmentation has larger variance; (ii) For $\gamma_c > 1$, both methods have bias $C(\beta)\|\beta\|^2(\gamma_c - 1)/2\gamma_c$ where $C(\beta)$ (eq. (56)) is an explicit constant measuring how much of $\beta$ lies in the coupling subspace, and if moreover $\gamma_0 - \gamma_c/2 < 1/2$, then data augmentation has larger variance for small $\sigma_w > 0$.*

---

[4]This involves a version of the "two-point" deterministic equivalence studied by Atanasov et al. (2024a), of which we provide a different proof.

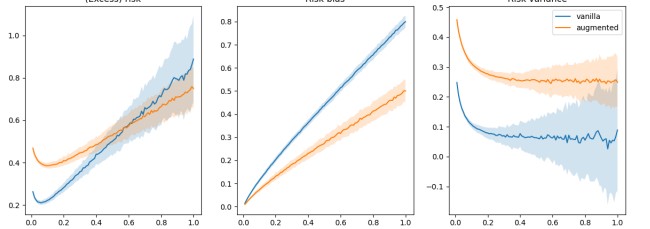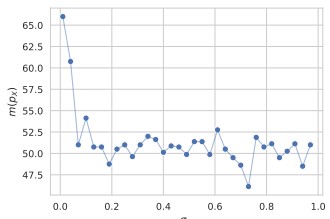

Figure 3: **Left:** Results in our minimal model for the over-parameterized regime, with mean and standard deviation across 200 trials. Small $\sigma_w$ corresponds to strong correlations between invariant and non-invariant features, and large $\sigma_w$ to no distributional symmetry breaking. In agreement with our theory, data augmentation is harmful in small $\sigma_w$ setting, as the excess risk is higher. See Appendix A.8.2 for details on the group and hyperparameters. **Right:** Corresponding values of $m(p_X)$ at varying $\sigma_w$, accurately reflecting the change in distributional symmetry-breaking.

Table 1: Comparison of train/test augmentation, group-averaged, and equivariant models across datasets. Augmentation: TT = train+test, TF = train only, FT = test only, FF = none. MNIST uses a $C_4$ group-averaged model; other datasets use stochastic group-averaging. MAE is reported for QM7b/QM9; equivariant baselines from e3nn Geiger et al. (2022). Best overall in bold, best within augmentation underlined. CNN used for MNIST, graph transformer for point clouds (Shi et al., 2022; Ying et al., 2021). See Figure 8 for results relative to FF. Results are averaged over 3 random seeds and $\pm$ 1 standard deviation is reported in subscript.

| Setting / Dataset Units | QM7b $\vec{\mu}$ $10^{-3}$ a.u. ($\downarrow$) | QM7b $\alpha_{iso}$ $10^{-1}a_0^3$ ($\downarrow$) | QM9 $C_v$ mcal/mol K ($\downarrow$) | QM9 $|\vec{\mu}|$ $10^{-3}$ D ($\downarrow$) | QM9 $\Delta\varepsilon$ meV ($\downarrow$) | MNIST % ($\uparrow$) | ModelNet40 % ($\uparrow$) |
|---|---|---|---|---|---|---|---|
| Equivariant | **41**$_{\pm2}$ | 6.8$_{\pm0.3}$ | **110**$_{\pm5}$ | **131**$_{\pm7}$ | **151**$_{\pm2}$ | 98.0$_{\pm0.4}$ | 60.06$_{\pm0.3}$ |
| Group Averaged | 43$_{\pm2}$ | **5.3**$_{\pm0.4}$ | 128$_{\pm7}$ | 217$_{\pm8}$ | 166$_{\pm3}$ | 98.0$_{\pm0.4}$ | 61.87$_{\pm1.3}$ |
| TT | 53$_{\pm5}$ | $\underline{5.5}_{\pm0.4}$ | 147$_{\pm3}$ | $\underline{243}_{\pm1}$ | $\underline{171}_{\pm1}$ | 97.6$_{\pm0.2}$ | 62.39$_{\pm0.4}$ |
| FF | 100$_{\pm1}$ | 7.2$_{\pm0.2}$ | $\underline{143}_{\pm6}$ | 275$_{\pm10}$ | 181$_{\pm6}$ | $\underline{\mathbf{98.8}}_{\pm0.1}$ | $\underline{\mathbf{78.14}}_{\pm0.3}$ |
| TF | $\underline{52}_{\pm5}$ | $\underline{5.5}_{\pm0.4}$ | 146$_{\pm3}$ | 245$_{\pm1}$ | 172$_{\pm1}$ | 97.6$_{\pm0.2}$ | 61.97$_{\pm0.7}$ |
| FT | 161$_{\pm6}$ | 12.5$_{\pm0.9}$ | 210$_{\pm50}$ | 471$_{\pm11}$ | 278$_{\pm10}$ | 40.7$_{\pm0.2}$ | 16.91$_{\pm0.8}$ |
| $m(p_X)$ (%) | 89.66$_{\pm1.02}$ | | 98.3$_{\pm0.1}$ | | | 86.9$_{\pm0.3}$ | 93.9$_{\pm0.7}$ |

# 5 EXPERIMENTS

Our theoretical analysis suggests that equivariant methods can be detrimental under distributional symmetry breaking; we now investigate this phenomenon on widely-used datasets. The experiments serve multiple goals. First, we *validate* $m(p_X)$ for quantifying distributional symmetry breaking by synthetically transforming subsets of datasets, verifying that $m(p_X)$ has the correct behavior (Figure 4). Second, we compute $m(p_X)$ to *investigate* the degree of distributional symmetry breaking in several benchmark datasets across domains, and detect high levels of symmetry bias, or distributional symmetry breaking. To understand downstream implications, we then compare equivariant and non-equivariant methods on the datasets' associated regression tasks, testing the predictive power of our theory (Table 1).[5] We expect that, due to the distribution shift induced by augmentation on highly canonicalized datasets, training augmentation will hurt performance. However, this is not the case for molecular datasets QM7b and QM9. These counterintuitive results motivate further investigation of task-dependent (Table 2) and local (Figure 5) distributional symmetry breaking, after which a more coherent picture emerges (Figure 6). See Appendix D for further experimental details and results. We now discuss each dataset in turn.

---

[5]The goal of this experiment is not to achieve state-of-the-art performance, but to study the impact of different augmentation settings across varied datasets and symmetry conditions.

We start with **MNIST** (Deng, 2012), where digits should intuitively be mostly canonicalized with respect to $90°$ rotations ($C_4$). $m(p_X)$ verifies this, showing that transformed and untransformed samples can be distinguished with nearly optimal ($1 - \frac{1}{2*4} = 87.5\%$) accuracy, matching the calculation from Section 2. We further sanity check $m(p_X)$ by rotating $p$-fractions of the dataset (Figure 12), where it achieves nearly optimal accuracy at intermediate levels of canonicalization, too. This is a relatively easy task, so there is not a large difference between augmentation settings, although the no augmentation (FF) setting does perform slightly better (Table 1).

Moving from 2D images to 3D shape classification, **ModelNet40** (Wu et al., 2015) provides a more complex benchmark dataset, consisting of 12,311 CAD models across 40 common object categories. The version most commonly used in recent works is a pre-aligned variant (Sedaghat et al., 2016), as confirmed by high $m(p_X)$. We also apply the metric per class (Figure 9), indicating that certain classes are more canonicalized than others. Consistent with our intuition, the FF setting outperforms other augmentation strategies, suggesting that training augmentation can destroy useful information contained in orientations.

Shifting to molecular property prediction, **QM9** consists of 133k small stable organic molecules with $\leq 9$ heavy atoms, together with scalar quantum mechanical properties (Ramakrishnan et al., 2014; Wu et al., 2017). $m(p_X)$ shows that QM9 is highly canonicalized with respect to rotations; see also Figure 2. The molecular conformers were generated using the commercial software CORINA Wu et al. (2017), which contains options to align SMILES strings by default (Sadowski et al., 1994; Schwab, 2010; Molecular Networks Altamira), an example of user-defined canonicalization (as in Figure 1) where we do not have direct acccs to the canonicalization function. Analyzing the decision boundary of $m(p_X)$ allows for fine-grained analysis of this unknown canonicalization, and can be used to probe the canonicalization for discontinuities (see Appendix D.4). We find that the degree to which equivariance is beneficial varies per property (also seen in e.g. Liao & Smidt (2022); see Table 4 for all properties), *yet for nearly all properties, training augmentation/equivariance still helps (or does not significantly hurt) performance, even on the original test set!* This is akin to computer vision, where augmentations like blurring induce distribution shift yet still aid performance. We next consider a molecular dataset with non-scalar labels to further investigate.

**QM7b** is a 7,211 molecule subset of GDB-13 (a database of stable and synthetically accessible organic molecules) composed of molecules with $\leq 7$ heavy atoms (Blum & Reymond, 2009; Montavon et al., 2013). We use a version of the dataset (Yang et al., 2019) containing non-scalar material response properties to explore how distributional symmetry breaking affects prediction of non-scalar geometric quantities. $m(p_X)$ demonstrates a high degree of distributional symmetry breaking, which we believe follows from pre-processing steps reported in Yang et al. (2019), such as using a kernel-based similarity metric to arrange atoms. We find that equivariance and augmentation are particularly beneficial for predicting the vector dipole moment ($\vec{\mu}$), more so than for scalar properties in the dataset (see Figure 8); nevertheless, augmentation again improves performance for both types of properties. Thus, we see a discrepancy between ModelNet40 and QM9/QM7b, in agreement with the theory that equivariance can be helpful or harmful depending on the dataset.

Before diving into this phenomenon, we finish quantifying symmetry bias in additional materials science datasets (including an LLM dataset) to demonstrate the utility of $m(p_X)$.

We explore two large scale materials datasets for predicting molecular energies and forces: **rMD17**, containing 100k structures from molecular dynamics simulations, and **OC20**, consisting of adsorbates placed on periodic crystalline catalysts (Christensen & von Lilienfeld, 2020; Chanussot* et al., 2021). Interestingly, the degree of distributional symmetry breaking varies widely between molecules in MD17 (Figure 4; see Figure 35 for all molecules). We hypothesize that this is both due to the initial conditions for the simulation, and the differing physical structures of each molecule. For OC20, both the adsorbate and the adsorbate + catalyst are highly canonicalized, likely due to the catalyst's alignment with the *xy* plane.

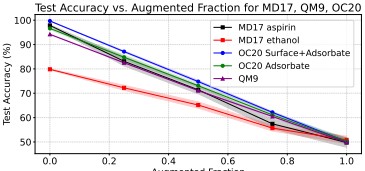

Figure 4: Test accuracy vs rotated fraction for aspirin and ethanol from rMD17, OC20 surface+adsorbate, OC20 adsorbate, and QM9.

Finally, we explore an **LLM materials dataset**, as there is growing interest in training large language models (LLMs) on diverse datatypes, including molecular data. To this end, Gruver et al.

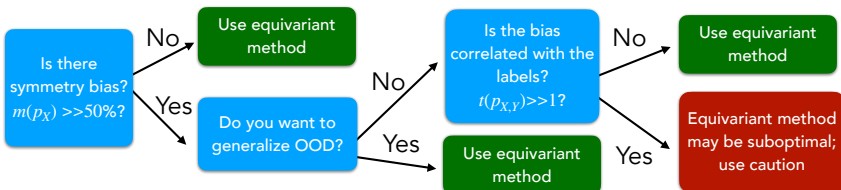

| $m(p_X)$(%) | Local QM9 | Global QM9 |
|---|---|---|
| Original | $65.9 \pm 0.3$ | $98.3 \pm 0.1$ |
| Rotated | $50.1 \pm 0.1$ | $50.0 \pm 0.3$ |
| Canonicalized | $99.8 \pm 0.04$ | $99.9 \pm 0.02$ |

| N | $m(p_X)$ (%, 5 runs) |
|---|---|
| 10 | $55.6\% \pm 8.44$ |
| 50 | $72.7\% \pm 4.51$ |
| 100 | $81.5\% \pm 4.79$ |
| 200 | $88.8\% \pm 0.62$ |
| 500 | $93.6\% \pm 0.39$ |
| 700 | $96.5\% \pm 0.12$ |
| 1024 (global) | $96.6\% \pm 0.04$ |

Figure 5: Left: The local QM9 dataset (top) and results (bottom). Right: Local ModelNet40 results.

Figure 6: Practical flowchart describing, if the unknown function is equivariant, what to expect from equivariant methods as a function of $m(p_X)$ and $t(p_{X,Y})$.

(2024) convert crystals into a text format, which requires listing their atoms in some ordering, and then train an LLM to generate new crystal structures. The authors independently noted that permutation augmentations hurt generative performance, even though the task is ostensibly permutation invariant. We postulated that this phenomenon was due to distributional symmetry-breaking, i.e. conventions in the generation of atom order. We thus trained a classifier head on a pretrained DistilBERT transformer to distinguish between permuted and unpermuted datapoints, and found $m(p_X) = 95\%$ accuracy. (Indeed, Figure 2 in Gruver et al. (2024) reveals clear ordering in the atoms; but with thousands of datapoints, a systematic test is useful for quantitative verification.)

In summary, our experiments thus far show that many benchmark point cloud datasets are actually quite aligned[6]. We emphasize that $m(p_X)$ is an easy-to-train metric that ML practitioners can use to detect and quantify distributional symmetry breaking, which is often not known *a priori*. Surprisingly, even though all datasets have high degrees of symmetry bias, the relative performance of data augmentation varies by task: train-time augmentation on ModelNet40 and MNIST hurts test-time performance on the original test set ("TF") relative to training without augmentations ("FF"), while train-time augmentation on QM9/QM7b does not hurt, and often even helps (for some properties), on the unaugmented test set ("TF")! We now explore two hypotheses for this differing behavior.

## 5.1 TASK-DEPENDENT METRIC

As discussed in Section 5.1, if the symmetry bias in a dataset is predictive of a particular task's labels, then training augmentation might be particularly harmful. We use $t(p_{X,Y})$ to assess this hypothesis. To validate $t(p_{X,Y})$, we first explore an artificial **QM7b dipole** canonicalization. This is a constructed example of a very task-relevant canonicalization, where molecules are aligned to place their dipole moments on the $z$ axis. This makes it trivial for a non-equivariant model to predict dipoles, while an equivariant model cannot exploit this alignment. Table 9 provides empirically confirmation: the FF augmentation setting outperforms an equivariant model. $t(p_{X,Y})$ is large and captures this behavior (Table 2). Similarly, **ModelNet** was another case where equivariance harmed performance, and $t(p_{X,Y})$ is significantly greater than 1. In contrast, for the **QM9** properties shown, the metric shows a relatively small signal. Thus, $t(p_{X,Y})$ is generally larger for tasks where equivariance hurts performance, leading us to the recommendations of Figure 6. In the future, we aim to more rigorously determine whether there exists a threshold above which equivariance is unlikely to provide benefits.

---

[6]Although high $m(p_X)$ does not precisely mean the datasets are perfectly canonicalized, particularly for infinite groups like $SO(3)$, it does mean that datapoints have clear, sparse preferred orientations.

Table 2: Task-dependent metric $t(p_{X,Y})$: Accuracy (ModelNet) or MAE (QM7b/QM9) of predicting $f(x)$ from $c(x)$, versus a random baseline. $t(p_{X,Y})$ shows how $\mathcal{L}$ deteriorates under rotation: $\mathcal{L}_{\text{rot}}/\mathcal{L}$ for loss, and $\mathcal{L}/\mathcal{L}_{\text{rot}}$ for accuracy. Values are averaged over five seeds.

| Dataset | $\mathcal{L}$ | $\mathcal{L}_{\text{rot}}$ | Relative Improvement $\mathcal{L}$ Compared to $\mathcal{L}_{\text{rot}}$ |
|---|---|---|---|
| QM7b Dipole $\mu$ ($\downarrow$) | $0.128 \pm .006$ | $0.39 \pm 0.04$ | $\mathbf{3.54 \pm 0.64}$ |
| QM7b Orig $\mu$ ($\downarrow$) | $0.38 \pm .04$ | $0.39 \pm 0.04$ | $1.03 \pm .03$ |
| ModelNet ($\uparrow$) | $12.5 \pm 0.1$ | $8.9 \pm 0.5$ | $\mathbf{1.41 \pm 0.08}$ |
| QM9 $C_v$ ($\downarrow$) | $3.07 \pm .03$ | $3.22 \pm .02$ | $1.05 \pm .01$ |
| QM9 $|\vec{\mu}|$ ($\downarrow$) | $1.14 \pm .01$ | $1.17 \pm .006$ | $1.02 \pm .01$ |
| QM9 $\Delta\varepsilon$ ($\downarrow$) | $1.02 \pm 0.01$ | $1.073 \pm .006$ | $1.048 \pm 0.005$ |

## 5.2 LOCALITY EXPERIMENTS

$t(p_{X,Y})$ thus signals when the symmetry bias of a dataset is correlated with its labels, providing an explanation for why equivariant methods might hurt. However, this doesn't explain why equivariant methods can perform so well on datasets where $t(p_{X,Y})$ is low. One hypothesis is that the features are equivariant functions of their receptive fields, meaning equivariant CNNs and GNNs naturally have local equivariance (Musaelian et al., 2023). It may be useful to compute locally equivariant features, e.g. featurizations of small, recurrent chemical motifs in molecules, rather than just globally equivariant features (Du et al., 2022; Lippmann et al., 2025). This provides a plausible explanation for the effectiveness of equivariant methods on highly canonicalized datasets such as QM9. Moreover, augmenting inputs to a local (e.g. message-passing) architecture implicitly conveys a bias towards local equivariance. While it is challenging to establish a causal link, we can use $m(p_X)$ to at least quantify the hypothesis that local motifs are comparatively more isotropic in orientation.

Concretely, we generate the local QM9 dataset by extracting local neighborhoods (by bonds) from each QM9 molecule (see Appendix D.6 for details). In Figure 5, we compare $m(p_X)$ between local and ordinary QM9 in three settings: the original datasets (exploration), and under random rotation and manual canonicalization (as sanity checks, which should and do yield 50% and 100%). We find that the detection accuracy is much lower for local QM9, indicating a lower degree of local distributional symmetry breaking! For ModelNet40, we analogously constructed a local dataset by randomly selecting one point from each original, 1024-point point cloud, and then collecting its $N$ nearest neighbors. When the number of sampled points is small, the metric drops significantly, indicating that local regions of the point clouds are not inherently canonicalized; this effect reduces with the size of the neighborhood. These findings suggest that the symmetry bias present in some point cloud benchmarks is weaker at the local scale, which may partially explain the success of equivariant methods. Investigating this hypothesis further is an interesting direction for future work.

## 6 CONCLUSION

In this work, we aimed to provide both empirical and theoretical analysis of distributional asymmetry and its implications for learning. Our interpretable metrics quantify the degree of symmetry-breaking present in a dataset without using any specific knowledge of the domain, thus providing practioners with a simple diagnostic for detecting distributional symmetry breaking (symmetry bias) in their datasets. Experiments revealed a high degree of symmetry-breaking in every benchmark dataset, yet augmentation only really impeded (test) performance for ModelNet40.

Overall, these findings have intriguing implications for equivariant learning. First, they affirm that if evaluated only on in-distribution validation data, non-equivariant models may appear accurate, yet fail to generalize under transformations. Assessing whether this is problematic requires domain expertise: see Figure 6 for a practical flowchart. Moreover, *applying* canonicalization to data has been proposed as a flexible method for making black-box models globally equivariant (Kaba et al., 2023). However, if molecular datasets are already nearly canonicalized yet still experience benefits from equivariance, this suggests that equivariant networks and augmentation may provide some *additional*, possibly *domain-specific* benefit beyond global equivariance that is currently unexplained. Finally, data augmentation is often considered universally beneficial for invariant tasks, yet we show that it can sometimes hurt performance on the test set.

**Acknowledgments**    We thank Julia Balla, Ameya Daigavane, Oumar Kaba, Teddy Koker, Mit Kotak, Yi-Lun Liao, Ryley McConkey, Ankur Moitra, Samuel Stanton, Behrooz Tahmasebi, Yan Zhang, and the anonymous reviewers for useful discussions and feedback.

R.W. would like to acknowledge support from NSF Grants 2442658 and 2134178. This work is supported by the National Science Foundation under Cooperative Agreement PHY-2019786 (The NSF AI Institute for Artificial Intelligence and Fundamental Interactions, `iaifi.org`). H.L. is supported by the Fannie and John Hertz Foundation. E.H was supported by the U.S. Department of Energy, Office of Science, Office of Advanced Scientific Computing Research, Department of Energy Computational Science Graduate Fellowship under Award Number DE-SC0024386. This research used resources of the National Energy Research Scientific Computing Center (NERSC), a Department of Energy User Facility using NERSC award ACSR-ERCAP0033254.

This report was prepared as an account of work sponsored by an agency of the United States Government. Neither the United States Government nor any agency thereof, nor any of their employees, makes any warranty, express or implied, or assumes any legal liability or responsibility for the accuracy, completeness, or usefulness of any information, apparatus, product, or process disclosed, or represents that its use would not infringe privately owned rights. Reference herein to any specific commercial product, process, or service by trade name, trademark, manufacturer, or otherwise does not necessarily constitute or imply its endorsement, recommendation, or favoring by the United States Government or any agency thereof. The views and opinions of authors expressed herein do not necessarily state or reflect those of the United States Government or any agency thereof.

**Reproducibility Statement**    We describe experimental and model details in Appendix D. Our code is also publicly available at `github.com/hannahlawrence/dist-symm-breaking`, and can be used to reproduce our experiments or try a new dataset.

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

APPENDICES

# A  THEORY

In this section, we elaborate on the theory of the main paper, including both more context and proofs of results. Most of the earlier parts are dedicated to our study of ridge regression in Section 4.

- Appendix A.1: we formalize the motivating intuition for the paper, i.e. that "equivariant methods lose information". In particular, we prove that under an invariant loss function, gradient descent on an equivariant architecture yields exactly the same learned parameters, regardless of the group transformations applied to the training data.

- Appendix A.2: we review the result of Elesedy & Zaidi (2021), which says that when data is invariant in distribution, test symmetrization always improves generalization error. (Unlike our analysis, this holds even in the non-linear setting.)

- Appendix A.3: we show that using invariant features, or equivalently, restricting to invariant estimators — which we call *training symmetrization* — is equivalent to data augmentation in the ridge regression setting.

- Appendix A.4: we record the bias-variance decompositions of risk for vanilla ridge regressions, test-time symmetrization, and data augmentation.

- Appendix A.5: we prove Theorem 1, which generalizes Elesedy & Zaidi (2021, Theorem 7) to non-invariant data, and to data augmentation.

- Appendix A.6: we demonstrate in an explicit example that test-time symmetrization can arbitrarily increase risk when the data distribution is not invariant (in the under-parameterized regime).

- Appendix A.7: using random matrix theory, we derive asymptotic expressions ("deterministic equivalents") for the bias and variance of each of our three estimators in the over-parameterized regime. Fact 1 is a direct corollary.

- Appendix A.8: we analytically study our minimal model of covariance, proving Theorem 2. We confirm our results empirically, as shown in Figure 3.

### A.1 GRADIENT DESCENT ON EQUIVARIANT NETWORKS IS INVARIANT UNDER PER-DATAPOINT TRANSFORMATIONS

The guiding motivation for this paper is that symmetry bias (i.e. distributional symmetry breaking) contains information, which is lost by an equivariant method. This is self-evident if that equivariant method consists of data augmentation (since the orientation information is literally thrown away) or a network that operates on invariant features, but is slightly more subtle for a generic equivariant function. Here, we formalize our assertion by proving that, when using gradient descent to learn the parameters of an equivariant network under a group-invariant loss, the gradient update at each step is agnostic to the exact orientation of each input datapoint. In other words, an equivariant network trained by gradient descent learns the same solution with a symmetry-biased dataset and with a fully-augmented version of that dataset; thus, it too is "losing" information contained in the distribution's asymmetry. (In fact, this is true not just for gradient descent, but for most optimization methods, since we'll show the loss landscape is unchanged.) We now formalize this below.

To begin, let $G$ be a group acting linearly on input space $V_0$ and output space $V_L$. We write $g \cdot x$ for the action of $g \in G$ on a vector in either space.

Let $\mathrm{NN}_\theta : V_0 \to V_L$ be a $G$-equivariant neural network, differentiable in $\theta \in \Theta \subseteq \mathbb{R}^p$:

$$\mathrm{NN}_\theta(g \cdot x) = g \cdot \mathrm{NN}_\theta(x) \qquad \forall g \in G, \ \forall x \in V_0. \tag{1}$$

Let the loss $\ell : V_L \times V_L \to \mathbb{R}$ be differentiable in its first argument and $G$-invariant:

$$\ell(g \cdot u, \ g \cdot v) = \ell(u, v) \qquad \forall g \in G, \ \forall (u, v) \in V_L \times V_L. \tag{2}$$

Let $S_1 = \{(x_i, y_i)\}_{i=1}^n$ be a training set. Fix arbitrary $g_1, \dots, g_n \in G$ and define

$$S_2 = \{(g_i \cdot x_i, \ g_i \cdot y_i)\}_{i=1}^n,$$

with empirical risks

$$\mathcal{L}_{S_k}(\theta) := \frac{1}{n} \sum_{i=1}^n \ell\big(\mathrm{NN}_\theta(x_i^{(k)}), \ y_i^{(k)}\big), \qquad k \in \{1, 2\}.$$

First, we have the relatively obvious statement of loss invariance.

**Lemma 1** (Per-example loss invariance). *For each i and all $\theta$,*

$$\ell\big(\mathrm{NN}_\theta(g_i \cdot x_i), \ g_i \cdot y_i\big) = \ell\big(\mathrm{NN}_\theta(x_i), \ y_i\big). \tag{3}$$

*Proof.* By equation 1 and equation 2:

$$\ell\big(\mathrm{NN}_\theta(g_i \cdot x_i), \ g_i \cdot y_i\big) = \ell\big(g_i \cdot \mathrm{NN}_\theta(x_i), \ g_i \cdot y_i\big) = \ell\big(\mathrm{NN}_\theta(x_i), \ y_i\big). \qquad \square$$

Using Lemma 1, we have that the two loss functions are equal, and so their gradients are too.

**Corollary 1** (Equality of risks and gradients). *For all $\theta$,*

$$\mathcal{L}_{S_2}(\theta) = \mathcal{L}_{S_1}(\theta) \qquad and \qquad \nabla_\theta \mathcal{L}_{S_2}(\theta) = \nabla_\theta \mathcal{L}_{S_1}(\theta).$$

At this point, the key result is fairly clear, but we state it below for completeness.

*Proof.* Lemma 1 gives pointwise equality of per-example losses as functions of $\theta$. Average for the risks; differentiate for the gradients. $\qquad \square$

**Theorem 3** (Identical optimization trajectories). *Fix $\{\eta_t\}_{t \geq 0}$ and $\theta_0 \in \Theta$. Define*

$$\theta_{t+1}^{(k)} := \theta_t^{(k)} - \eta_t \nabla_\theta \mathcal{L}_{S_k}\big(\theta_t^{(k)}\big), \qquad k \in \{1, 2\},$$

*with $\theta_0^{(1)} = \theta_0^{(2)} = \theta_0$. Then $\theta_t^{(1)} = \theta_t^{(2)}$ for all $t \geq 0$.*

*Proof.* By induction. The base case holds by construction. If $\theta_t^{(1)} = \theta_t^{(2)}$, then Corollary 1 gives equal gradients, so the updates coincide. $\qquad \square$

It is important to note here that the $g_i$ are arbitrary per-example; no common global transformation is assumed.

## A.2 REVIEW OF THE GENERALIZATION GAP OF ELESEDY & ZAIDI (2021)

Consider data $(x, y) \in \mathbb{R}^d \times \mathbb{R}$ generated as $y = f^*(x) + \varepsilon$ for $x \sim p_x$, a ground-truth invariant function $f^*$, and independent mean-zero finite-variance noise $\varepsilon$. Considering $L^2$ loss, the excess risk of a given function $f$ (say the result of learning on some fixed training dataset) is

$$R(f) = \mathbb{E}[(y - f(x))^2] - \mathbb{E}[(y - f^*(x))^2] = \mathbb{E}[(f(x) - f^*(x))^2] \tag{4}$$

since $\mathbb{E}[\varepsilon(f(x) - f^*(x)] = 0$. We define a new inner product on functions, $\langle f_1, f_2 \rangle_{p_x} = \mathbb{E}[f_1(x)f_2(x)]$. The excess risk of $f$ is then $\|f - f^*\|_{p_x}^2$, with the norm induced by this inner product.

Let $\bar{f}(x) = \mathbb{E}_g[f(gx)]$ be the symmetrization of $f$ with respect to uniformly random $g \in G$. We can think of this as test-time augmentation. We can ask what the difference is between the excess risk of $f$ and $\bar{f}$,

$$\Delta(f, \bar{f}) := \|f - f^*\|_{p_x}^2 - \|\bar{f} - f^*\|_{p_x}^2 = -2 \left\langle f^* - \bar{f}, f - \bar{f} \right\rangle_{p_x} + \|f - \bar{f}\|_{p_x}^2. \tag{5}$$

Elesedy and Zaidi show that when $x$ is invariant in distribution, $f - \bar{f}$ is orthogonal (in the inner product defined above) to invariant functions, and thus in particular to $f^* - \bar{f}$. In this case $\Delta(f, \bar{f}) \geq 0$, meaning for any $f$ one can always achieve better generalization using $\bar{f}$. When $p_x$ is not invariant, however, the inner product might make the overall expression negative. This case thus warrants further investigation.

## A.3 EQUIVALENCE OF TRAINING SYMMETRIZATION AND DATA AUGMENTATION

We consider the estimator obtained by infinitely many augmentations,

$$\hat{\beta}_{\lambda, \text{aug}} = \arg\min_\beta \frac{1}{n} \sum_{i=1}^n \mathbb{E}_g[(y_i - (gx_i)^\top \beta)^2] + \lambda \|\beta\|^2 = (\hat{\Sigma}_{\text{aug}} + \lambda I)^{-1} \hat{\Sigma}_{yx, \text{inv}} \tag{6}$$

where $\hat{\Sigma}_{\text{aug}} = \mathbb{E}_g[g\hat{\Sigma}g^\top]$. As a linear map $\mathbb{R}^d \to \mathbb{R}^d$, $\hat{\Sigma}_{\text{aug}}$ is equivariant, so by Schur's lemma it is block-diagonal in $V_0, V_\perp$ (or more generally, in irreps). Since $\hat{\Sigma}_{yx, \text{inv}}$ is non-zero only in the $V_0$ component, $(\hat{\Sigma}_{\text{aug}} + \lambda I)^{-1} \hat{\Sigma}_{yx, \text{inv}} = (\hat{\Sigma}_{\text{inv}} + \lambda I)^{-1} \hat{\Sigma}_{yx, \text{inv}}$, and thus $\hat{\beta}_{\lambda, \text{aug}} = \hat{\beta}_{\lambda, \text{inv}}$. We therefore refer to $\hat{\beta}_{\lambda, \text{inv}}$ interchangeably as using data augmentation or invariant features.

## A.4 BIAS-VARIANCE DECOMPOSITIONS

The risk of any estimator $\hat{\beta}$ has a bias-variance decomposition $R_X(\hat{\beta}) = B_X(\hat{\beta}) + V_X(\hat{\beta})$ with

$$B_X(\hat{\beta}) = \left\| \mathbb{E}\left[\hat{\beta} \mid X\right] - \beta \right\|_\Sigma^2 \qquad V_X(\hat{\beta}) = \text{Tr}(\text{Cov}(\hat{\beta} \mid X)\Sigma) \tag{7}$$

In the case of vanilla ridge(less) regression, the expressions above have well-known and easily derived forms (Bach, 2024a; Hastie et al., 2022). We list the equivalent expressions for test-time symmetrization and data augmentation below. The proof is standard, being a simple expansion of definitions. One may notice that the expressions are the same as the vanilla case except (1) test-time symmetrization replaces $\Sigma$ with $\Sigma_{\text{inv}} = P_0 \Sigma P_0$, and (2) data augmentation replaces $\hat{\Sigma}$ with $\hat{\Sigma}_{\text{inv}}$.

**Lemma 2.** *For unaugmented ridge(less) regression, the bias and variance terms are standard:*

$$B_X(\hat{\beta}_\lambda) = \lambda^2 \beta^\top (\hat{\Sigma} + \lambda I)^{-1} \Sigma (\hat{\Sigma} + \lambda I)^{-1} \beta \qquad V_X(\hat{\beta}_\lambda) = \frac{\sigma^2}{n} \text{Tr}(\hat{\Sigma}(\hat{\Sigma} + \lambda I)^{-2} \Sigma) \tag{8}$$

$$B_X(\hat{\beta}) = \beta^\top \Pi \Sigma \Pi \beta \qquad V_X(\hat{\beta}) = \frac{\sigma^2}{n} \text{Tr}(\hat{\Sigma}^+ \Sigma) \tag{9}$$

*where $\Pi = I - \hat{\Sigma}^+ \hat{\Sigma}$ projects onto the null space of $X$. Test-time symmetrization gives*

$$B_X(P_0 \hat{\beta}_\lambda) = \lambda^2 \beta^\top (\hat{\Sigma} + \lambda I)^{-1} \Sigma_{\text{inv}} (\hat{\Sigma} + \lambda I)^{-1} \beta \qquad V_X(P_0 \hat{\beta}_\lambda) = \frac{\sigma^2}{n} \text{Tr}(\hat{\Sigma}(\hat{\Sigma} + \lambda I)^{-2} \Sigma_{\text{inv}}) \tag{10}$$

$$B_X(P_0 \hat{\beta}) = \beta^\top \Pi \Sigma_{\text{inv}} \Pi \beta \qquad V_X(P_0 \hat{\beta}) = \frac{\sigma^2}{n} \text{Tr}(\hat{\Sigma}^+ \Sigma_{\text{inv}}), \tag{11}$$

*whereas for invariant features and data augmentation, we obtain*

$$B_X(\hat{\beta}_{\lambda,\text{inv}}) = \lambda^2 \beta^\top (\hat{\Sigma}_{\text{inv}} + \lambda I)^{-1} \Sigma (\hat{\Sigma}_{\text{inv}} + \lambda I)^{-1} \beta \qquad V_X(\hat{\beta}_{\lambda,\text{inv}}) = \frac{\sigma^2}{n} \text{Tr}(\hat{\Sigma}_{\text{inv}} (\hat{\Sigma}_{\text{inv}} + \lambda I)^{-2} \Sigma) \quad (12)$$

$$B_X(\hat{\beta}_{\text{inv}}) = \beta^\top \Pi_{\text{inv}} \Sigma \Pi_{\text{inv}} \beta \qquad\qquad\qquad V_X(\hat{\beta}_{\text{inv}}) = \frac{\sigma^2}{n} \text{Tr}((\hat{\Sigma}_{\text{inv}})^+ \Sigma), \qquad\qquad (13)$$

*where* $\Pi_{\text{inv}} = P_0 - (\hat{\Sigma}_{\text{inv}})^+ \hat{\Sigma}_{\text{inv}}$. *In latter case, we note that every instance of* $\Sigma$ *can equivalently be replaced with* $\Sigma_{\text{inv}}$, *being multiplied "on both sides" by invariant objects.*

## A.5 PROOF OF THEOREM 1

When $d < n - 1$, the matrices $\Pi$ and $\Pi_{\text{inv}}$ defined in Lemma 2 are almost surely equal to the zero matrix. Thus the vanilla, test-time symmetrization, and data augmentation estimators are all unbiased, and we compare only their variances.

Having assumed $x_i \sim \mathcal{N}(0, \Sigma)$, the empirical covariance is a scaling of Wishart-distributed matrix: $n\hat{\Sigma} \sim \mathcal{W}(\Sigma, n)$. Since $\Sigma$ is full rank, the standard form for the expectation of the inverse Wishart $\frac{1}{n}\hat{\Sigma}^{-1} \sim \mathcal{W}^{-1}(\Sigma^{-1}, n)$ gives

$$\mathbb{E}[\hat{\Sigma}^{-1}] = \frac{n\Sigma^{-1}}{n - d - 1} \qquad \Rightarrow \qquad \mathbb{E}[V_X(\hat{\beta})] = \frac{\sigma^2 \text{Tr}(\Sigma^{-1}\Sigma)}{n - d - 1} = \frac{\sigma^2 d}{n - d - 1}. \qquad (14)$$

For test-time symmetrization, note that the trace term in the variance only depends on $V_0$ components of the inverse empirical covariance: $\text{Tr}(\hat{\Sigma}^{-1}\Sigma_{\text{inv}}) = \text{Tr}(P_0 \hat{\Sigma}^{-1} P_0 \Sigma_{\text{inv}})$. We thus use the fact that diagonal sub-matrices of inverse-Wishart matrices are inverse-Wishart of a certain form. Letting $V$ be the change of basis matrix into $V_0, V_\perp$, so that any matrix $M$ can be written as

$$V^\top M V = \begin{pmatrix} M_{00} & M_{0\perp} \\ M_{\perp 0} & M_{\perp\perp} \end{pmatrix}, \qquad\qquad (15)$$

we have $nV^\top \hat{\Sigma} V \sim \mathcal{W}(V^\top \Sigma V, n)$ and $\frac{1}{n}(\hat{\Sigma}^{-1})_{00} \sim \mathcal{W}^{-1}((\Sigma^{-1})_{00}, n - d_\perp)$ where $d_\perp = \dim V_\perp$. We thus have

$$\mathbb{E}[V_X(P_0 \hat{\beta})] = \frac{\sigma^2 \text{Tr}((\Sigma^{-1})_{00}\Sigma_{00})}{(n - d_\perp) - d_0 - 1} = \frac{\sigma^2 \text{Tr}(\Sigma^{-1}\Sigma_{\text{inv}})}{n - d - 1}. \qquad (16)$$

For invariant features, the relevant trace term is $\text{Tr}((\hat{\Sigma}_{\text{inv}})^+ \Sigma_{\text{inv}}) = \text{Tr}((\hat{\Sigma}_{00})^{-1}\Sigma_{00})$. The result follows the same logic as in the vanilla case: $n\hat{\Sigma}_{00} \sim \mathcal{W}(\Sigma_{00}, n)$, and thus

$$\mathbb{E}[(\hat{\Sigma}_{00})^{-1}] = \frac{n(\Sigma_{00})^{-1}}{n - d_0 - 1} \qquad \Rightarrow \qquad \mathbb{E}[V_X(\hat{\beta}_{\text{inv}})] = \frac{\sigma^2 \text{Tr}((\Sigma_{00})^{-1}(\Sigma_{00}))}{n - d_0 - 1} = \frac{\sigma^2 d_0}{n - d_0 - 1}. \qquad (17)$$

## A.6 PERMUTATION EXAMPLE

Consider the case of $G = S_3$ acting on three-dimensional inputs $x \in \mathbb{R}^3$ by permuting coordinates. Let $V$ be the change of basis matrix into the $G$-invariant subspaces $V_0, V_\perp$, and write $M_{00}$ for the $(V_0, V_0)$-block of $V^\top M V$. We then consider a covariance

$$V^\top \Sigma V = \begin{pmatrix} \sigma_{\text{inv}}^2 & \rho & \rho \\ \rho & 1 & \tau \\ \rho & \tau & 1 \end{pmatrix} \qquad\qquad \Rightarrow \qquad\qquad (\Sigma^{-1})_{00} = \frac{1}{\sigma_{\text{inv}}^2 - \frac{2\rho^2}{1+\tau}} \qquad (18)$$

such that $\text{Tr}(\Sigma^{-1}\Sigma_{\text{inv}}) = \left(1 - \frac{2\rho^2}{\sigma_{\text{inv}}^2(1+\tau)}\right)^{-1}$. This term is large when $|\rho|$, the correlation strength between invariant and non-invariant features, is large compared to the invariant signal $\sigma_{\text{inv}}^2$. In particular, we have $\mathbb{E}[R_X(\hat{\beta})] < \mathbb{E}[R_X(P_0\hat{\beta})] \to \infty$ as $2\rho^2$ grows from $\frac{2}{3}\sigma_{\text{inv}}^2(1+\tau)$ to $\sigma_{\text{inv}}^2(1+\tau)$.

While we do not do so here, this example can be extended to general $G$ and $\Sigma$ by using the Schur complement formula for $(\Sigma^{-1})_{00}$, in which case the "size" of $\Sigma_{0\perp}$ in the Loewner order plays the role of the correlation $\rho$.

## A.7 Deterministic equivalents for bias and variance

In the proportional asymptotic regime, where $n, d \to \infty$ and $d/n \to \gamma$, we leverage the notion of *deterministic equivalence* of possibly random matrices $A_n$ and $B_n$. In particular, we write $A_n \simeq B_n$ when for any matrices $C_n$ of bounded trace norm,

$$|\mathrm{Tr}((A_n - B_n)C_n)| \to 0. \tag{19}$$

In our derivations below, we take advantage of the "calculus of deterministic equivalents" as developed by Dobriban & Sheng (2018); Sheng & Dobriban (2020), as well as proof techniques of Hastie et al. (2022), which in turn rely on the generalized Marchenko-Pastur theorem of Rubio & Mestre (2011). We also utilize the notions of first- and second-order degrees of freedom, $\mathrm{df}^1$ and $\mathrm{df}^2$, used by Atanasov et al. (2024b); Bach (2024b),[7] and introduced by Caponnetto & De Vito (2007) as "effective dimension."

Our goal is to find deterministic equivalents for the matrix products appearing in the bias and variance expressions in Lemma 2 (for $\lambda > 0$), which all take the two forms

$$B_{\mu\nu} = \lambda^2 \beta^\top (\hat{\Sigma}_\mu + \lambda I)^{-1} \Sigma_\nu (\hat{\Sigma}_\mu + \lambda I)^{-1} \beta \qquad V_{\mu\nu} = \frac{\sigma^2}{n} \mathrm{Tr}((\hat{\Sigma}_\mu + \lambda I)^{-2} \hat{\Sigma}_\mu \Sigma_\nu) \tag{20}$$

where $\mu, \nu$ run over empty or inv subscripts. Here, we assume the setting of the generalized Marchenko-Pastur theorem — namely, that we have the deterministic equivalence

$$\lambda (\hat{\Sigma}_\mu + \lambda I)^{-1} \simeq \kappa_\mu (\Sigma_\mu + \kappa_\mu I)^{-1} \tag{21}$$

where $\kappa_\mu$ is the unique positive solution to

$$\kappa_\mu = \frac{\lambda}{1 - T_\mu(\kappa_\mu)} \qquad T_\mu(\kappa) = \frac{1}{n} \mathrm{df}_\mu^1(\kappa) = \frac{1}{n} \mathrm{Tr}((\Sigma_\mu + \kappa I)^{-1} \Sigma_\mu) \tag{22}$$

and can be seen as the effective or renormalized ridge parameter; this includes the setting of i.i.d. Gaussian data, but extends much further, to the *Gaussian universality* regime (Hastie et al., 2022; Zavatone-Veth, 2024). We prove the deterministic equivalences

$$B_{\mu\nu} \simeq \frac{\kappa_\mu^2 \alpha_{\mu\nu}}{1 - \alpha_{\mu\mu}} \beta^\top (\Sigma_\mu + \kappa_\mu I)^{-2} \Sigma_\mu \beta + \kappa_\mu^2 \beta^\top (\Sigma_\mu + \kappa_\mu I)^{-1} \Sigma_\nu (\Sigma_\mu + \kappa_\mu I)^{-1} \beta \tag{23}$$

$$V_{\mu\nu} \simeq \sigma^2 \frac{\alpha_{\mu\nu}}{1 - \alpha_{\mu\mu}} \tag{24}$$

where we define generalized second-order degrees of freedom,

$$\alpha_{\mu\nu} = \frac{1}{n} \mathrm{df}_{\mu\nu}^2(\kappa_\mu) \qquad \mathrm{df}_{\mu\nu}^2(\kappa) = \mathrm{Tr}((\Sigma_\mu + \kappa)^{-2} \Sigma_\mu \Sigma_\nu). \tag{25}$$

We do not claim our calculation of these deterministic equivalents is novel; indeed our presentation closely follows the notes of Zavatone-Veth (2024), and the "two-point" equivalences we consider were recently analyzed in the context of cross validation by Patil (2022) and Atanasov et al. (2024a). To our knowledge, however, our work is the first to apply these techniques to study invariant learning.

### A.7.1 Bias term

Note that

$$\lambda^2 (\hat{\Sigma}_\mu + \lambda I + \lambda \tau \Sigma_\nu)^{-1} \Sigma_\nu (\hat{\Sigma}_\mu + \lambda I + \lambda \tau \Sigma_\nu)^{-1} \Big|_{\tau=0} \tag{26}$$

$$= -\partial_\tau \lambda (\hat{\Sigma}_\mu + \lambda I + \lambda \tau \Sigma_\nu)^{-1} \Big|_{\tau=0} \tag{27}$$

We find a deterministic equivalent for the expression inside the derivative. First, note

$$\lambda (\hat{\Sigma}_\mu + \lambda I + \lambda \tau \Sigma_\nu)^{-1} = \lambda (I + \tau \Sigma_\nu)^{-1/2} (\hat{\Sigma}_\tau + \lambda I)^{-1} (I + \tau \Sigma_\nu)^{-1/2} \tag{28}$$

---

[7]Note that the notion used by Atanasov et al. (2024b) is scaled by $1/d$ with respect to that of Bach (2024b); we use the latter convention.

where we define

$$\hat{\Sigma}_\tau = (I + \tau\Sigma_v)^{-1/2}\hat{\Sigma}_\mu(I + \tau\Sigma_v)^{-1/2}, \tag{29}$$

That is, our expression is a product of deterministic matrices with the matrix ridge resolvent for a scaled version of the empirical covariance. This resolvent thus has the deterministic equivalent

$$\lambda(\hat{\Sigma}_\tau + \lambda I)^{-1} \simeq \kappa_\tau(\Sigma_\tau + \kappa_\tau I)^{-1} \tag{30}$$

where $\Sigma_\tau$ is the population covariance $\Sigma_\mu$ scaled in the same way as $\hat{\Sigma}_\tau$ is, and $\kappa_\tau$ is the unique positive solution to

$$\kappa_\tau = \frac{\lambda}{1 - T_\tau(\kappa_\tau)}. \tag{31}$$

We thus have

$$\lambda(\hat{\Sigma}_\mu + \lambda I + \lambda\tau\Sigma_v)^{-1} \simeq \kappa_\tau(\Sigma_\mu + \tau I + \kappa_\tau\tau\Sigma_v)^{-1}. \tag{32}$$

Under the assumption that $\Sigma_\mu$ is trace class, one can exchange the $n, d \to \infty$ limit and the derivative to obtain

$$\lambda^2(\hat{\Sigma}_\mu + \lambda I)^{-1}\Sigma_v(\hat{\Sigma}_\mu + \lambda I)^{-1} \simeq -\partial_\tau\kappa_\tau(\Sigma_\mu + \kappa_\tau I + \kappa_\tau\tau\Sigma_v)^{-1}\Big|_{\tau=0} \tag{33}$$

Let us write $\delta = \partial_\tau\kappa_\tau|_{\tau=0}$, and note that $\kappa_\tau|_{\tau=0} = \kappa_\mu$. We get, first using the matrix identity $\partial M^{-1} = -M^{-1}(\partial M)M^{-1}$, and then combining terms,

$$- \delta(\Sigma_\mu + \kappa_\mu I)^{-1} + \kappa_\mu(\Sigma_\mu + \kappa_\mu I)^{-1}(\delta I + \kappa_\mu\Sigma_v)(\Sigma_\mu + \kappa_\mu I)^{-1} \tag{34}$$

$$= - \delta\Sigma_\mu(\Sigma_\mu + \kappa_\mu I)^{-2} + \kappa_\mu^2(\Sigma_\mu + \kappa_\mu I)^{-1}\Sigma_v(\Sigma_\mu + \kappa_\mu I)^{-1}. \tag{35}$$

It remains to evaluate $\delta$, which we do by differentiating the fixed-point equation at $\tau = 0$,

$$\kappa_\tau - \kappa_\tau T_\tau = \lambda \qquad\qquad\rightarrow\qquad\qquad \delta - \delta T_\tau\Big|_{\tau=0} - \kappa_\mu\partial_\tau(T_\tau)\Big|_{\tau=0} = 0. \tag{36}$$

Recognizing $1 - T_\tau|_{\tau=0}$ as $\lambda/\kappa_\mu$ (from the fixed-point equation at $\tau = 0$), we get

$$\delta = \frac{\kappa_\mu^2}{\lambda}\partial_\tau(T_\tau)\Big|_{\tau=0} = -\frac{\kappa_\mu^2}{\lambda n}\delta\,\mathrm{Tr}((\Sigma_\mu + \kappa_\mu I)^{-2}\Sigma_\mu) - \frac{\kappa_\mu^3}{\lambda}\alpha_{\mu v} \tag{37}$$

Since $(\Sigma_\mu + \kappa_\mu I)^{-1} = (I + (\Sigma_\mu + \kappa_\mu I)^{-1}\Sigma_\mu)/\kappa$,

$$\frac{1}{n}\mathrm{Tr}((\Sigma_\mu + \kappa_\mu)^{-2}\Sigma_\mu) = \frac{1}{\kappa_\mu}(T_\tau|_{\tau=0} - \alpha_{\mu\mu}) = \frac{1}{\kappa_\mu}\left(1 - \frac{\lambda}{\kappa_\mu} - \alpha_{\mu\mu}\right). \tag{38}$$

Subsequently,

$$\delta = -\frac{\kappa_\mu\delta}{\lambda}\left(1 - \frac{\lambda}{\kappa_\mu} - \alpha_{\mu\mu}\right) - \frac{\kappa_\mu^3}{\lambda}\alpha_{\mu v} = -\left(\frac{\kappa_\mu}{\lambda}(1 - \alpha_{\mu\mu}) - 1\right)\delta - \frac{\kappa_\mu^3}{\lambda}\alpha_{\mu v}. \tag{39}$$

We can then solve,

$$\delta = -\frac{\lambda}{\kappa_\mu(1 - \alpha_{\mu\mu})}\frac{\kappa_\mu^3}{\lambda}\alpha_{\mu v} = -\kappa_\mu^2\frac{\alpha_{\mu v}}{1 - \alpha_{\mu\mu}}. \tag{40}$$

### A.7.2  VARIANCE TERM

We begin by noting that

$$\frac{\sigma^2}{n}\mathrm{Tr}((\hat{\Sigma}_\mu + \lambda I)^{-2}\hat{\Sigma}_\mu\Sigma_v) = -\frac{\sigma^2}{n}\partial_\lambda\,\mathrm{Tr}((\hat{\Sigma}_\mu + \lambda I)^{-1}\hat{\Sigma}_\mu\Sigma_v) = \frac{\sigma^2}{n}\partial_\lambda\lambda\,\mathrm{Tr}((\hat{\Sigma}_\mu + \lambda I)^{-1}\Sigma_v). \tag{41}$$

By assumption $\lambda(\hat{\Sigma}_\mu + \lambda I)^{-1} \simeq \kappa_\mu(\Sigma_\mu + \kappa_\mu I)^{-1}$. Exchanging limits and the derivative (which is justified by the assumption of bounded trace norm) we get the deterministic equivalent

$$\frac{1}{n}\partial_\lambda\kappa_\mu\,\mathrm{Tr}((\Sigma_\mu + \kappa_\mu I)^{-1}\Sigma_v) = \frac{1}{n}\partial_\lambda(\kappa_\mu)\,\mathrm{Tr}((\Sigma_\mu + \kappa_\mu I)^{-2}\Sigma_\mu\Sigma_v) = \partial_\lambda(\kappa_\mu)\alpha_{\mu v}. \tag{42}$$

To find the derivative, we differentiate the fixed-point equation

$$\kappa_\mu - \kappa_\mu T_\mu = \lambda \qquad\qquad\rightarrow\qquad\qquad \partial_\lambda(\kappa_\mu) - \partial_\lambda(\kappa_\mu T_\mu) = \partial_\lambda(\kappa_\mu)(1 - \alpha_{\mu\mu}) = 1, \tag{43}$$

where in the first equality on the right we used the chain rule and $\partial_{\kappa_\mu}(\kappa_\mu T_\mu) = \alpha_{\mu\mu}$. Thus, $\partial_\lambda(\kappa_\mu) = 1/(1 - \alpha_{\mu\mu})$, which plugged into the expression above proves the result.

A.8 ANALYSIS OF THE MINIMAL MODEL FOR COVARIANCE

Recall that our minimal model is

$$\Sigma = \sigma_c \sum_{k=1}^{d_c} u_k u_k^\top + \sigma_w \sum_{k=d_c+1}^{d} u_k u_k^\top \tag{44}$$

where the first $d_c$ eigenvectors $u_k = (v_{0,k} + v_{\perp,k})/\sqrt{2}$ represent coupling modes, and the remaining $u_k$ complete the orthonormal basis. Note that when the coupling and weak directions have the same strength ($\sigma_c = \sigma_w$), we reduce to the isotropic case. We thus are interested what changes as $\sigma_c/\sigma_w$ grows, which we study by taking the $\sigma_w \to 0$ limit.

The fixed-point equation for $\kappa$ is

$$\kappa \left( 1 - \gamma_c \frac{\sigma_c}{\sigma_c + \kappa} + (\gamma - \gamma_c) \frac{\sigma_w}{\sigma_w + \kappa} \right) = \lambda. \tag{45}$$

This has the same solutions as a cubic in $\kappa$. One option is to directly study the large $\sigma_c$ limit, in which case the equation becomes independent of $\sigma_c$. We instead take the over-parameterized ridgeless limit ($\lambda \to 0$ and $\gamma > 1$), where $\kappa$ solves

$$1 = \gamma_c \frac{\sigma_c}{\sigma_c + \kappa} + (\gamma - \gamma_c) \frac{\sigma_w}{\sigma_w + \kappa}. \tag{46}$$

Comparison to training symmetrization must be delicate. The effective ridge parameter solves

$$\kappa_{\text{inv}} \left( 1 - \gamma_c \frac{\bar{\sigma}}{\bar{\sigma} + \kappa_{\text{inv}}} + (\gamma_0 - \gamma_c) \frac{\sigma_w}{\sigma_w + \kappa_{\text{inv}}} \right) = \lambda \tag{47}$$

where $\bar{\sigma} = (\sigma_c + \sigma_w)/2$. We again take $\lambda \to 0$. When $\gamma_0 < 1$ we obtain an effective ridge of $\kappa_{\text{inv}} = 0$. This is intuitive: if $d_0 < n$ we are back in the ordinary least squares regime once we project the data down into $V_0$. If $d_0 = O(1)$ (i.e. the number of invariant features in the problem is finite) then we are in the $\gamma_0 \to 0$ regime of Theorem 1, where training symmetrization helps. However, as in the isotropic example, if $\gamma > 1$ but $0 < \gamma_0 < 1$ then $R(\hat{\beta}_{\text{inv}})$ can still grow arbitrarily large as we approach the new interpolation threshold. (This also means the order one takes the $\gamma_0 \to 0$ and $\gamma \to \infty$ limits matters; taking the latter first, for example fixing $d_0$ and $n$ and taking $d \to \infty$, still shows harmful effects for training symmetrization.)

A.8.1 PROOF OF THEOREM 2

We consider the case $\gamma_0 > 1$ and $\lambda \to 0$, in which

$$1 = \gamma_c \frac{\bar{\sigma}}{\bar{\sigma} + \kappa_{\text{inv}}} + (\gamma_0 - \gamma_c) \frac{\sigma_w}{\sigma_w + \kappa_{\text{inv}}}. \tag{48}$$

This has the same form as the equation for $\kappa$. Indeed, we can write $\kappa = \kappa(\sigma_c, \sigma_w, \gamma)$ and $\kappa_{\text{inv}} = \kappa(\bar{\sigma}, \sigma_w, \gamma_0)$ where $\kappa(s, w, g)$ solves the quadratic system

$$\kappa(s, w, g)^2 + b(s, w, g) \kappa(s, w, g) + c(s, w, g) = 0 \tag{49}$$
$$b(s, w, g) = (s + w) - \gamma_c s + (g - \gamma_c) w \tag{50}$$
$$c(s, w, g) = (1 - g) s w \tag{51}$$

For $g > \gamma_c$, one can observe $\kappa(s, w, g)$ is increasing in its arguments. Thus, $\kappa_{\text{inv}} \le \kappa$ — training symmetrization has a smaller effective regularization. In the "strong correlation" limit $w \to 0$, we have $\kappa(s, 0, g) = \max(0, s(\gamma_c - 1))$. That is, we have a new threshold, corresponding to when the model is over-parameterized with respect to the number of correlational modes.

Similarly, $\alpha = \alpha(\sigma_c, \sigma_w, \gamma)$ and $\alpha_{\text{inv,inv}} = \alpha(\bar{\sigma}, \sigma_w, \gamma_0)$ where

$$\alpha(s, w, g) = \gamma_c \left( \frac{s}{s + \kappa(s, w, g)} \right)^2 + (g - \gamma_c) \left( \frac{w}{w + \kappa(s, w, p)} \right)^2. \tag{52}$$

The second term and approaches $(1 + \kappa'(s, 0, g))^{-1}$ as $w \to 0$ when $\gamma_c < 1$, where we use $'$ to denote differentiation with respect to $w$. Evaluating the derivative, in the regime of $\gamma_c < 1$ we have $\alpha(s, w, g) \to \gamma_c + \frac{(1 - \gamma_c)^2}{g - \gamma_c}$, and thus $\alpha_{\text{inv,inv}} > \alpha$.

In the correlationally over-parameterized regime $\gamma_c > 1$, the second term vanishes as $w \to 0$, and we get $\alpha(s, w, p) \to \gamma_c^{-1}$ (using our result for $\kappa(s, 0, g)$), which we note is independent of $s$ and $g$. So, in the limit of strong correlations, $\alpha = \alpha_{\text{inv,inv}}$. We thus examine the derivatives $\alpha'(s, 0, p)$. Doing so, we again find that $\alpha_{\text{inv,inv}} > \alpha$ in a neighborhood of $w = 0$ when $\gamma_0 - (\gamma_c/2) < 1/2$, i.e. when a good portion of the invariant features are captured in correlational modes.

Since $x \mapsto x/(1-x)$ is monotonically increasing, the above behavior fully describes how $V_X(\hat{\beta})$ compares asymptotically to $V_X(\hat{\beta}_{\text{inv}})$.

Understanding the biases

$$B_X(\hat{\beta}) \simeq \frac{\kappa^2}{1-\alpha} \left( \frac{\sigma_c}{(\sigma_c + \kappa)^2} \sum_{k=1}^{d_c} (u_k^\top \beta)^2 + \frac{\sigma_w}{(\sigma_w + \kappa)^2} \sum_{k=d_c+1}^{d} (u_k^\top \beta)^2 \right) \tag{53}$$

$$B_X(\hat{\beta}_{\text{inv}}) \simeq \frac{\kappa_{\text{inv}}^2}{1-\alpha_{\text{inv,inv}}} \left( \frac{\bar{\sigma}}{(\bar{\sigma} + \kappa_{\text{inv}})^2} \sum_{k=1}^{d_c} (v_k^\top \beta)^2 + \frac{\sigma_w}{(\sigma_w + \kappa_{\text{inv}})^2} \sum_{k=d_c+1}^{d_0} (v_k^\top \beta)^2 \right) \tag{54}$$

requires handling the dependence on $\beta$. Since $\beta$ is assumed invariant, and thus $\|\beta\|^2 = \sum_{k=1}^{d_0} (v_k^\top \beta)^2$,

$$\sum_{k=d_c+1}^{d} (u_k^\top \beta)^2 = \|\beta\|^2 - \sum_{k=1}^{d_c} (u_k^\top \beta)^2 = \|\beta\|^2 \left(1 - \frac{C(\beta)}{2}\right), \tag{55}$$

where we define the coupling factor

$$C(\beta) = \sum_{k=1}^{d_c} (v_k^\top \beta)^2 / \|\beta\|^2. \tag{56}$$

The expressions for biases become

$$B_X(\hat{\beta}) \simeq \frac{\kappa^2 \|\beta\|^2}{1-\alpha} \left( \frac{\sigma_c}{(\sigma_c + \kappa)^2} \frac{C(\beta)}{2} + \frac{\sigma_w}{(\sigma_w + \kappa)^2} \left(1 - \frac{C(\beta)}{2}\right) \right) \tag{57}$$

$$B_X(\hat{\beta}_{\text{inv}}) \simeq \frac{\kappa_{\text{inv}}^2 \|\beta\|^2}{1-\alpha_{\text{inv,inv}}} \left( \frac{\bar{\sigma}}{(\bar{\sigma} + \kappa_{\text{inv}})^2} C(\beta) + \frac{\sigma_w}{(\sigma_w + \kappa_{\text{inv}})^2} (1 - C(\beta)) \right). \tag{58}$$

The $\gamma_c > 1$ regime is straightforward reusing our previous calculations, giving

$$B_X(\hat{\beta}) \simeq B_X(\hat{\beta}_{\text{inv}}) \simeq \frac{\sigma_c(\gamma_c - 1)C(\beta)\|\beta\|^2}{2\gamma_c}, \tag{59}$$

at $\sigma_w = 0$, while for $\gamma_c < 1$ both go to zero with $\sigma_w \to 0$.

### A.8.2 SIMULATION DETAILS

We now describe the simulations used to obtain Figure 3. We first describe the hyperparameter settings, and then the group symmetry.

We fix values of $n = 100$, $d = 5n$, $d_0 = 2n$, and $d_c = \lfloor n/2 \rfloor$, which puts us in the over-parameterized regime, but where the model is "correlationally under-parameterized" ($\gamma_c < 1$). In this setting, our theory predicts that as $\sigma_w \to 0$, both methods become unbiased, but data augmentation has higher variance. This is indeed what we observe in simulations. To generate Figure 3, we set $\sigma_c = 1$, so that we examine $\sigma_w$ as a fraction of $\sigma_c$. The noise is generated as $\varepsilon_i \sim \mathcal{N}(0, \sigma^2)$ with $\sigma = 0.5$, and we set nominal regularization $\lambda = 10^{-8}$ to approximate the ridgeless setting.

In order to set $d_0$, we use the permutation group $G = S_{d-d_0+1}$ acting on the first $d - d_0 + 1$ coordinates of $\mathbb{R}^d$. The invariant space $V_0$ then consists of a one-dimensional subspace of $\mathbb{R}^{d-d_0+1}$ (the one with equal entries) together with the remaining $d - (d - d_0 + 1) = d_0 - 1$ coordinates unaffected by $G$. We therefore indeed get $d_0$ total invariant directions.

For the rightmost panel of the figure, we trained a 3-layer MLP with hidden dimension 256 as the classification network.

## B    LEARNING THEORETIC CONTEXT FOR $m(p_X)$

To formalize the intuition associated with $m(p_X)$, we piece together some learning theoretic context below.

First, what would the classifier metric actually measure, if we had infinite data? In other words, if $\mathscr{F}$ is our class of neural networks, $\ell$ is a binary classification loss function, $p$ is our data distribution and $\tilde{p}$ is its randomly symmetrized version, we want to understand the following:

$$\min_{f \in \mathscr{F}} \frac{1}{2} \int_x \ell(f(x),0)dp + \frac{1}{2} \int_x \ell(f(x),1)d\tilde{p} \qquad (60)$$

Since these are continuous integrals rather than finite-sample approximations, this is an "infinite-data" setting.

In fact, this expression has been studied extensively in various settings (see Sriperumbudur et al. (2009), Lopez-Paz & Oquab (2017) and references within), which usually vary in their choice of $\mathscr{F}$ and $\ell$ (and with $\tilde{p}$ replaced by a more general second distribution $q$). For example, when $\mathscr{F}$ is the set of all measurable functions and $\ell$ is the 0/1 loss, eq. (60) is the optimal Bayes risk, classically known to be a simple affine transformation of the total variation norm.

When instead using the loss function $\ell(\alpha,0) = \alpha$ and $\ell(\alpha,1) = -\alpha$, Sriperumbudur et al. (2009) showed straightforwardly that eq. (60) is exactly the integral probability metric associated with $\mathscr{F}$:

$$\max_{f \in \mathscr{F}} \left| \int_x f(x)dp - \int_x f(x)d\tilde{p} \right| \qquad (61)$$

Integral probability metrics recover a range of familiar quantities for different choices of $\mathscr{F}$. For example, when $\mathscr{F}$ consists of all functions with Lipschitz norm bounded by 1, eq. (61) reduces to the Wasserstein metric via Kantorovich-Rubinstein duality. Similarly, when the functions in $\mathscr{F}$ are norm-bounded in some Hilbert space, the resultant integral probability metric is the kernel maximum mean discrepancy (MMD), as used by Chiu & Bloem-Reddy (2023).

For other loss functions and with unrestricted class $\mathscr{F}$, Nguyen et al. (2009) showed that there exists some convex $f$ such that eq. (60) is equal to the corresponding $f$-divergence. For example, when $\ell$ is the exponential logistic loss and $\mathscr{F}$ is all measurable functions, eq. (60) reduces to known divergences (Hellinger and $\chi^2$, respectively).

Thus, while we cannot precisely articulate eq. (60) as it depends on both the exact class of neural networks and the loss function, it reduces to reasonable and familiar quantities in special cases.

We now turn to the finite sample complexity aspect of the metric. Since we cannot truly compute eq. (60), we follow standard learning practice and use a dataset of finite samples to approximate the continuous integrals. Standard generalization bounds apply, restricting how much our finite-sample approximation to eq. (60) can diverge. In particular, let $\delta \in (0,1)$, let $x_i$ be drawn with equal probability from $p$ or $\tilde{p}$ with $y$ denoting the corresponding binary class, and let $\mathscr{R}_m(\ell \circ \mathscr{F})$ be the Rademacher complexity, i.e.

$$\mathscr{R}_m(\ell \circ \mathscr{F}) := \mathbb{E}_{(x_1,y_1)...(x_m,y_m)} \mathbb{E}_{\sigma_i} \left[ \sup_{f \in \mathscr{F}} \left( \frac{1}{m} \sum_{i=1}^{m} \sigma_i \ell(f(x_i),y_i) \right) \right] \qquad (62)$$

Then a standard generalization bound (see e.g. Appendix A of Cohen et al. (2019)) tell us that, with probability $\geq 1 - \delta$ over the randomness of the training set, the following holds $\forall f \in \mathscr{F}$:

$$\underbrace{\mathbb{E}_{(x,y)}[\ell(f(x),y)]}_{\text{optimized in } eq. (60)} \leq \underbrace{\frac{1}{|S|} \sum_{(x_1,y_1)...(x_m,y_m)} \ell(f(x_i),y_i)}_{\text{what we actually optimize}} + 2\mathscr{R}_m(\ell \circ \mathscr{F}) + \sqrt{\frac{8\ln(2/\delta)}{m}} \qquad (63)$$

With these tools in hand, we now interpret what learning theory has to say on the impact of the dataset size and model family on $m(p_X)$.

**Impact of dataset size** The dataset size with which we train our metric affects only the generalization bound from finite to infinite sample loss, not the underlying pseudometric itself. According to the generalization bound above, the the dataset size contributes inversely to an additive term in the overall generalization error, and should scale roughly with $\log(1/\delta)$.

**Impact of model family** $\mathscr{F}$ The choice of $\mathscr{F}$, i.e. the family of neural networks, affects both the generalization bound eq. (63) and the pseudometric itself eq. (60). In particular, as the architecture class becomes more expressive, the pseudometric eq. (63) grows "stronger" (i.e. can better discern subtle distributional asymmetries, with the TV norm between distributions as the limiting case), but the generalization bound eq. (63) also grows larger (i.e. the risk of overfitting is greater). In practice, of course, generalization bounds are not perfectly predictive of modern deep learning, and more nuanced bounds are possible (e.g. which take into account the complexity of the data distribution itself, rather than just the hypothesis class).

## C  Task-dependent metric

In this section, we walk through the task-dependent metric in more detail, including its derivation and a variant that used binary classification (which was ultimately less performant).

### C.1  Derivation and explanation

$m(p_X)$ does not capture whether distributional symmetry breaking contains useful information for the task at hand. If it does, for example in cases of inherent symmetry breaking as in Figure 1, then we predict that performing full-group data augmentation is a poor choice, as it discards task-relevant information contained in the exact position within the orbit (Appendix A.1). However, if the distributional symmetry breaking is superficial in the sense that it has no relation to the task of interest, then it is more subtle whether full-group data augmentation will hurt performance, as shown in Section 5. As such, we seek to refine $m(p_X)$ from Section 2 to produce a stronger signal for when data augmentation is harmful.

Intuitively, we wish to capture how much information about the task labels are captured by the non-uniformity in the data points' orbits. Let $c\colon \mathscr{X} \to G$ be a canonicalization function, such that $c(x)$ denotes where on each orbit $x$ is[8]. Since data augmentation and invariant featurizations destroy any information contained in $c(x)$, we wish to understand the dependence between orientations $c(x)$ and labels $f(x)$.

We explore two potential task-dependent metrics. A standard information-theoretic quantity for measuring dependence is the mutual information, which is the KL-divergence between the joint distribution and the product of the marginals

$$\mathrm{MI}(c(\cdot), f(\cdot)) := \mathrm{KL}\Big((c(x), f(x)) \,\Big|\Big|\, (c(x), f(x'))\Big)$$

Here, $x$ and $x'$ are independent draws from $p_x$. However, the KL divergence is inefficient to approximate with finite samples, places stringent requirements on the distributions' supports, and does not capture any notion of ease of learnability or computability. Thus, a potential task-dependent metric is replacing this divergence between product distributions with the classifier distance $d_{class}\Big((c(x), f(x)), (c(x), f(x'))\Big)$ referred to as the task-dependent detection metric below. In other words, we train a small network to classify whether pairs of group elements and labels are mismatched. Note that the task-independent and task-dependent metric are not necessarily correlated; one can be high while the other is low, and vice versa, as verified in Section 5.

One complication for this metric is that it depends on the choice of canonicalization $c(\cdot)$. We assume there is some "natural" choice of $c(\cdot)$, i.e. which is easily computable by a neural network, as we care about the implications of distributional symmetry-breaking on downstream learning tasks. To give

---

[8]Formally, all this requires is that $c(gx) = gc(x)$. Although such a map is not well-defined for objects $x$ with self-symmetry, we ignore this issue for the sake of exposition. Note that $c(\cdot)$ makes an arbitrary, but hopefully logical (barring unavoidable discontinuity (Dym et al., 2024)) choice of which $x$ to assign to the identity element of the group.

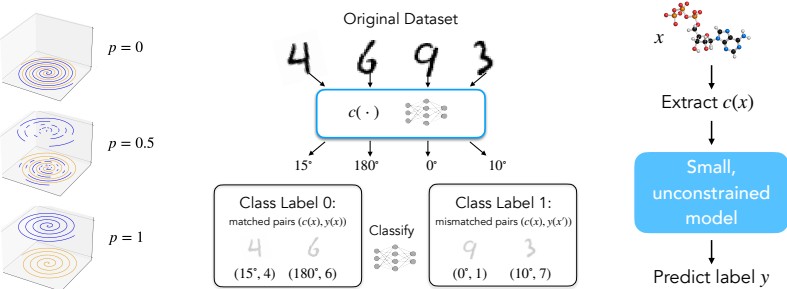

Figure 7: (Left) The swiss roll dataset (Wang et al., 2024b) provides varying levels of dependence between the canonicalization (with respect to the group of discrete vertical translation), and the task. Augmentation by vertical shifts destroys the useful canonicalization of the $p = 1$ dataset, forcing the network to learn a complicated rather than a simple decision boundary. (Middle) For the classification version of the task-dependent metric, input datapoints are assigned orientations by a small equivariant network $c(\cdot)$, and we create pairs of orientations with labels that may or may not match. A binary classifier network then tries to distinguish between matched and mismatched (orientation, labels) pairs, as a proxy for how informative the orientation is for the task. (Right) For the direct prediction version of the task-dependent metric, which we focus on in the main text, input datapoints are also assigned orientations by a small equivariant network $c(\cdot)$. These are used directly to predict the labels, and compared to the label prediction of a random group element input.

an example of a "bad" choice, imagine a $c(\cdot)$ which is discontinuous in $x$, or which is so complex to compute that even if it correlates very well with $f(\cdot)$, it would be difficult for a network to compute it as a feature. Therefore, we parametrize $c(\cdot)$ via a *small* equivariant network, which can be either trained alongside the binary classifier for $d_{class}$ or just randomly initialized.

Another choice for the task-dependent metric is the accuracy of predicting $f(x)$ directly from $c(x)$ (referred to as the direct task-dependent metric or $t(p_{X,Y})$ in our experiments). In Appendix C.2, we show that $t(p_{X,Y})$ and $d_{class}\big((c(x), f(x)), (c(x), f(x'))\big)$ are closely related. This is in turn closely related to the concepts of V-information (Xu & Raginsky, 2017) and the information bottleneck (via the canonicalization) (Tishby et al., 1999). We find empirically that the direct task-dependent metric $t(p_{X,Y})$ (results presented in 5.1), is easier to optimize and the performance is reasonable on test cases.

## C.2 THEORY

For simplicity, throughout this section we assume $c(\cdot)$ is not learned, e.g. coming from a small equivariant network with frozen weights as in the experiments.

We introduced the task-dependent metric as a measure of the dependence between the canonicalization $c(x)$ and the label $f(x)$. Instead of using the mutual information, as below,

$$\text{MI}(c(\cdot), f(\cdot)) := \text{KL}\Big((c(x), f(x)) \,\Big|\Big|\, (c(x), f(x'))\Big),$$

we used the classifier distance (from the task-independent metric) between the joint distribution and the product of the marginals. Recall the classifier distance:

$$d_{class}(p_0, p_1)) = \mathbb{E}_{b \sim \text{Bern}(\frac{1}{2})} \mathbb{E}_{x \sim p_b} \big[ \mathbb{1}\big(\text{NN}(x) = b\big) \big]$$

Specializing to our distributions, where $p_0$ is the joint and $p_1$ is the product of marginals:

$$m_1(c(\cdot), f(\cdot)) := d_{class}\big((c(x), f(x)), (c(x), f(x'))\big) = \frac{1}{2}\mathbb{E}_x \big[ \mathbb{1}\big(\text{NN}(c(x), f(x)) = 0\big) \big] +$$
$$\frac{1}{2}\mathbb{E}_{x,x'} \big[ \mathbb{1}\big(\text{NN}(c(x), f(x')) = 1\big) \big]$$

In other words, we assess a classifier (NN)'s ability to distinguish between pairs of canonicalization and label that are matched, vs mismatched. In practice, we of course train NN on a training set, and then approximate this expectation via a held-out test set.

However, another natural measure of the dependence between $c(x)$ and $f(x)$ is to assess how predictive $c(x)$ is of $f(x)$, that is: how well can a neural network predict $f(x)$ directly from $c(x)$? If there is no dependence between them, then it can do no better than random. Letting $\ell$ be a loss function, we define

$$m_2(c(\cdot), f(\cdot)) := \mathbb{E}_{x,x'} \left[ \ell(\mathrm{NN}'(c(x)), f(x')) \right] - \mathbb{E}_x \left[ \ell(\mathrm{NN}'(c(x)), f(x)) \right]$$

Here, the second term captures how well $c(x)$ can be used to predict $f(x)$, while the first term regularizes/calibrates by how well NN performs with independent inputs.

Intuitively, $m_1$ and $m_2$ are quite related to each other, and it is a straightforward exercise to make this precise. Letting $\ell$ be the 0/1 loss (i.e. 0 if $\mathrm{NN}'(c(x)) = f(x)$ and 1 otherwise), one can obtain one direction of a bound between $m_1$ and $m_2$ by using NN' to define NN. In particular, define $\mathrm{NN}(c, f) := 0$ if $\mathrm{NN}'(c) = f$, and 1 otherwise. Then,

$$\begin{aligned}
m_2(c(\cdot), f(\cdot)) &:= \mathbb{E}_{x,x'} \left[ 1 - \mathbb{1}(\mathrm{NN}'(c(x)) = f(x')) \right] - \mathbb{E}_x \left[ 1 - \mathbb{1}(\mathrm{NN}'(c(x)) = f(x)) \right] \\
&= \mathbb{E}_x \left[ \mathbb{1}(\mathrm{NN}'(c(x)) = f(x)) \right] - \mathbb{E}_{x,x'} \left[ \mathbb{1}(\mathrm{NN}'(c(x)) = f(x')) \right] \\
&= \mathbb{E}_x \left[ \mathbb{1}(\mathrm{NN}(c(x), f(x)) = 0) \right] - \mathbb{E}_{x,x'} \left[ \mathbb{1}(\mathrm{NN}(c(x), f(x')) = 0) \right] \\
&= \mathbb{E}_x \left[ \mathbb{1}(\mathrm{NN}(c(x), f(x)) = 0) \right] - \mathbb{E}_{x,x'} \left[ 1 - \mathbb{1}(\mathrm{NN}(c(x), f(x')) = 1) \right] \\
&= 2m_1(c(\cdot), f(\cdot)) - 1
\end{aligned}$$

In the other direction, we could start with NN and define $\mathrm{NN}'(c)$ to be any $f$ such that $\mathrm{NN}(c, f) = 0$.

Therefore, when optimizing independently over NN and NN', $m_2$ is at least $2m_1 - 1$, while at the same time, $m_1$ is at least $\frac{m_2+1}{2}$. The two quantities are thus related by an affine transformation — at least under a certain choice of loss (and optimization practicalities notwithstanding).

These quantities are also very related to $V$-information (Xu & Raginsky, 2017). In particular, when $\ell$ in the definition of $m_2$ is the cross-entropy loss, $m_2$ is essentially the predictive $V$-information from $c(\cdot)$ to $f(\cdot)$ (Xu & Raginsky, 2017). In subsequent experiments, when we report $m_2$ (the "task-dependent direct prediction" metric), we report only the latter term $\mathbb{E}_x \left[ \ell(\mathrm{NN}'(c(x)), f(x)) \right]$.

### C.2.1 Task-dependent metrics for finite and infinite groups

When $G$ is finite and $|G|$ is much smaller than the number of class labels, then it is clear that $c(x)$ can not be expected to predict $f(x)$ perfectly (hence the role of the first term in the expression for $m_2$). In the case of MNIST, for example, the group of 90° rotations has 4 elements, while there are 10 digits to classify (which occur with equal probabilities). Therefore, directly predicting the digit label from $c(x)$ is impossible, regardless of the dataset distribution; one can only associate one label to each of the four elements of $c(x)$. (Indeed, Fano's inequality can provide a lower bound on this probability of error.)

On the other hand, when $G$ is infinite and the dataset is essentially canonicalized according to our main, task-independent metric, an element $c(x) \in G$ can information theoretically capture exactly the identity of $x$, i.e. such that $x$ can be recovered from $c(x)$, for sufficiently expressive choice of $c$. The extent to which this generalizes will depend on the smoothness of the canonicalization, but not necessarily on how "task-correlated" it is (if NN is powerful enough to simply recover $x$ from $c(x)$ and use it to predict $f(x)$). Thus, considering the version of the task-dependent metric that aims to predict $f(x)$ directly from $c(x)$, overfitting is a concern, and may explain some of the sensitivity to architecture choice from Table 6 and Table 7.

# D    EXPERIMENTS

For all experiments, we provide further details on the training, model specifications, and results. To reproduce our experiments, we include a README.md file in the supplemental codebase, specifying the exact commands for each dataset and setting. We also summarize the task-independent metric in Table 3. As shown, every dataset experiences distributional symmetry-breaking, to varying (but all high) degrees.

| Dataset | Test Acc. |
|---|---|
| rMD17 Aspirin | 97.869 |
| rMD17 Ethanol | 79.834 |
| OC20 Surface+Adsorbate | 99.280 |
| OC20 Adsorbate | 96.529 |
| QM9 | 97.6 |
| Local QM9 | 67.6 |
| QM7b | 89.93 |
| MNIST | 87.50 |
| ModelNet40 | 92.45 |
| Local ModelNet40 $N = 10$ | 55.6 |
| Local ModelNet40 $N = 100$ | 81.5 |
| LLM Materials | 95 |

Table 3: Task-independent metric on selected datasets (omitting the toy Swiss Roll dataset and ModelNet40 reported per class in Figure 10).

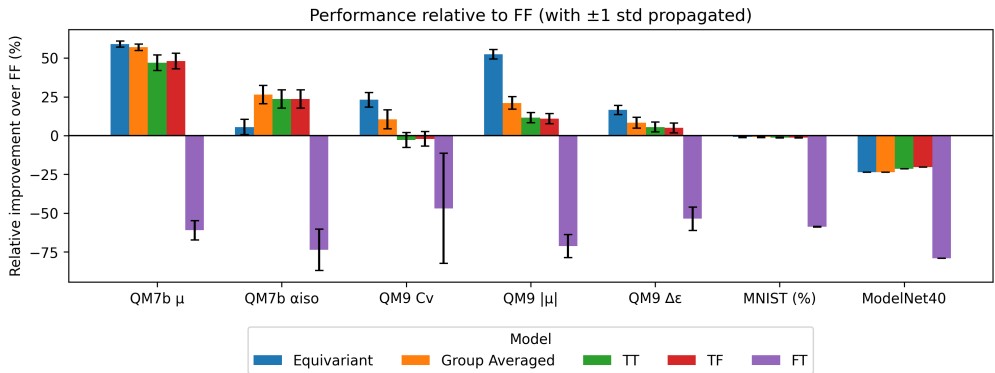

Figure 8: To complement Table 1, we show the percentage improvement relative to the FF setting for the different augmentation settings (TT, TF, FT), equivariant, and group averaged models. We see that the relative improvement is largest for the vector quantity $\vec{\mu}$, and there is no improvement for MNIST/ModelNet40.

## D.1    MODELNET40

### D.1.1    CLASSIFICATION RESULTS ON TRANSFORMER MODEL

To show that the results in the main text are not specific to the Graphormer architecture, we also run experiments with a transformer architecture. We train a transformer with the four different augmentation settings and report the test accuracies: TF=76.778%, TT=75.723%, FT=7.86%, and **FF=84.49%**. Thus, for this dataset, FF setting is better which aligns with the results in the main text.

Figure 9 shows per-class results for $m(p_X)$.

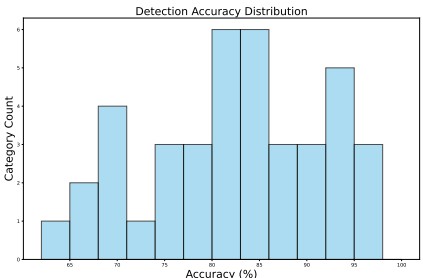

Figure 9: ModelNet40 $m(p_X)$ histogram over classes.

### D.1.2 RELATION BETWEEN THE DEGREE OF CANONICALIZATION AND ACCURACY ON FF/TF AUGMENTATION SETTINGS PER CLASS

We show the scatterplot of the FF/TF test accuracy vs. the degree of canonicalization per class in Figure 10. As above, FF indicates no augmentation at train or test time and TF indicates augmentation at train but not test time. We do not notice a trend in the relative improvement between FF and TF as a function of the task-independent metric, but is interesting to note a weak positive trend between the task-independent metric and both test accuracies. It is unclear why this is the case. We hypothesize that perhaps certain classes are defined by simple features, which would then tend to result in both the task-independent metric and the test accuracy being higher – but further exploration is needed to truly explain this phenomenon.

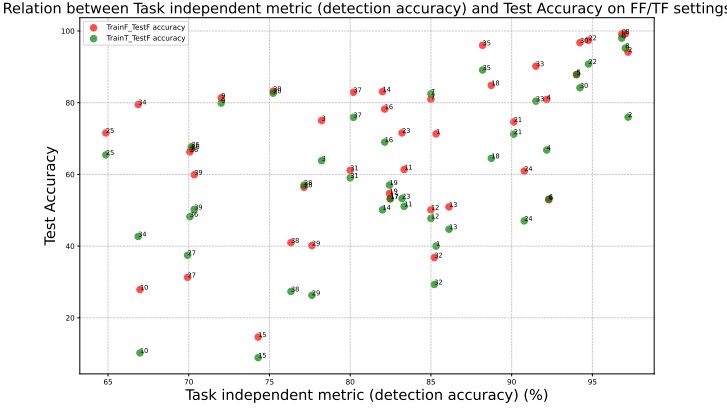

Figure 10: Relation between task independent metric and accuracy on FF/TF augmentation settings.

### D.1.3 TASK-DEPENDENT METRIC

We use a vector neuron network (Deng et al., 2021) for canonicalization. For the direct prediction task-dependent metric $t(p_{X,Y})$, we use a four-layer MLP.

### D.1.4 TRAINING CURVE

We show the training curve of ModelNet40 classification task with different augmentation settings in Figure 11. The training curve shows that the FF setting achieves the best performance all the time, while TF and TT settings achieve similar performance, and FT setting achieves the worst performance. It is interesting to contrast this result with QM9, where the best-performing setting on the unaugmented test-set is to still augment (TF). This suggests a fundamental difference between ModelNet40 and QM9.

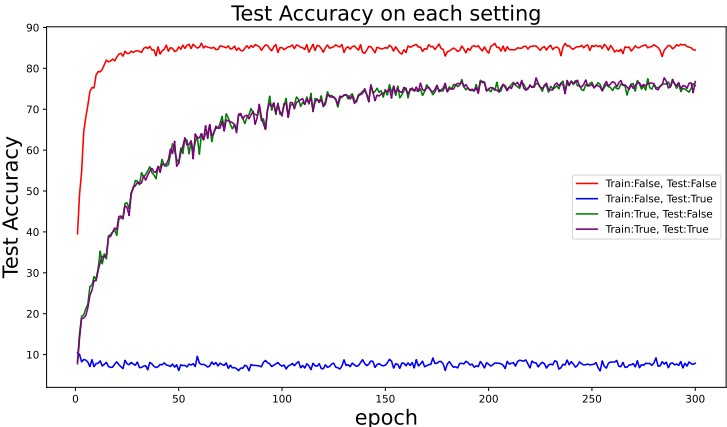

Figure 11: Test accuracy curve of ModelNet40 classification task with different augmentation settings.

### D.1.5 MODEL DETAILS

For training the task-independent metric, we use a four layer transformer architecture with four attention heads and 128 hidden dimensions as the backbone. The number of parameters in the model is 793k. For the task-dependent detection metric and task-dependent direct prediction metric, we use a three layer vector neuron network (Deng et al., 2021) as the canonicalization network, and a four layer MLP as the prediction backbone, respectively. The number of parameters for the vector neuron network is 6.1K, and for the MLP is 14.2K for direct prediction and 13.8K for detection. To ensure fair model comparison, we use the Graphormer model used in QM9/QM7b for the classification task in the main text Shi et al. (2022). In this setting, each point cloud is uniformly downsampled to 512 points. We run the base Graphormer setup with 4 blocks, 6 transformer layers, 8 heads, an FFN width of 256, and distance encodings using 32 Gaussian kernels. The regularization applied is dropout=0.1 on attention and on the final layer only—no dropout is used on inputs or intermediate activations. The number of parameters in this model is 822K.

For the equivariant counterpart, we build on e3nn(Geiger et al., 2022; Kleinhenz & Daigavane). Graph edges are defined via k-nearest neighbors with $k = 15$. Point embeddings are first lifted to the mixed representation irreps hidden = 64x0e + 16x1o. Edge attributes derive from relative offsets within a cutoff distance (max radius = 5.0) and are then expanded in spherical harmonics with irreps sh = 1x0e + 1x1o to encode angular structure. We stack three equivariant convolutional stages with gated nonlinearities and include linear self-interaction terms. The head performs global node pooling and an equivariant MLP, producing outputs with irreps out = 40x0e. The number of parameters in this model is 772K.

For the transformer baseline, the architecture we use is a six layer transformer architecture with eight attention heads and 256 hidden dimensions as the backbone. The number of parameters in this model is 4.7M.

### D.1.6 TRAINING DETAILS

The data split for all ModelNet40 experiments is 80/10/10 for training/validation/testing. The training details of classification experiments in the main text are as follows: Graphormer uses Adam with learning rate 1e−4 and batch size 16. The e3nn uses Adam at 1e-3 with batch size 16. To make the comparison fair, we implement a stochastic, group-averaged Graphormer: at each forward pass we sample $n = 3$ random rotations from $SO(3)$, run the network on each, and average the resulting predictions. For transformer models, we trained each setting for 300 epochs with batch size 128.

We train the task-independent metric for 30 epochs, the task-dependent detection metric for 1200 epochs and the direct prediction task-dependent metric for 300 epochs. All models are trained on one NVIDIA GeForce RTX 4090. The training time for the task-independent metric was about 20 minutes, for the task-dependent detection metric was about 9.5 hours, and the direct prediction

task-dependent metric was about 8.5 hours. The classification task took about two hours and a half for each setting for transformer model. For graphormer model, it will take about 8 hours for each setting. For the equivariant model, it will take about 16 hours for each setting.

## D.2 MNIST

### D.2.1 TRAINING AND MODEL DETAILS

All experiments were run on a single NVIDIA RTX A5000 with batch size 128, 50 epochs, and standard 3e-4 learning rate for the Adam optimizer, which took roughly 30 minutes each. For the data splits, we split the original training set of 60k images, and split it into 60%/20%/20% for train, validation, and test. MNIST training runs with/without augmentation used as a base network a basic 421k-parameter CNN with two convolutional layers, followed by a two-layer MLP. For the task-independent metric, the training hyperparameters and model architecture were the same, with only the final number of model outputs modified from 10 to 2. The group-averaged model used this base architecture taking the average over the $C_4$ group at each forward pass. Note this is equivalent to an equivariant model over a discrete group.

For the task-dependent detection metric, we use as the canonicalization network $c(\cdot)$ a 19k-parameter 90°-rotation equivariant classifier outputting a four-dimensional vector, corresponding to the four elements of $G$ (90°-rotations). (The classifier essentially applies a 2-layer CNN to all four rotations of the input image.) To obtain a single element of $G$ from this vector, we simply apply a softmax with low temperature (1e-3), effectively setting it to be one-hot at the index of maximum value. For the network that predicts a binary class from pairs $(c(x), f(x))$, we simply concatenate all of these inputs into a 4-layer 13.5k-parameter MLP, with 64 hidden features per layer.

### D.2.2 TASK-INDEPENDENT AND -DEPENDENT METRICS

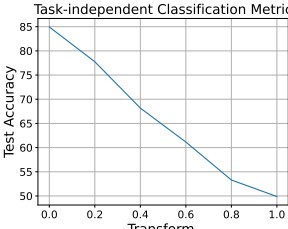

Figure 12: MNIST task-independent metric as function of proportion randomly rotated.

In Figure 12, we report $m(p_X)$ as a function of what fraction of the dataset was randomly rotated, and recover the predicted optimal accuracies.

In Figure 13 and Figure 14, we report the task-dependent metrics. They all performed somewhat poorly, possibly as a result of on how much information can be encoded in a canonicalization with respect to a group of only four elements (see Appendix C.2), or possibly because the orientation is not practically that informative for most of the digits (despite the 6/9 toy example), which is consistent with the fact that all augmentation settings (outside of "FT") worked very (and nearly equally) well.

### D.2.3 LOSS CURVES

See Figure 15. We can also analyze the test accuracy per class as in Figure 16.

### D.2.4 SELECTIVE AUGMENTATION USING THE TRAINED CLASSIFIER

We have focused on evaluating whether or not a dataset with equivariant labels has distributional symmetry breaking, and what the implications are on whether or not to employ group augmentation. However, augmenting versus not augmenting does not have to be a strict dichotomy – one can apply a restricted set of augmentations, or only on certain datapoints. Indeed, this is the approach taken by methods which learn how to augment, including e.g. Augerino (Benton et al., 2020b). We now explore how one might selectively augment using our trained classifier, interpolating between full data augmentation (when there is no distributional symmetry-breaking) and no data augmentation (when there is significant distributional symmetry-breaking).

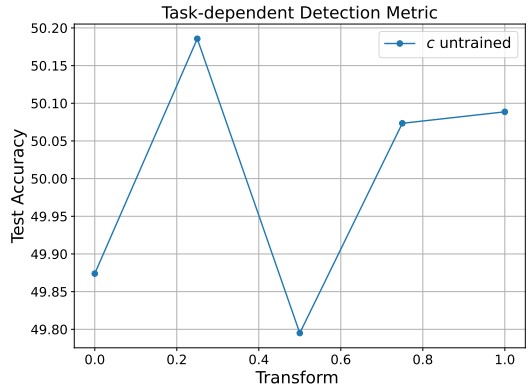

Figure 13: MNIST task-dependent detection metrics.

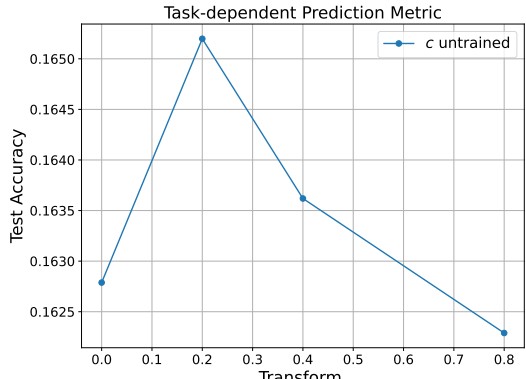

Figure 14: MNIST task-dependent prediction metrics.

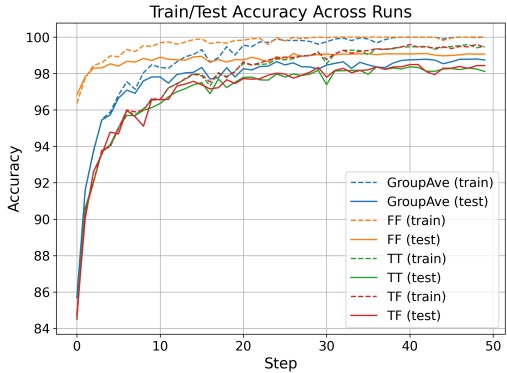

Accuracy over the course of training for the MNIST classification task, in different augmentation settings ("TF" = augmentation for training, no augmentation for testing, etc).

Figure 15: MNIST loss curves. Dashed lines indicate test losses, while solid lines indicate train losses. We omit the FT setting as the test accuracy was around 40%.

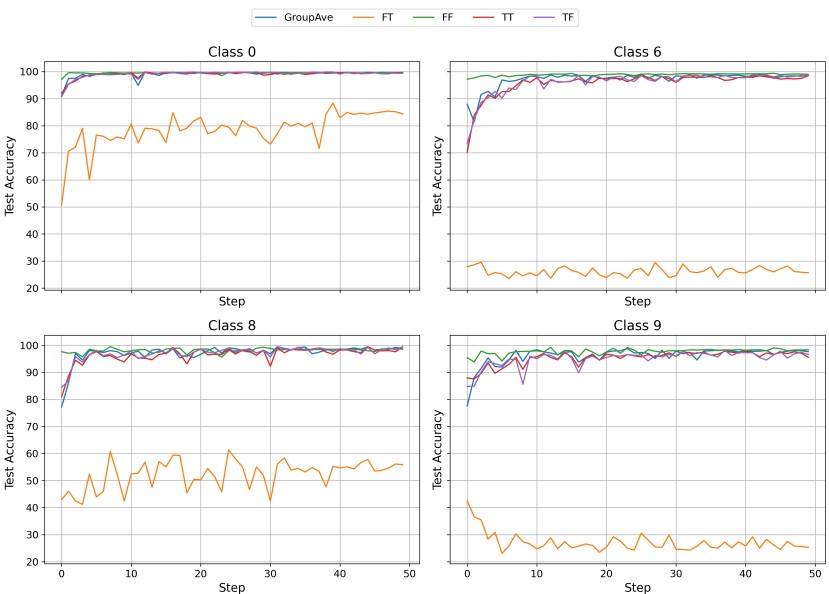

Figure 16: MNIST loss curves per selected classes. It is interesting to note the variability in the FT setting, which corresponds to whether test-time augmentation destroys useful information or not. For example, FT is much worse than the other settings for 6/9. However, for more symmetric shapes like 0/8, FT performs better.

As a first step towards this goal, we use the binary classifier (that was trained to compute $m(p_X)$) as part of a *mechanism for deciding how to augment an input datapoint*. For now, we use a simple heuristic as a proof of concept, but more advanced methods of harnessing the classifier should be possible. In particular, we threshold at 0.45 the probability the trained classifier placed on each rotation of the input being from the original (non-rotated) half of the dataset. We then only augment from the set of per-datapoint elements which pass this threshold. The threshold is a hyperparameter, but our rationale for choosing 0.45 is based on the behavior of an optimal classifier on a simplified dataset with finite orbits, as evaluated at the end of Section 2. In particular, for an input $x$, let $\mathscr{O}_x := \{gx : g \in G\}$ be the orbit of $x$, and let $S \subseteq \mathscr{O}_x$ denote the subset of the orbit of $x$ on which the natural data distribution places probability mass. For simplicity, we assume each element of $S$ is equally likely under the data distribution. The true log odds between the original dataset and the rotated dataset, for each datapoint, is $\frac{1/m}{1/|\mathscr{O}_x|} = \frac{\mathscr{O}_x}{m} := r$. Setting $\frac{p}{1-p} = r$, we obtain $p = \frac{r}{1+r}$. Thus, for inputs $x$ in the support of the data distribution, the optimal $p$ can range between $\frac{1}{2}$ – when all elements of the orbit appear under the data distribution (no distributional symmetry breaking) – and $\frac{|\mathscr{O}_x|}{|\mathscr{O}_x|+1}$ – when only one element of the orbit is natural to the data distribution. Thus, we threshold near 0.5, augmenting only on inputs that the classifier believes are likely to fall in-distribution. In an ideal world, this method should automatically interpolate between "no augmentation" and "full augmentation": if the classifier is always 50/50, then we always augment, and if the classifier is very confident that a datapoint is or is not natural, we do or do not augment accordingly.

In Figure 17, we try this out with three variants of MNIST: with the original dataset ("None"), a rotated version ("All"), and a partially rotated version with only certain digits rotated ("345"). To make the task harder and therefore benefitting from augmentation, we randomly select a subset of 1,000 images to use. The selective augmentation method described above automatically interpolates between no augmentation and full augmentation, performing near the best of the two in each case. These are promising first results, and can be investigated further in future work.

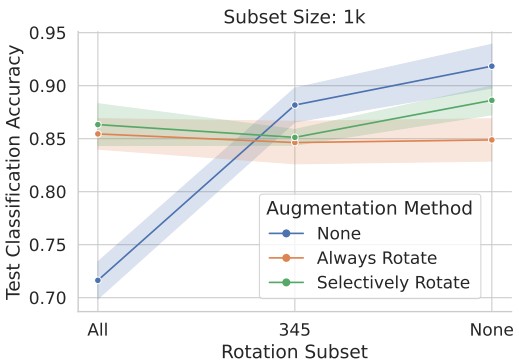

Figure 17: Classification results with a CNN using selective augmentation, averaged over 4 trials. In each trial, a new data split was used and a new detection network was trained.

## D.3 SWISS ROLL

As shown in Figure 7, the swiss roll dataset consists of two interleaved spirals (Wang et al., 2024b). The spirals have distinct $z$ values, so they are easily separable by a horizontal plane. However, there is also a more complex function fitting the data that is invariant to $z$-shifts (the group $Z_2$). Following (Wang et al., 2024b), we create a continuous family of datasets in which only a $p$-fraction of one spiral are separated vertically. This creates a spectrum of tasks, where $p = 1$ is canonicalized in a task-useful way, whereas $p = 0$ is not. We find that augmentation of a simple MLP indeed hurts performance on this task, with the effect increasing along with $p$ (**??**). This is captured by the task-dependent metrics, which increases along with $p$. However, the task-independent metric cannot capture the dataset canonicalization, as this would nearly require solving the hard spiral task itself!

To elaborate, we can think of the $p = 1$ distribution as a perfectly canonicalized dataset. The reason that our task-dependent metric does not pick up on this, and instead has only 50% accuracy, is essentially that the canonicalization was not simple – in fact, it effectively solved the prediction task (i.e. given a point $(x, y, -)$, $z$ was set based on $spiral - class(x, y)$). So, it is hard for a small network to detect on its own whether an input is canonicalized or not.

When $p = 0$ (see Figure 7), the classifier knows that any datapoint with $z = 1$ came from the transformed distribution, since the original $p = 0$ distribution always has $z = 0$. The classifier can guess the label corresponding to the original dataset when $z = 0$, and this will achieve 75% accuracy as shown. Moving from $p = 0$ to $p = 1$ simply interpolates between these two scenarios, and this is why the task-independent metric drops. In a sense, as $p$ goes from 0 to 1, we continuously transform from a task-useless canonicalization to a task-useful canonicalization, which is reflected more accurately by the task-dependent metrics (see Figure 18).

### D.3.1 TRAINING AND MODEL DETAILS

All experiments were run on a single NVIDIA RTX A5000. In our work, we modified the swiss rolls of their extrinsic equivariance setting by adding a hyperparameter $p$, to denote how much of the data on top spirals are randomly chosen to be flipped to the bottom spiral.

All experiments were trained with the Adam optimizer with learning rate 3e-4, batch size 100, and for 150 epochs. The model is a 3-layer 67k-parameter MLP for the original classification task and for the detection task; for the task-dependent metrics, a $C_2$-equivariant network with 3k parameters is used to canonicalize, composed with a 4-layer 13k-parameter MLP to perform the final prediction. The dataset consists of 1,000 examples, split randomly as 60%/20%/20% train/validation/test.

### D.3.2 RESULTS

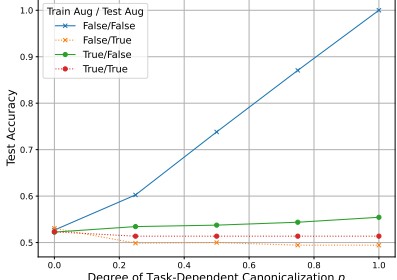 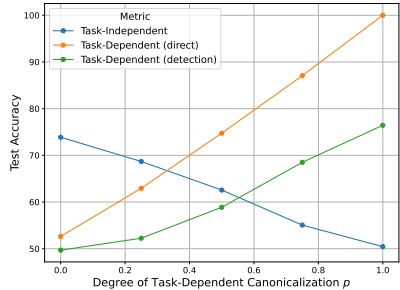

Figure 18: Swiss roll augmentation performance and metrics as functions of dataset canonicalization.

### D.3.3 LOSS CURVES

See Figure 19 and Figure 20.

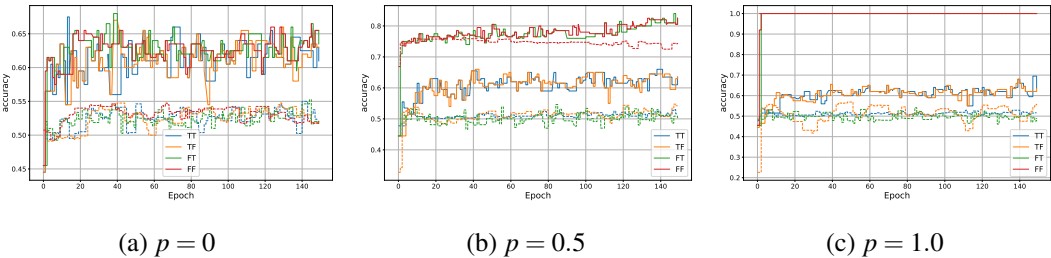

(a) $p = 0$            (b) $p = 0.5$            (c) $p = 1.0$

Figure 19: Accuracy over the course of training for the swiss roll classification task, in different augmentation settings ("TF" = augmentation for training, no augmentation for testing, etc). Dashed lines indicates test losses, while normal lines indicate train losses.

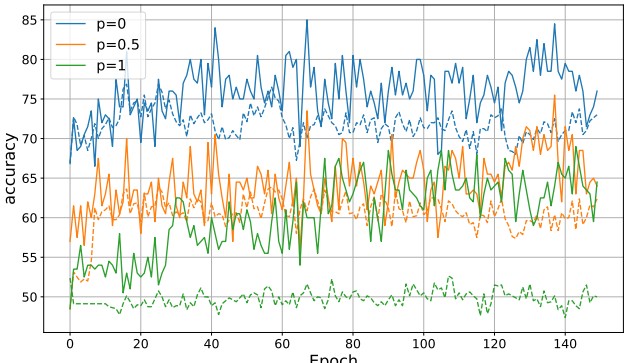

Figure 20: Accuracy over the course of training for the task-independent detection metric on the swiss roll dataset, at different levels of task-correlatedness ($p$). Dashed lines indicates test losses, while normal lines indicate train losses.

### D.4 QM9

We describe setup details, as well as further experiments on (1) the task-dependent metric, (2) the symmetry-breaking of local motifs, (3) interpreting the learned classifier.

#### D.4.1 DATASET, MODELS, AND TRAINING DETAILS

We obtain the QM9 dataset (Ramakrishnan et al., 2014) from `https://doi.org/10. 6084/m9.figshare.c.978904.v5`. The original dataset has 133885 molecules, 3054 of which are uncharacterized, as found in `https://springernature.figshare.com/ ndownloader/files/3195404`. The uncharacterized molecules are removed during preprocessing. We follow (Anderson et al., 2019) and split the dataset into training/validation/test partitions consisting of 100k, 18k, and 13k molecule examples.

For the task-independent metric, we train a generic transformer architecture with 812k parameters for 20 epochs and the Adam optimizer at learning rate 1e-5 and batch size 128. For the task-dependent metrics, we used a 3-layer, 28.5k parameter e3nn canonicalization network, using Gram-Schmidt orthogonalization to turn the `2x1e` outputs into a proper rotation matrix, and a basic 4-layer, 13.8k parameter MLP for the final task-dependent predictions. We used the Adam optimizer with learning rate 3e-4, 50 epochs, and batch size 128.

For the regression tasks, as we are studying the need for data augmentation, the principal model used should be non-equivariant/non-invariant. We note that many of the recent top performing models on QM9 are equivariant or invariant, so we use a slightly older Graphormer Shi et al. (2022); Ying et al. (2021) architecture from 2021. We include an embedding depending on the position of each atom (not solely the relative position) so the model is not invariant. Each node in the graph thus has a scalar feature (the atom number) and a 3D position associated with it. We use an embedding dimension $d_{\text{embed}} = 128$ for both the atom positions (embedded with a learnable linear layer) and for the atom types. The edges between atoms are encoded using a set of learned Gaussian radial basis functions. We adopt the following parameters of the Graphormer base architecture: 4 blocks, 8 layers, 32 attention heads, a feedforward dimension of 128, and 32 Gaussian kernels for distance encoding. Regularization uses a dropout rate of 0.1 for both attention and final layer dropout, with no input or activation dropout. We train a separate model for each property with different data augmentation settings (TT = train/test augmented, FF = none, TF = train-only, FT = test-only). For training Graphormer, we use the Adam optimizer with a learning rate of $3e-5$.

For QM9 property regression, we compare to a simple equivariant convolutional neural network architecture using e3nn(Geiger et al., 2022; Kleinhenz & Daigavane), as by equivariance, predictions should not change whether train/test are augmented or not. The network uses a learnable embedding (embedding dimension = 32) for atomic species and lifts the atom embeddings into a mixed representation `irreps hidden = 64x0e + 16x1o`. Edge features are computed through relative between atoms within a cutoff radius (`max radius = 5.0`). These features are then projected to spherical harmonics transforming as `irreps sh = 1x0e + 1x1o`, capturing the angular dependence. Radial dependence is captured via Gaussian radial basis functions applied to interatomic distances. We then use 3 layers of equivariant convolutions with gated non-linearities and linear self-interactions. The final layer pools over nodes and uses an equivariant MLP to return the final output as `irreps out = 1x0e` (a scalar for example for predicting one of the QM9 properties). The E3ConvNet model is trained with the Adam optimizer and a learning rate of $1e-4$. For an apples to apples comparison, we implement a stochastic group-averaged variant of Graphormer, in which $n = 5$ random rotations of the input are sampled from $SO(3)$ at each forward pass, and the corresponding outputs are averaged to produce the final prediction. While neither of these architectures are near state-of-the art for QM9, for our studies it suffices to use smaller models (each with approximately 800k parameters) to understand how augmenting impacts results. For both the e3nn model and the Graphormer model with augmentation settings for each property , we train each model for 150-200 epochs (depending on property) on a NVIDIA RTX A5000, which takes 2-3 hours. Minimum test MAE values and test MAE curves are reported in Table 4 and Figure 22.

#### D.4.2 TASK DEPENDENT METRIC

We describe further details of the task-dependent metric for QM9. Both the detection metric $d_{class}$ and the direct task-dependent metric $t(p_{X,Y})$ from the main text were trained for QM9 properties.

Table 4: MAE on the QM9 dataset for Graphormer under different data augmentation settings (TT = train/test augmented, FF = none, TF = train-only, FT = test-only). We include an e3nn convolutional neural network model with a similar number of parameters for comparison. The best-performing model is in bold, and the best performing-model within the augmentation settings is underlined. Results are averaged across 3 random seeds and $\pm$ 1 standard deviation is reported.

| Target | Unit | TT | FF | TF | FT | E3NN | Group Avg |
|---|---|---|---|---|---|---|---|
| $\mu$ | D | $0.243_{\pm 0.001}$ | $0.275_{\pm 0.010}$ | $0.245_{\pm 0.001}$ | $0.471_{\pm 0.011}$ | $\mathbf{0.131}_{\pm 0.007}$ | $0.217_{\pm 0.008}$ |
| $\alpha$ | $a_0^3$ | $\underline{0.480}_{\pm 0.013}$ | $0.482_{\pm 0.002}$ | $0.482_{\pm 0.009}$ | $0.891_{\pm 0.006}$ | $\mathbf{0.3733}_{\pm 0.0008}$ | $0.41_{\pm 0.06}$ |
| HOMO | eV | $\underline{0.095}_{\pm 0.001}$ | $0.108_{\pm 0.004}$ | $0.095_{\pm 0.001}$ | $0.165_{\pm 0.008}$ | $0.099_{\pm 0.002}$ | $\mathbf{0.094}_{\pm 0.003}$ |
| LUMO | eV | $\underline{0.121}_{\pm 0.003}$ | $0.128_{\pm 0.001}$ | $0.121_{\pm 0.002}$ | $0.2121_{\pm 0.0003}$ | $\mathbf{0.102}_{\pm 0.004}$ | $0.110_{\pm 0.003}$ |
| $\Delta\varepsilon$ | eV | $\underline{0.171}_{\pm 0.001}$ | $0.181_{\pm 0.006}$ | $0.1721_{\pm 0.0005}$ | $0.278_{\pm 0.010}$ | $\mathbf{0.151}_{\pm 0.002}$ | $0.166_{\pm 0.003}$ |
| $R^2$ | $a_0^2$ | $4.7_{\pm 0.2}$ | $4.46_{\pm 0.03}$ | $\underline{4.4}_{\pm 0.3}$ | $10.48_{\pm 0.05}$ | $4.6_{\pm 0.1}$ | $\mathbf{3.0}_{\pm 0.4}$ |
| ZPVE | eV | $0.0123_{\pm 0.0003}$ | $\underline{0.0110}_{\pm 0.0005}$ | $0.0121_{\pm 0.0003}$ | $0.0111_{\pm 0.0013}$ | $0.0091_{\pm 0.0013}$ | $\mathbf{0.0080}_{\pm 0.0007}$ |
| $U_0$ | eV | $6.6_{\pm 0.5}$ | $\underline{5.6}_{\pm 2.1}$ | $6.7_{\pm 0.5}$ | $6.7_{\pm 2.9}$ | $12_{\pm 5}$ | $\mathbf{4.0}_{\pm 2.3}$ |
| $U$ | eV | $\underline{6.1}_{\pm 0.9}$ | $6.3_{\pm 1.4}$ | $6.1_{\pm 0.9}$ | $8.3_{\pm 2.0}$ | $11_{\pm 5}$ | $\mathbf{5.8}_{\pm 2.0}$ |
| $H$ | eV | $\underline{6.1}_{\pm 0.9}$ | $5.2_{\pm 2.1}$ | $6.2_{\pm 0.8}$ | $8.9_{\pm 2.2}$ | $11_{\pm 6}$ | $\mathbf{5.6}_{\pm 2.7}$ |
| $G$ | eV | $7.05_{\pm 0.29}$ | $\underline{5.5}_{\pm 2.6}$ | $7.03_{\pm 0.33}$ | $8.2_{\pm 2.1}$ | $12.6_{\pm 4.4}$ | $\mathbf{5.6}_{\pm 1.2}$ |
| $c_v$ | cal/mol K | $0.147_{\pm 0.005}$ | $\underline{0.143}_{\pm 0.006}$ | $0.147_{\pm 0.003}$ | $0.21_{\pm 0.05}$ | $\mathbf{0.110}_{\pm 0.005}$ | $0.128_{\pm 0.007}$ |

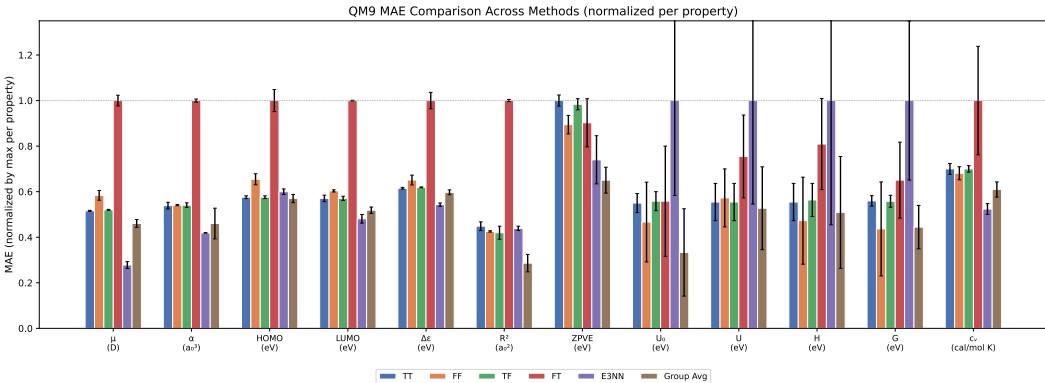

Figure 21: Bar plot visualization of Table 4, where the scale is normalized by the maximum MAE per-property across all six methods compared.

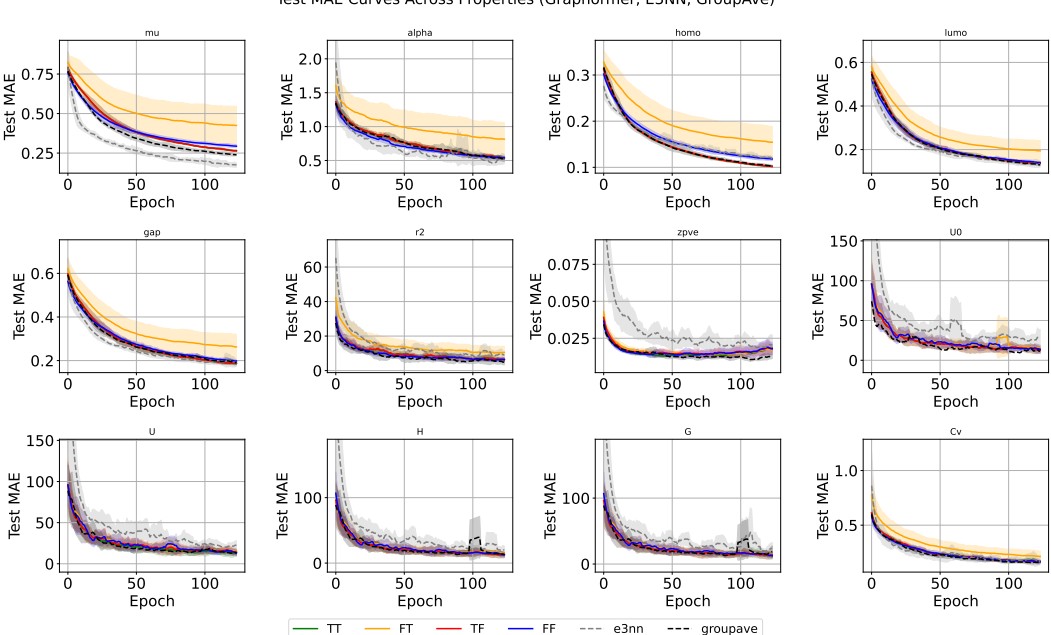

Figure 22: Test MAE per epoch for QM9 predicted properties using the Graphormer-like architecture. We show the augmentation types train/test aug=TT, train aug only=TF, test aug only=FT, no aug=FF. We also show an E3NN network with a similar number of parameters and a group-averaged graph transformer.

We observe that both task-dependent metrics have small signals (see Figure 23). This aligns with our observation that equivariant models/data augmentation perform well for QM9 as there may be a lack of significantly task relevant information contained in the QM9 canonicalization.

### D.4.3 INTERPRETABILITY OF CLASSIFIER FOR DISTRIBUTIONAL SYMMETRY BREAKING

A primary motivator for using the classifier distance for distributional asymmetry detection is because of the opportunity to explore and interpret the trained classifier. As a first step, we focus on the task-independent classifier trained on the QM9 dataset with Anderson splits as outlined in the previous section. To probe the decision boundary, we evaluate the classifier predictions on the test set with no augmented rotations (e.g. all have label 0 and are from the original dataset). It is thus easy to interpret which molecules are "hard" for the classifier to distinguish as being from the original dataset. We apply PCA to the learned embeddings (i.e., the layer immediately preceding the final output layer) and visualize them in Figure 24, revealing that the misclassified examples tend to cluster together in PCA space.

We evaluate the sigmoid of the classifiers logits on a discrete grid of 3D rotations (representing the probability that the given sample has label 0 or is from the original dataset rather than the augmented version). In order to visualize the probabilities over $SO(3)$, rotations with non-negligible probability are plotted as dots using a Mollweide projection (Murphy et al., 2022; Klee et al., 2023), with rotations orthogonal to the sphere encoded as colors and the size of the dot representing the magnitude of the probability. We explore correctly classified molecules, incorrectly classified molecules, and samples that are close to the decision boundary and show examples of each. We also investigate the stability and robustness of the classifier's decision boundary by identifying rotations of a given sample that lead to a change in its predicted label. To probe the stability of the classifier, we identify pairs of rotations that are close together yet lead to large changes in the classifier's output logits. Given two rotations represented by quaternions $p, q$, the distance between rotations is

$$\theta = 2 \arccos |<p, q>| \tag{64}$$

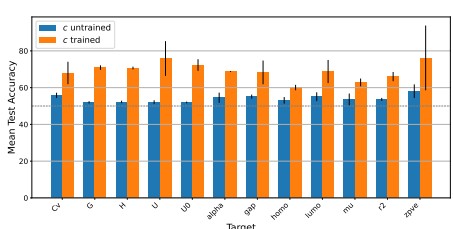
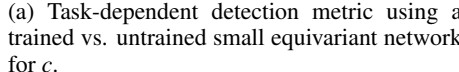

(a) Task-dependent detection metric using a trained vs. untrained small equivariant network for $c$.

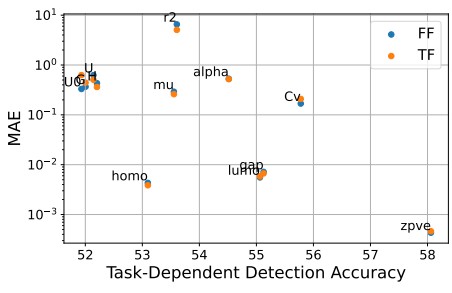

(b) Normalized MAE for QM9 vs the task-dependent detection metric, with $c$ untrained.

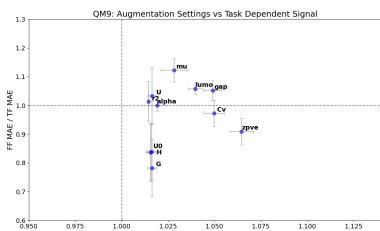

(c) MAE FF/TF improvement vs QM9 direct task-dependent metric.

Figure 23: QM9 task-dependent detection metric figures with MAE table and plots.

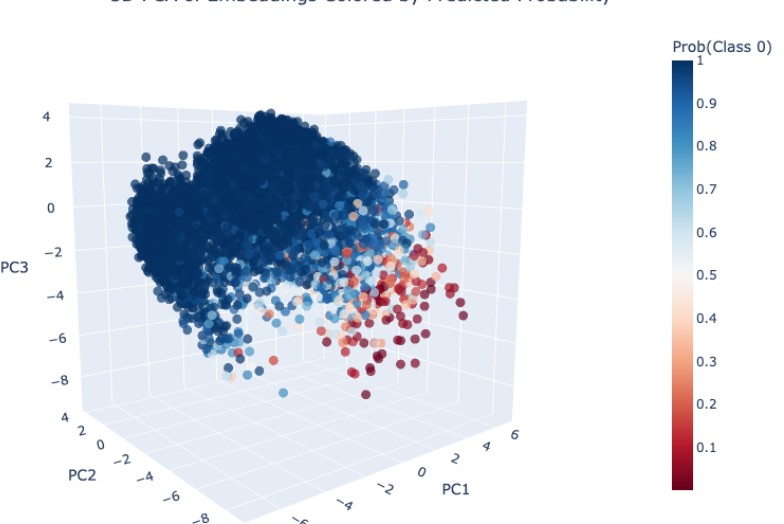

Figure 24: PCA of learned embeddings for the task-independent classifer for QM9 applied to the test dataset with no rotations. The misclassified examples thus are shown in red.

**Example Correctly Classified with High Probability.**    We select a sample classified correctly with high probability as being from the original dataset and investigate the classifier outputs per rotation angle, as shown in Figure 25.

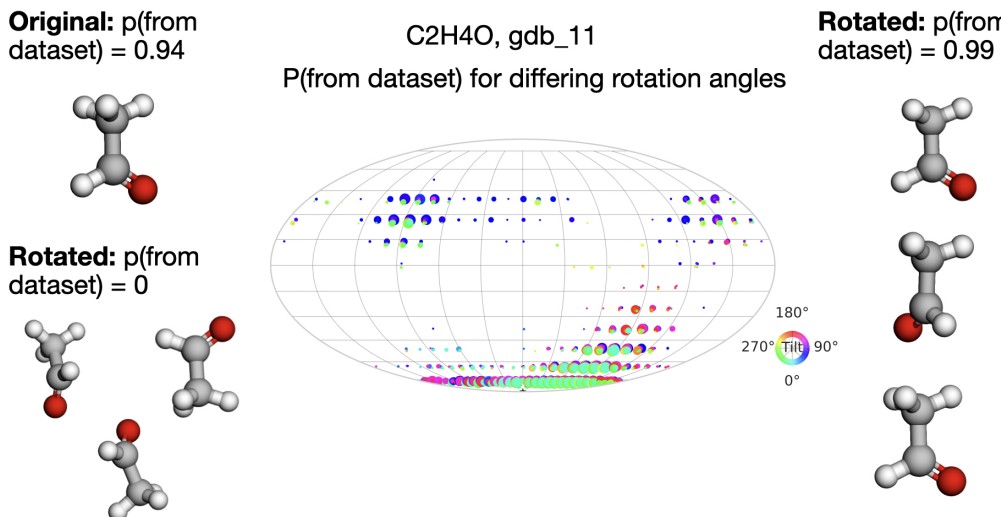

Probabilities per rotation angle for a sample that is classified correctly. The colors correspond to rotations orthogonal to the sphere and the size of the dots corresponds to the probability value. The original molecule in the dataset is shown on the upper left. The lower left shows rotations that cause `prob(original)` to be zero. The right shows rotations that cause `prob(original)` to be high.

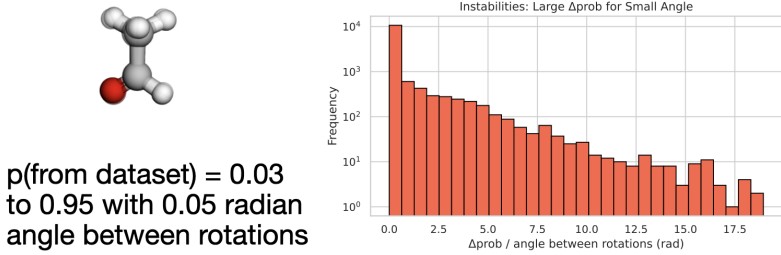

Instability in the decision boundary (left): two nearby rotations cause a large change in predicted probability. Histogram (right): certain examples exhibit such instabilities more frequently.

Figure 25: Visualizations of classifier outputs for an example classified incorrectly.

**Example Incorrectly Classified.**    We select a sample classified incorrectly with low probability as being from the original dataset and investigate the classifier outputs per rotation angle, as shown in Figure 26.

**Example Close to the Decision Boundary.**    We select also select an example from the non-augmented test set that the model assigns a 50% probability of belonging to the true dataset (correctly classified but close to the decision boundary).

It is interesting to note that for each example, we find instabilities in the decision boundary (rotations that are very close together but correspond to very large changes in the classifier output). This demonstrates that our method could perhaps be used to probe the instabilities of a given canonicalization – we know that each canonicalization has such instabilities (Dym et al., 2024), although we

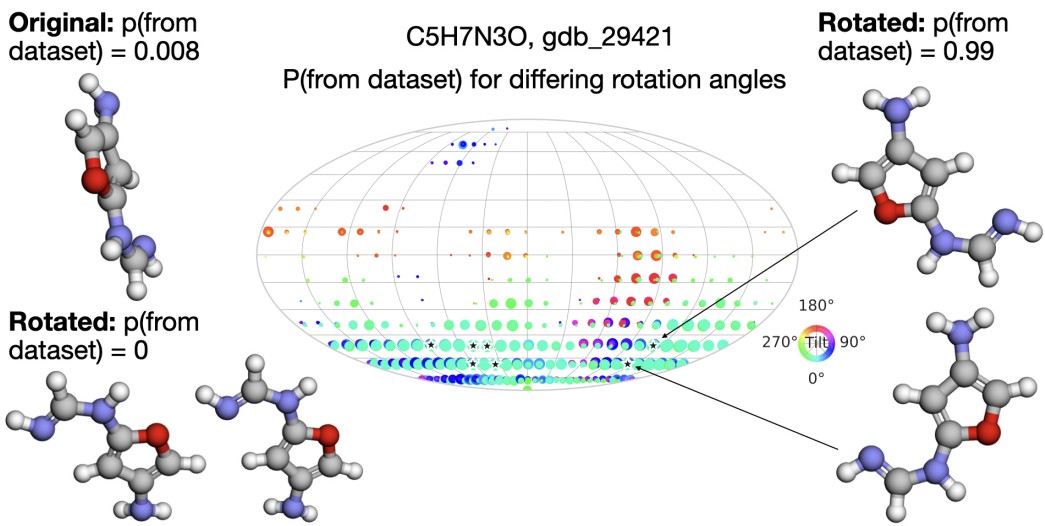

Probabilities per rotation angle for a sample that is classified incorrectly. The colors correspond to rotations orthogonal to the sphere and the size of the dots corresponds to the probability value. The original molecule in the dataset is shown on the upper left. The lower left shows rotations that cause `prob(original)` to be zero. The right shows rotations that cause `prob(original)` to be high, with arrows pointing to corresponding (starred) points on the Mollweide projection plot.

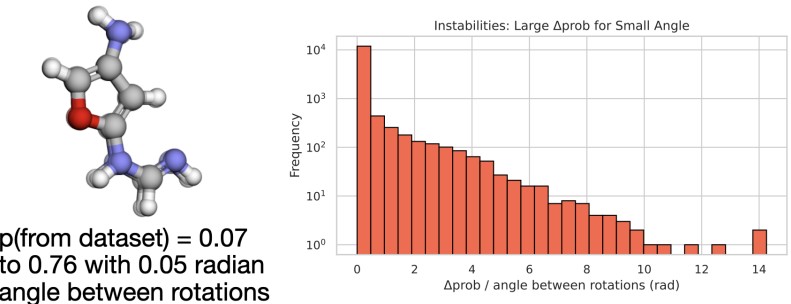

Instability in the decision boundary (left): two nearby rotations cause a large change in predicted probability. Histogram (right): certain examples exhibit such instabilities more frequently.

Figure 26: Visualizations of classifier outputs for an example classified incorrectly.

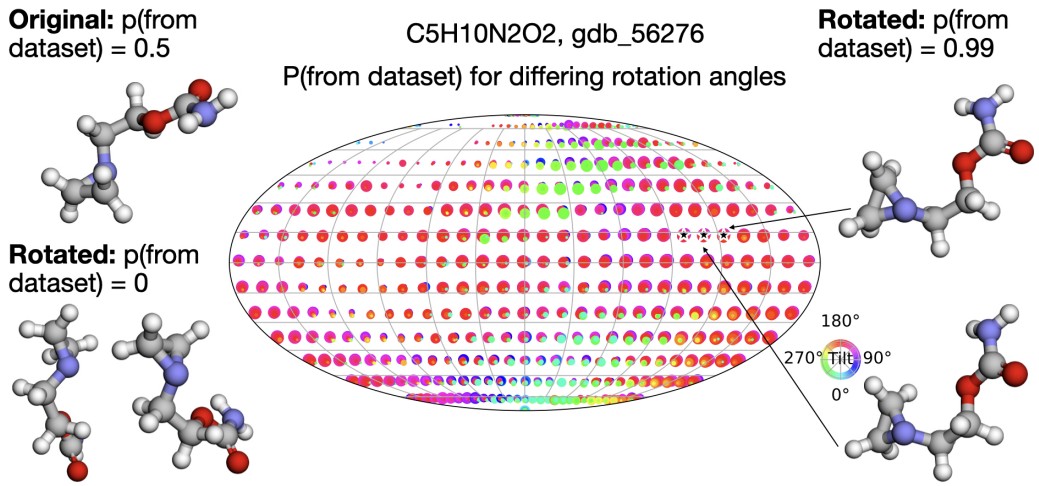

Probabilities per rotation angle for a sample on the decision boundary. The colors correspond to rotations orthogonal to the sphere and the size of the dots corresponds to the probability value. The original molecule in the dataset is shown on the upper left. The lower left shows rotations that cause `prob(original)` to be zero. The right shows rotations that cause `prob(original)` to be high, with arrows pointing to corresponding (starred) points on the Mollweide projection plot.

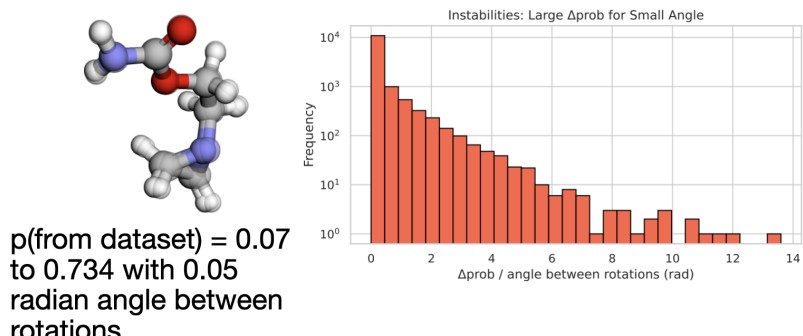

Instability in the decision boundary (left): two nearby rotations cause a large change in predicted probability. Histogram (right): certain examples exhibit such instabilities more frequently.

Figure 27: Visualizations of classifier outputs for an example close to the decision boundary.

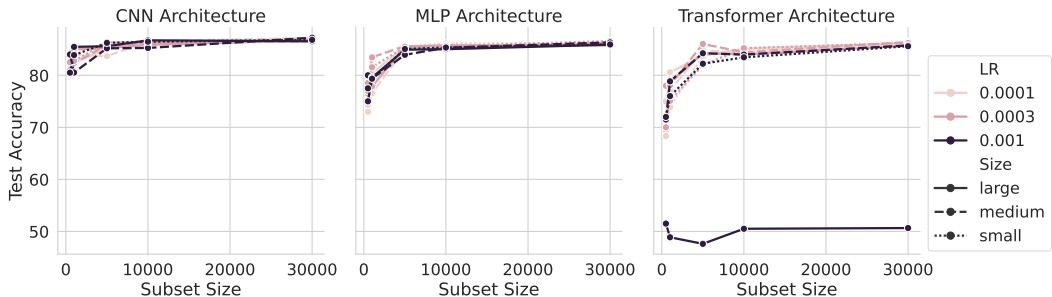

Figure 28: Ablations of $m(p_X)$ on MNIST with respect to architecture type, architecture size, learning rate, and dataset size. Curves are averaged over two random seeds.

cannot guarantee that the instabilities in the model's predictions arise for this reason (and not e.g. due to a failure to learn). Nonetheless, the models probed achieved very high test accuracy, lending confidence that the identified example instabilities are genuine.

### D.5 ABLATIONS

As shown in Table 5, the task-independent metric on QM9 is robust to the choice of architecture.

Table 5: The classifier network is a transformer architecture, in which we vary the depth, number of heads, and hidden dimension. The task-independent metric is robust with respect to the architecture size.

| Setting | Depth | Heads | Hidden Dimension | Test Accuracy | Parameters |
|---|---|---|---|---|---|
| tiny | 2 | 2 | 64 | 98.3 | 110,000 |
| small_hidden | 4 | 4 | 64 | 98.6 | 210,000 |
| large_hidden | 4 | 4 | 256 | 98.4 | 3.2e6 |
| many_heads | 4 | 8 | 128 | 98.5 | 810,000 |
| shallow | 2 | 4 | 128 | 98.3 | 420,000 |
| micro | 2 | 2 | 32 | 98.0 | 27,000 |
| few_heads | 4 | 2 | 128 | 98.7 | 810,000 |
| deep | 8 | 4 | 128 | 98.7 | 1.6e6 |

#### D.5.1 HYPERPARAMETER ABLATIONS

We further ablate sample complexity, together with architecture and learning rate, for MNIST in Figure 28 and QM9 in Figure 29. To vary sample complexity (on the x-axes), we randomly select a subset of the original dataset of the specified size. As shown, there is little sensitivity to these factors, except when using a learning rate that is too large for the transformer architecture (but this is easily detectable from the loss curves). The greatest variation is visible at the lowest-sample regimes, but the classifier performance in all cases is still significantly better than random, which would be 50% accuracy (and recall that the optimal accuracy for MNIST is 85.7%, since the group of 90° rotations only has four elements). Overall, the metric is remarkably sample-efficient on these two datasets. This matches our intuition that the distributional symmetry-breaking is quite pronounced, as shown by the high accuracies, $m(p_X)$, for each. (In the language of Appendix B, we can think of the hypothesis class/network family as being relatively simple; thus, the networks can be trained with even small fractions of each dataset.)

#### D.5.2 DIRECT TASK-DEPENDENT METRIC ABLATIONS

As motivation for using the direct task-dependent metric $t(p_{X,Y})$ instead of the task-dependent detection metric $d_{class}$, we note that $d_{class}$ is overly sensitive to the choice of canonicalization and classifier network architectures (see Table 6 and Table 7). In contrast, $t(p_{X,Y})$ is relatively robust

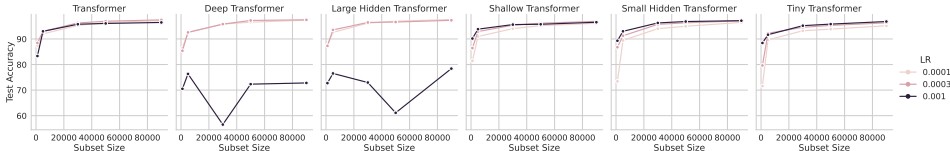

Figure 29: Ablations of $m(p_X)$ on QM9 with respect to architecture size, learning rate, and dataset size. Curves are averaged over two random seeds.

to architecture choice of both the canonicalization network and the classifier network: as shown in Figure 30 and Figure 31, the value stays close to 1 and does not predictably increase or decrease as a function of network hyperparameters.

Table 6: Ablations on the canonicalization network architecture for the task-dependent *detection* metric (not $t(p_{X,Y})$), averaged over 2 independent runs on QM9. As shown, there is indeed variation in the task-dependent metric with the architecture of the canonicalization network. Although we expect some variation—since the task-dependent metric is supposed to pick up on a "simple" canonicalization, we did intend to restrict the maximum size of the network—the ablations below surprisingly demonstrate that the highest accuracies were achieved by the smallest networks. We note that the loss curves were fairly unstable, possibly pointing to optimization difficulties with tensor product equivariant networks that might be alleviated for smaller/shallower networks. For a fair comparison in practice, one should fix an architecture size, and only compare accuracies computed with the same architecture.

| Layers | Irrep Dimension | Test Accuracy | Parameters | acc_mean | acc_std | param_float |
|---|---|---|---|---|---|---|
| 4 | 16x0e + 4x1e | $72 \pm 3$ | 38,000 | 72 | 3.0 | 38,000 |
| 2 | 16x0e + 4x1e | $69 \pm 0.9$ | 19,000 | 69 | 0.9 | 19,000 |
| 2 | 32x0e + 8x1e | $72 \pm 2$ | 74,000 | 72 | 2.0 | 74,000 |
| 4 | 32x0e + 8x1e | $76 \pm 2$ | 150,000 | 76 | 2.0 | 150,000 |
| 3 | 32x0e + 8x1e | $74 \pm 1$ | 110,000 | 74 | 1.0 | 110,000 |
| 3 | 16x0e + 4x1e | $86 \pm 2$ | 29,000 | 86 | 2.0 | 29,000 |
| 2 | 8x0e + 2x1e | $89 \pm 0.1$ | 5,400 | 89 | 0.1 | 5,400 |
| 4 | 8x0e + 2x1e | $89 \pm 0.4$ | 10,000 | 89 | 0.4 | 10,000 |
| 3 | 8x0e + 2x1e | $88 \pm 2$ | 7,900 | 88 | 2.0 | 7,900 |

Table 7: Ablations on the classifier network architecture for task-dependent *detection* metric (not $t(p_{X,Y})$). The classifier network is an MLP, for which we vary the number of layers and the hidden dimension.

| Depth | Hidden Dimension | Test Accuracy | Parameters |
|---|---|---|---|
| 4 | 32 | 68.2 | 3,800 |
| 4 | 128 | 88.1 | 52,000 |
| 2 | 64 | 88.2 | 5,200 |
| 4 | 64 | 70.0 | 14,000 |
| 8 | 64 | 87.6 | 31,000 |
| 2 | 32 | 74.9 | 1,600 |

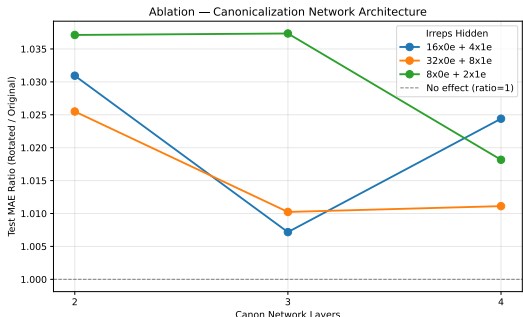

Figure 30: Ablating canonicalization network architecture when computing $t(p_{X,Y})$ for the $U_0$ value of $QM9$. The variation in ratio between losses is low.

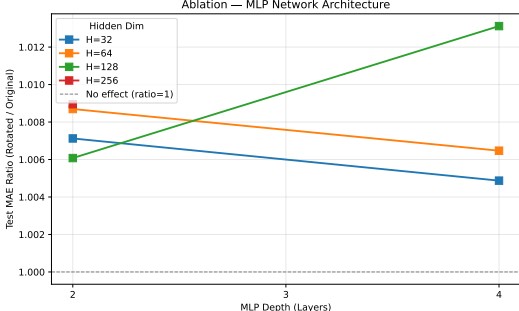

Figure 31: Ablating classifier (MLP) network architecture when computing $t(p_{X,Y})$ for the $U_0$ value of $QM9$. The variation in ratio between losses remains low.

### D.6 Local QM9

We have shown that the QM9 dataset is highly canonicalized, yet data augmentation and equivariant methods both perform well even on the original, canonicalized test set. This behavior is distinct from ModelNet40, where train-time augmentation impedes performance on the (also canonicalized) test set. This poses a question: why are equivariant methods so helpful for QM9, even though it's already canonicalized? We explore the hypothesis that locality is an important factor impacting performance (which would be captured by equivariant methods, but not canonicalization). In particular, we seek to understand whether local graph motifs in QM9 are *less canonicalized* — i.e. more likely to appear in a variety of rotations — than the full molecules. If true, then augmentation and equivariant architectures might both benefit from exposing the network to full group orbits of local motifs.

Concretely, our question is: do the local motifs present in QM9 graphs experience distributional symmetry breaking? To address this, we create a new dataset from the original QM9 dataset by randomly selecting three nodes from each molecule, and creating a new molecule fragment out of only each node and its neighborhood (as determined by its edges/bonds). As shown in Figure 32, this often includes repeated neighborhoods. This creates a dataset of size 392k. We first simply apply the task-independent detection metric, asking a network to distinguish between rotated and unrotated motifs. (All experimental and model details are preserved from the ordinary QM9 setting). As shown in Figure 5, the local dataset has lower accuracy than the QM9 dataset. However, this does not provide a maximally fine-grained distinction between different kinds of distributional symmetry breaking. For example, suppose a molecule always appears in one of two possible canonicalizations. With an infinite group like $SO(3)$, this detection problem is still likely to be perfectly solvable, as two orientations are still only a measure zero set of $SO(3)$. Yet, this case is distinct from the perfectly canonicalized case.

To assess whether a dataset is truly canonicalized, we train a network to predict $g$ from $gx$, where $g$ is drawn randomly from the Haar measure. **Solving this task to high accuracy is only possible when the distribution is truly canonicalized (only one element per orbit appears).** We use the same transformer architecture to output 9 values as the entries of a rotation matrix, and trained it according to the MSE. (Neither backpropagating through a Gram-Schmidt procedure to make it a proper rotation matrix, nor training an equivariant architecture, nor backpropagating through the angle of rotation error instead of the MSE, were as effective as this simple method, which also circumvents the symmetry-breaking that would be required to output a group element on symmetric inputs (Smidt et al., 2021).) As shown in Table 8, there is a discrepancy between the best test accuracy achieved on the original QM9 dataset, and that achieved on the local neighborhood version. **Therefore, it appears that the original QM9 dataset is more canonicalized, whereas the local motifs presented in the QM9 dataset can appear in several orientations (although still far from uniform over $SO(3)$).** This provides some evidence for the hypothesis that methods which involve equivariance to local motifs – including data augmentation and equivariance, but not canonicalization – may be providing an additional advantage on QM9.

Consistent with Figure 33, it also took much longer to train these models (500 epochs took 6 hours on the original QM9 dataset, and nearly 15 hours on the local QM9 dataset, likely due to slower dataloading), which contrasts with the efficient convergence (around 30 minutes) of our main task-independent detection metric.

|  | QM9 | Local QM9 |
|---|---|---|
| Test Error (degrees) | 13.5 | 53.7 |

Table 8: Predicting $g$ from $gx$.

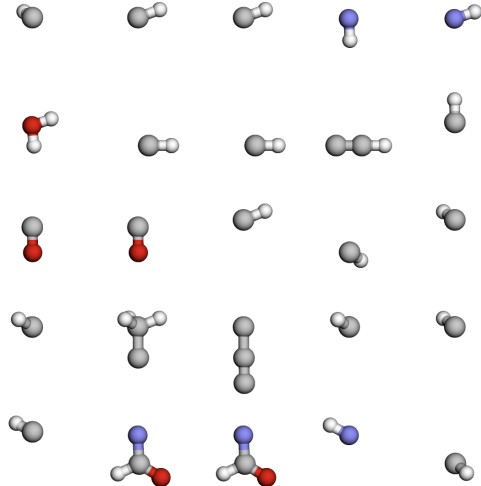

Figure 32: Local QM9 dataset visualization.

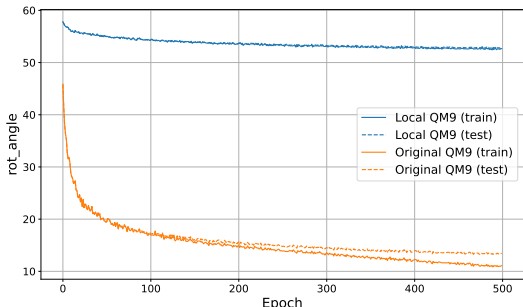

Figure 33: QM9 and local QM9 dataset training curves for predicting *g* from *gx*.

## D.7 QM7B

### D.7.1 DATASET DETAILS

In (Yang et al., 2019), density functional theory (DFT) and linear-response coupled-cluster theory including single and double excitations (LR-CCSD) is used to compute vector and tensorial molecular response properties for the 7,211 molecules in the QM7b database (Blum & Reymond, 2009; Montavon et al., 2013). LR-CCSD is generally more computationally expensive (scaling $O(N_e^6)$ with the number of electrons $N_e$) yet more accurate than DFT, which scales as $O(N_e^3)$. As the QM7b dataset is composed of small molecules, computing material response properties with LR-CCSD is feasible. Quantities computed include the dipole vector $\vec{\mu}$, polarizability $\alpha$, and quadrupole moment $Q$. The molecular dipole polarizability $\alpha$ describes the tendency of a molecule to form an induced dipole moment in the presence of an external electric field (Yang et al., 2019). It can be computed by taking the second derivative of the electronic energy $U$ with respect to an applied electric field $\vec{E}$:

$$\alpha_{ij} = \frac{\partial^2 U}{\partial E_i \partial E_j}. \tag{65}$$

Scalar polarizability response quantities are the isotropic polarizability $\alpha_{\text{iso}}$ and the anistropic polarizability $\alpha_{\text{aniso}}$

$$\alpha_{\text{iso}} = \frac{1}{3}(\alpha_{\text{xx}} + \alpha_{\text{yy}} + \alpha_{\text{zz}}) \tag{66}$$

$$\alpha_{\text{aniso}} = \frac{1}{\sqrt{2}}\left[ (\alpha_{\text{xx}} - \alpha_{\text{yy}})^2 + (\alpha_{\text{yy}} - \alpha_{\text{zz}})^2 \right. \\ \left. + (\alpha_{\text{zz}} - \alpha_{\text{xx}})^2 + 6(\alpha_{\text{xy}}^2 + \alpha_{\text{xz}}^2 + \alpha_{\text{yz}}^2) \right]^{1/2} \tag{67}$$

The dipole moment is the first derivative:

$$\vec{\mu} = \frac{\partial U}{\partial \vec{E}}. \tag{68}$$

The quadrupole moment $Q$ is a rank-2 tensor that characterizes the second-order spatial distribution of the molecular charge density, capturing deviations from spherical symmetry and providing information about the shape and anisotropy of the electron cloud beyond the dipole approximation:

$$Q_{ij} = \sum_{\alpha} q_{\alpha} \left( 3r_{\alpha i} r_{\alpha j} - \delta_{ij} r_{\alpha}^2 \right). \tag{69}$$

$q_{\alpha}$ is the charge of particle $\alpha$, $\hat{r}_{\alpha i}$ is its $i$-th coordinate operator relative to the molecular center of mass, and $\delta_{ij}$ is the Kronecker delta.

Data can be downloaded from `https://archive.materialscloud.org/record/ 2019.0002/v3`. For our studies, we use the most accurate level of theory available in the dataset—linear-response coupled cluster with single and double excitations (LR-CCSD)—in combination with the d-aug-cc-pVDZ (daDZ) basis set, to reduce basis set incompleteness error (Yang et al., 2019) (specifically, the file CCSD_daDZ.tar.gz available at the link above). The data is then converted from XYZ format into a `torch_geometric` dataset.

### D.7.2 MODEL AND TRAINING DETAILS

For the task-independent metric, we use the same generic transformer used for QM9 to find that the dataset is canonicalized. We train for 100 epochs with a batch size of 128 and a learning rate of 1e-5 with the Adam optimizer. From (Yang et al., 2019), this is expected as the molecules were reordered using a kernel-based similarity measure from (Bartók et al., 2013). For the task-dependent metric, we use $c$ untrained, as we found that using $c$ trained allowed the network to learn the dipole vector itself. We use the same parameters as for the task-dependent metric for QM9.

For the regression tasks, we use the same graph transformer architecture as described in Appendix D.4 and compare to the same $E(3)$-equivariant neural network (now with a vector or $\ell = 2$ output rather than a scalar as in QM9)/group-averaged network with 5 sampled rotations. We train each for 500 epochs with a batch size of 128 on a single NVIDIA RTX A5000. The E3ConvNet model is trained with a learning rate of 1e-4 and the Graphormer model is trained with a learning rate of 3e-5, both with the Adam optimizer. As anticipated, the $E(3)$-equivariant model achieves better performance than the Graphormer in predicting dipole moments, owing to its physically consistent treatment of vector-valued (non-scalar) quantities.

### D.7.3 TASK-RELEVANT CANONICALIZATION

To investigate the impacts of a task-relevant canonicalization, we run further experiments on the QM7b dataset. Consider aligning molecules such that their dipole moments coincide with the z-axis, filtering for molecules with non-zero dipole moments. This canonicalization clearly makes it easier for a non-equivariant model to solve the task, whereas an equivariant model will be unable to use this information. We test different data augmentation settings to illustrate the impacts of the task-useful canonicalization (train/test aug=TT, train aug only=TF, test aug only=FT, no aug=FF). Values reported in the table below are the MAE across the dipole vector components in atomic units (a.u). For the task-useful canonicalization, the FF setting (train/test fully canonicalized by dipole, no augmentation) outperforms the equivariant model (shown in bold). In the original dataset without canonicalizing based on the dipole, FF does not outperform the equivariant model. These results provide an interesting avenue for future work/for testing the task-dependent metric.

Table 9: Dipole prediction MAE for different datasets and models. Lower values are better. For the task-relevant dipole canonicalization, only FF is reported; other augmentation settings are unchanged. Equivariant model and group averaging is also unaffected by canonicalization. Note the FF setting with task-relevant canonicalization outperforms equivariant methods.

| Dataset | FF (Dipole Canon) | e3nn | Group Avg 5 rot |
|---|---|---|---|
| Dipole Canon | $\mathbf{0.037 \pm 0.003}$ | $0.041 \pm 0.002$ | $0.043 \pm 0.002$ |
| Original Dataset | $0.100 \pm 0.001$ | $0.041 \pm 0.002$ | $0.043 \pm 0.002$ |

### D.7.4 LOSS CURVES

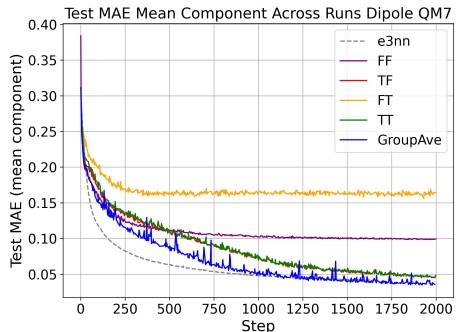 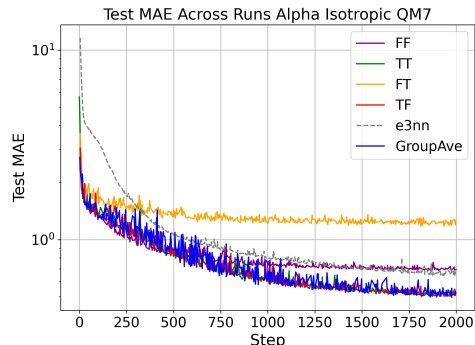

Figure 34: (left) Test MAE per epoch for predicting the dipole moment QM7b (with the e3nn model/group averaged shown for reference). (right) Test MAE per epoch for predicting the isotropic component of the $\alpha$ tensor.

### D.8 RMD17

We use the revised MD17 dataset Christensen & von Lilienfeld (2020), as the original MD17 dataset has a high level of numerical noise (Chmiela et al., 2017). The revised MD17 dataset was calculated with a more accurate DFT functional/convergence criteria than the original MD17. For this dataset, we currently have explored the task-independent metric. We use the provided five train/test splits from `https://figshare.com/articles/dataset/Revised_MD17_dataset_rMD17_/12672038` and train a separate model for each molecule. Note it is not recommended to train a model on more than 1,000 samples from rMD17 Christensen & von Lilienfeld (2020), even though the dataset has 100,000 conformers for each trajectory. We train a generic transformer with 812k parameters for 50 epochs on the train/test splits provided with the Adam optimizer at learning rate `1e-5` and batch size `128`.

As seen in Figure 35, all molecules are canonicalized. However, the task-independent metric of test accuracy yields significantly different values per molecules. For example, aspirin has a test accuracy of 97.869%, but ethanol yields 79.834 %. In Figure 35, ethanol and malohaldehyde have a noticeably lower degree of canonicalization.

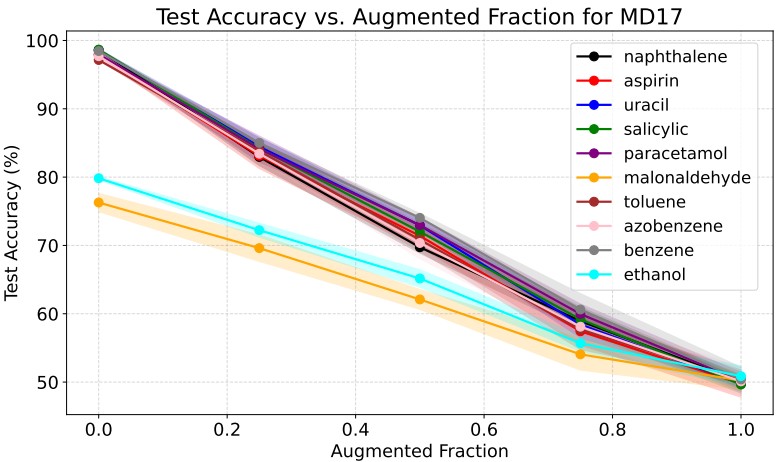

Figure 35: Test accuracy vs. augmented fraction for all molecules in rMD17. Note the difference between the 8 more canonicalized molecules and ethanol/malonaldehyde.

### D.9 OPEN CATALYST PROJECT 2020 (OC20)

For our study, we use the 200K subset from the structure to energy and forces (S2EF) task, available at `https://fair-chem.github.io/core/datasets/oc20.html#structure-to-energy-and-forces-s2ef-task`. For this dataset, we have explored the task-dependent metric. It would be interesting in the future to explore other tasks (e.g. Initial Structure to Relaxed Structure) and larger dataset sizes, as the OC20 dataset training set alone has 20 million structures. We use the preprocessing pipeline provided at `https://fair-chem.github.io/core/datasets/oc20.html`. Positions for each catalyst+adsorbate are tagged with 0: catalyst surface, 1: catalyst sub-surface, and 2: adsorbate. The unit cell for the catalyst is repeated twice in the $x$ direction, twice in the $y$ direction, and once in the $z$ direction, leading to the slab's alignment with the $xy$ plane. This alignment most likely trivially causes our metric to report distributional symmetry breaking. We also expect the adsorbate alone to be slightly less canonicalized than the combined catalyst surface–adsorbate system (as the adsorbate alone is not a periodically repeating slab). This is supported by the test accuracy, which is 96.529% for the adsorbate alone compared to 99.280% for the surface plus adsorbate system. It would thus be interesting in future work to consider how to treat periodic crystalline systems.

## D.10 $p$-VALUES FOR $m(p_X)$

Following (Chiu & Bloem-Reddy, 2023), our computation of $m(p_X)$ can be extended to yield $p$-values via Algorithm 2. To construct a null distribution, we repeatedly sample train/test splits from the original dataset, apply a random rotation to all points in each split, and record the resulting pairwise distances. We then compute a test statistic by sampling a second collection of train/test splits, applying random rotations to only a subset of points in each, and taking the mean distance across these splits. The $p$-value is the proportion of null distances exceeding this test statistic. Here, the distance function is the classifier test accuracy used to compute $m(p_X)$.

We also consider Maximum Mean Discrepancy (MMD) as an alternative distance metric for comparing distributions over point clouds (Chiu & Bloem-Reddy, 2023). MMD measures the discrepancy between two distributions without requiring explicit density estimation or distributional assumptions, making it well-suited to high-dimensional settings. The advantage of our method is its flexibility and interpretability, but we show here that the symmetry biases we detected are strong enough to also be detected by MMD, dependent on the choice of kernel. To compute MMD in practice, we use an unbiased empirical estimator with three different kernel choices reflecting natural notions of distance between point clouds. The most naive choice, which we call the Mean/Covar kernel, computes distances between the means and covariances of two point clouds; while simple and efficient, it fails to capture local geometric structure. The Chamfer distance kernel instead measures the average nearest-neighbor distance between two point clouds, capturing local structure but ignoring global shape distribution and density. Finally, the Hausdorff distance kernel replaces the averaging in Chamfer distance with a maximum operation, making it more sensitive to outliers and worst-case geometric discrepancies.

---

**Algorithm 2** $p$-value Computation

---

1: **Input:** Training set $\mathscr{D}_{\text{train}}$, test set $\mathscr{D}_{\text{test}}$, calibration distances sample size $n_1$, actual distances sample size $n_2$, distance function Distance$(\cdot, \cdot)$.
2: **Output:** $p$-value
3: actual_dists $\leftarrow$ []
4: calibration_dists $\leftarrow$ []
$\qquad\qquad\qquad\qquad\qquad\qquad\qquad\qquad$ ▷ Compute calibration distances under null hypothesis
5: **for** $i = 1$ to $n_1$ **do**
6: $\qquad$ Sample training set $\tilde{\mathscr{D}}_{\text{train}}$ and test set $\tilde{\mathscr{D}}_{\text{test}}$ from $\mathscr{D}_{\text{train}}$ and $\mathscr{D}_{\text{test}}$.
7: $\qquad$ Apply rotation transformation to all data
8: $\qquad$ $d_c \leftarrow$ Distance$(\tilde{\mathscr{D}}_{\text{train}}, \tilde{\mathscr{D}}_{\text{test}})$
9: $\qquad$ calibration_dists.append$(d_c)$
10: **end for**
$\qquad\qquad\qquad\qquad\qquad\qquad\qquad\qquad\qquad\qquad\qquad\qquad\qquad$ ▷ Compute actual distances
11: **for** $i = 1$ to $n_2$ **do**
12: $\qquad$ Sample training set $\tilde{\mathscr{D}}'_{\text{train}}$ and test set $\tilde{\mathscr{D}}'_{\text{test}}$ from $\mathscr{D}_{\text{train}}$ and $\mathscr{D}_{\text{test}}$.
13: $\qquad$ Apply rotation transformation to subset of data
14: $\qquad$ $d_a \leftarrow$ Distance$(\tilde{\mathscr{D}}'_{\text{train}}, \tilde{\mathscr{D}}'_{\text{test}})$
15: $\qquad$ actual_dists.append$(d_a)$
16: **end for**
17: $\bar{d}_a \leftarrow \frac{1}{n_2} \sum_{i=1}^{n_2}$ actual_dists$[i]$ $\qquad\qquad\qquad\qquad$ ▷ Compute mean of actual distances
18: count $\leftarrow |\{d_c \in$ calibration_dists $: d_c > \bar{d}_a\}|$
19: $p$-value $\leftarrow \frac{1+\text{count}}{1+n_1}$
$\qquad$ **return** $p$-value

---

### D.10.1 QM9 $p$-VALUES

Figure 36 demonstrates the values used in our computation of the $p$-values for each method (on a row) and different levels of augmentation in the detection dataset (column) for QM9. The $p$-value plots were computed using 20 samples (for each histogram) of size 1k, trained for 20 epochs (in the case of the classifier metric). All methods exhibit the expected behavior: as the augmented fraction increases —i.e. as the distribution becomes more similar to the reference, perfectly symmetrized distribution—the distance decreases.

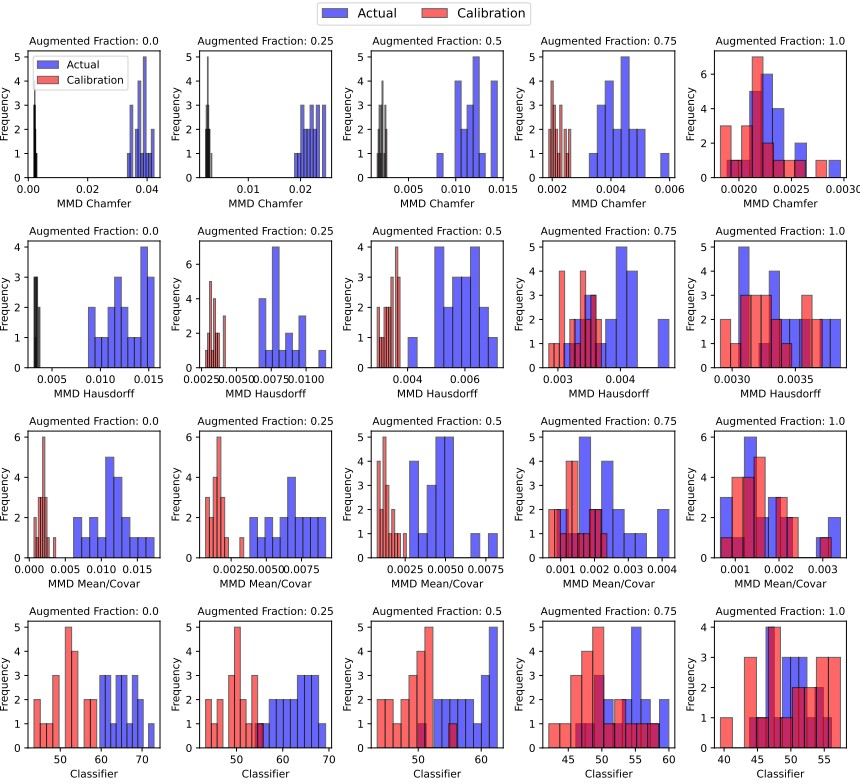

Figure 36: Distance metrics for different methods, and at different levels of augmentation (i.e. different levels of underlying distributional similarity).

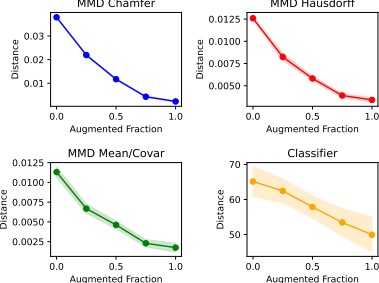

Figure 37: Different distance metrics from a perfectly symmetrized distribution, as a function of the degree of synthetic augmentation of the QM9 dataset. (Higher augmented fraction indicates a greater similarity to the symmetrized distribution.)

### D.10.2    RMD17 $p$-VALUES

Figure 41 demonstrates the values used in our computation of the $p$-values for each method (on a row) and different levels of augmentation in the detection dataset (column) for one of the molecules in rMD17 (benzene). The $p$-value plots were computed using 20 samples (for each histogram) of size 1k corresponding to the given train/test splits, trained for 20 epochs (in the case of the classifier metric). As shown, all methods separate the calibration distances from the actual distances,

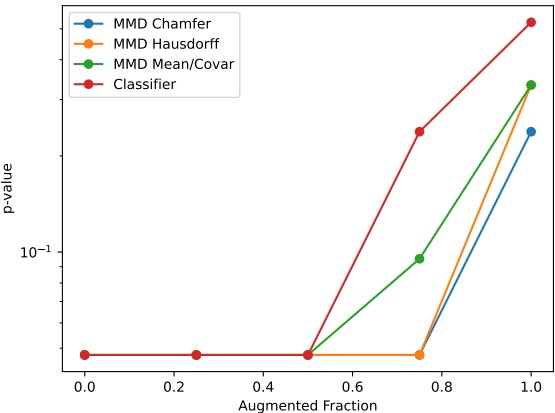

Figure 38: *p*-values for different methods, and at different levels of augmentation (i.e. different levels of underlying distributional similarity).

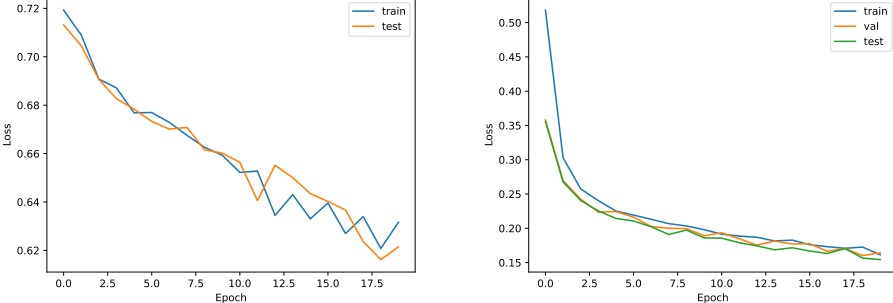

Figure 39: Left: the loss curve from one of the 20 training runs used to compute the classifier distance in the *p*-value computation, on 1k examples. Right: the loss curve from a training run used to compute the classifier distance over the full dataset. As shown, the loss converged much faster for the full dataset, whereas with only 1k examples (one one-hundredth of the size), convergence is much slower.

resulting in identical, statistically significant $p$-values. As the tests are asked to distinguish between increasingly similar datasets (moving from left to right), the histograms gradually move closer together, until they overlap. For ease of visualization, Figure 40 plots the mean distance computed from each histogram for benzene (excluding the calibration distances). We also plot the $p$-value vs. the augmented fraction Figure 42. The Chamfer and Hausdorff kernels exhibit similar trends to the classifier, and the naive mean/covar kernel exhibits less reasonable behavior. This illustrates the importance of choosing a good kernel and provides a relative advantage of our method. All other molecules in rMD17 exhibted similar trends for the $p$-values.

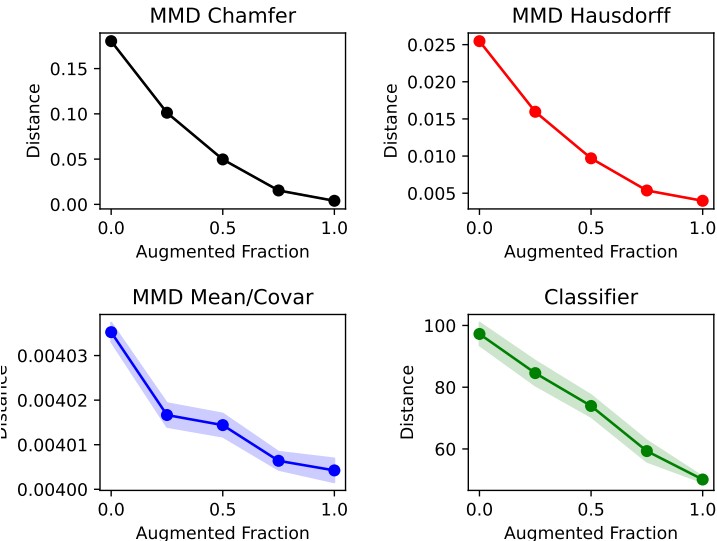

Figure 40: Different distance metrics from a perfectly symmetrized distribution, as a function of the degree of synthetic augmentation of the rMD17 dataset for benezene. (Higher augmented fraction indicates a greater similarity to the symmetrized distribution).

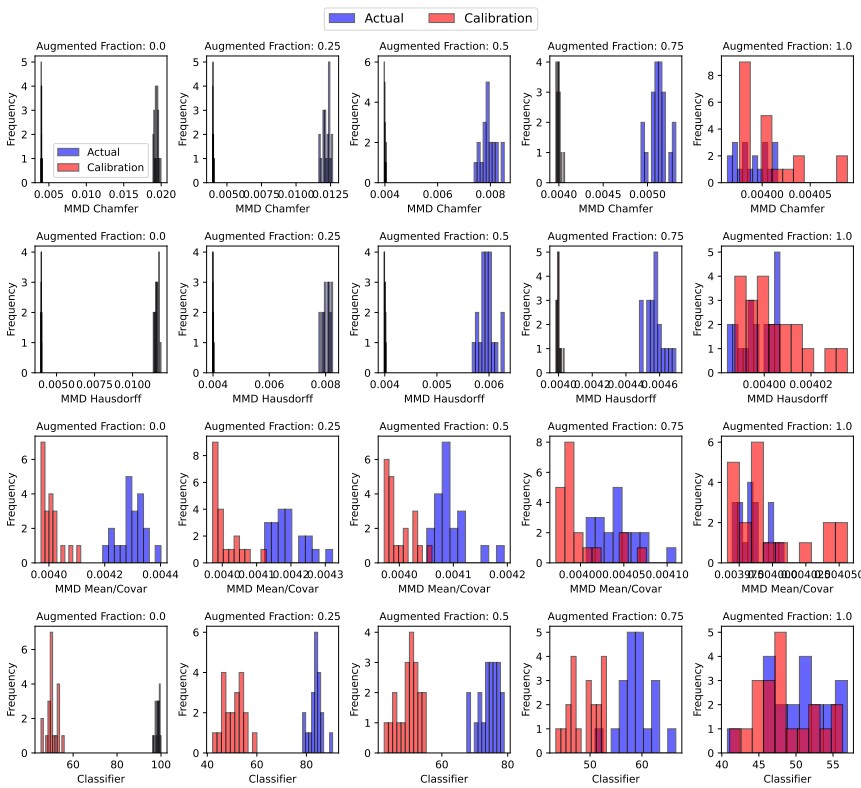

Figure 41: Distance metrics for different methods, and at different levels of augmentation for benzene (i.e. different levels of underlying distributional similarity).

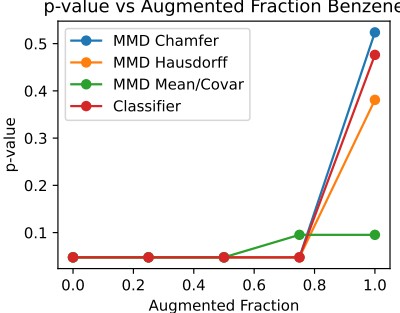

Figure 42: *p*-value vs augmented fraction for benzene rMD17.

