# OpenReview forum: "To Augment or Not to Augment? Diagnosing Distributional Symmetry Breaking"
_ICLR.cc/2026/Conference — ICLR 2026 Poster_

### Official Review · Reviewer_BM93 · 2025-10-31

**Soundness:** 2
**Presentation:** 2
**Contribution:** 2
**Rating:** 2
**Confidence:** 4

**Summary:**

This work proposes a metric based on neural network (NN) binary classifier training to evaluate the degree of distributional symmetry breaking in data. The authors provide both theoretical analyses and empirical validations on several datasets, including MNIST (2D images), ModelNet40 (3D image sets), QM9 and QM7B (molecular data), and large-scale materials datasets.

**Strengths:**

It provides a comprehensive demonstration of the proposed ideas.
It includes theoretical efforts to clarify the impact of data-specific invariant and equivariant properties in terms of generalization.

**Weaknesses:**

1. Unclear Problem Definition

 It is unclear whether this paper aims to propose a new metric, present derived theoretical findings, or introduce a new concept (distributional symmetry breaking). If the goal is to propose a metric, the paper should clearly demonstrate how and why it functions effectively as one. If the focus is on presenting theoretical findings, it should explain why those findings are significant from both theoretical and experimental perspectives. If the work intends to overcome limitations of the underlying assumption, it should show what problems actually occur and that problem is solved by removing the assumption directly. Overall, the paper seems to mix these three goals without providing sufficient support for each direction as follows.

2. Weak Justification of the Impact of Motivation

On page 1, line 42, p(x) is similar to p(gx) is described as the main cause of the problem to be tackled. However, it seems likely observed only in Elesedy & Zaidi’s work. To justify why it is worth tackling, the author needs to provide more explanation and literature works, because equivariance is used in various ways, whether full or partial, implicit or explicit, in practical neural networks, this assumption could have a large impact. However, this assumption is not explicitly made in many existing works.

3. Loose Connection of the Theoretical Works to Contribution

In my view, conclusions such as the one on page 5, line 216, are not novel at all, which the author also notes as intuitive; the same applies to page 6, line 289. If clarifying the conclusions as a theoretical result, it requires justification of the novelty and impact of the "conclusions".

4. Weak Validation As as a Metric

The proposed metric is a binary classifier NN–based approach. Even if it can be used as a test setup (Lopez-Paz & Oquab, 2017), it is a training-based metric, so many hyperparameter conditions affect the variance of test results, which is a significant risk for a metric used in fair comparisons. To make a contribution as a metric, it is necessary to demonstrate robustness to the training environment and sensitivity to actual equivariance changes. (Honestly, it may not be possible to provide data-agnostic standards. Even in this case, to use it as a metric, the risks should be analyzed to determine whether they are within an acceptable range.)


5. A Flaw in Motivation

The distributional symmetry breaking is measured on page 3, line 142. This part seems flawed, because gx may be another observed sample x', which should have p(x'), but it is integrated to be matched to p(x) again. It means that the metric tries to represent one point of the data manifold with a region of the data manifold including it, which reduces the total density of the manifold. This confuses what distribution p(x) the metric is expected to represent at an optimum, given the broken probability mass.

**Questions:**

Minor comments

1. Term Clarification

Terms such as anisotropy, symmetry, equivariance, distributional symmetry-breaking, distributional symmetry, perfectly symmetric, and invariant are used in many places, but they should be introduced only after clear definitions and used more consistently.

---

> ### Author Response · Authors · 2025-11-25
> **Thanks for the review**
>
> We thank the reviewer for taking the time to read and consider our work. We respond to each point below.
> # W1: “Unclear Problem Definition”
> * The problem definition of our paper is: **quantify distributional symmetry breaking and its impact on learning under different model classes.** We thank the reviewer for the opportunity to clarify the goals of our work. The reviewer said, “*It is unclear whether this paper aims to propose a new metric, present derived theoretical findings, or introduce a new concept (distributional symmetry breaking)*”, and that we “*mix these three goals without providing support for each direction.*” We will clarify our explicit goals below, and the strong support we provided for each of them.
> * *“Introduce a new concept (distributional symmetry breaking)”*: We do not claim to have introduced the concept of distributional symmetry breaking. As cited in L048 and L076, the concept of distributional symmetry breaking was introduced (under the name “extrinsic equivariance”) in “A General Theory of Correct, Incorrect, and Extrinsic Equivariance” by Wang et al 2023, and studied further in termed in Wang et al., Discovering symmetry breaking in physical systems with relaxed group convolution, 2024. However, these works introduced the concept without rigorously quantifying its prevalence in real datasets. Doing so was our first goal.
> * *“Propose a new metric”*: In short, the classifier metric is not new (Lopez-Paz & Oquab 2017), but our application of it for detecting distributional symmetry-breaking is. Our first goal was to quantitatively assess how much distributional symmetry breaking is present in benchmark equivariant datasets [L125]. To address this goal, we quantified the amount of distributional symmetry breaking with a two-sample classifier test [Section 2, L135]. Surprisingly, our metric showed that commonly used benchmarking datasets have orientational alignment. **This is our first key contribution.** Given these findings on benchmark datasets, we asked: what does distributional symmetry-breaking imply for practitioners? We sought to answer this hard question through both theory and experiments.
> * *“Present derived theoretical findings”*: Next, we aimed to gain theoretical intuition for when distributional symmetry breaking can be helpful vs. harmful for an actual learning task [L127] (see also the response below to W3). These results are novel, and to our knowledge the first formal study of group augmentation under distributional symmetry breaking. **This is our second key contribution.**
> * *Extensive experiments on learning tasks*: We then study the behavior of equivariant and non-equivariant models on the benchmark datasets, and find that our results align with the theory that data augmentation can either be helpful or harmful under distributional symmetry breaking. Since $m(p_X)$ only detects distributional symmetry breaking, these findings motivate our discussion of a task-dependent metric (see the main response to all reviewers, “Highlighting the task-dependent metric”), as well as locality. **This is our third key contribution.** Together, we believe these results form a coherent narrative of (1) measuring distributional symmetry breaking, (2) understanding its impact on learning through theory, and (3) evaluating its impact on learning empirically. Although it does not fit simply into the box of “a theory paper” or “an empirical paper,” these are the best tools we have available for understanding distributional symmetry breaking, and we believe they are of very real interest to communities working with equivariant datasets.

---

> > ### Author Response · Authors · 2025-11-25
> > **Thanks for the review**
> >
> > # W2: “Weak Justification of the Impact of Motivation”
> > Thanks to the reviewer for the opportunity to clarify our motivation. In fact, the assumption $p(x) \approx g(x)$ is explicitly present in many theoretical papers about equivariant models and data augmentation; we now list some of them.
> > * *“Learning with invariances in random features and kernel models”* by Mei et al. 2020: explicitly assumes that the input distribution is exactly invariant under the group action (uniform distribution over the sphere or hypercube), and proves a corresponding sample complexity improvement compared to non-invariant models.
> > * *“On the Benefits of Invariance in Neural Networks”* by Lyle et al 2020: explicitly assumes invariance of the data distribution (Section 1.1, first page), and proves corresponding improved generalization bounds under data augmentation and feature averaging
> > * *“The Exact Sample Complexity Gain from Invariances for Kernel Regression”* by Tahmasebi et al 2023: assumes uniform distribution over the data manifold, and shows that invariance then imparts an advantage in sample complexity.
> > * *“A Group-Theoretic Framework for Data Augmentation”* by Chen et al. 2020: shows that “when the data is exactly invariant in distribution (exact invariance)...averaging over the group orbit (e.g., all rotations) reduces the variance of any function”, and uses this assumption to characterize the variance reduction obtained.
> > * Moreover, it is common for more empirical works to assume that an equivariant problem will benefit from equivariant augmentation or methods; the equivariance of the underlying function is usually all the justification that is included. This further motivates our work: although not stated explicitly, this logic only holds when there is no distributional symmetry breaking or when one wishes to generalize out-of-distribution, since we demonstrate that distributional symmetry breaking can make equivariant augmentation harmful (again, even if the underlying function is truly equivariant).

---

> > > ### Author Response · Authors · 2025-11-25
> > > **Thanks for the review**
> > >
> > > # W3: “Loose Connection of the Theoretical Works to the Contribution”
> > > We respectfully but strongly disagree; our theoretical contributions are novel, non-trivial, and closely related to the rest of the paper. However, perhaps we did not convey this clearly enough in the submission. We clarify below, with a focus on disambiguating between known and novel results.
> > > * The reviewer points out that our statement that “data augmentation can be harmful when invariant and non-invariant features are strongly correlated” is “not novel at all.” We believe it’s helpful to clarify and distinguish between:
> > >   * what is known on the theory of over-parameterized ridge regression (but we included as context for the equivariance community)
> > >   * what we claim is novel and worth bringing to the attention of the broader community
> > > * *What is known*: that risk blows up at an interpolation threshold is known (see e.g. Hastie et al.), and we in fact cite this result on line 298. While we obtain essentially a similar result for data augmentation (which decreases the ratio of data/samples) do not claim this is novel. We may have given the reviewer this impression, and have thus changed this from a “Theorem” to a “fact”. Nonetheless, this is an effect we believe may not be known to a wide audience in the context of invariant models/data augmentation, so we believe it is valuable to include in the paper.
> > > * *What is novel*: We provide explicit analysis showing that even when the task is invariant ($y=f(x)=f(gx)$), data augmentation/test-time symmetrization can hurt. While there may have been “folklore intuition” (though we do not know of many papers; did the reviewer have any in mind?), we believe a theoretical model confirming this effect is valuable. Put differently, we think it’s valuable to convey that the story is nuanced. Generally, with appropriate distributional (but not functional) symmetry breaking: test-time symmetrization hurts in the under-parameterized regime, while data augmentation may also hurt in the over-parameterized regime. We prove a toy setting in which exactly this effect is demonstrated, with data-augmentation performing worse/better than the vanilla model under more/less symmetry breaking.
> > >   * This is plainly visible in Figure 3, which we have added to the main text. To better illustrate the connection between our theory and our metric, in Figure 3 we also compute our metric on the Gaussian construction from our theory. It indeed detects the distributional symmetry-breaking.
> > >   * Note however that this behavior isn’t obvious — only that it is an example confirming our intuition. In particular, Theorem 2 provides concrete conditions under which symmetry breaking makes data augmentation harmful (in our model). This is not the case for *all* settings. We welcome the reviewer to explore this in our interactive plot: https://www.desmos.com/calculator/kll2v9h5qd
> > > * We view our conclusions being “intuitive” as a strength. Intuition does not always align with what can be shown theoretically: the theory serves as a model for an effect we hypothesized affects data augmentation in practice. Intuition provided post-hoc for a result is generally considered a sign of clear communication, rather than an invalidation of the result itself. Overall, distributional symmetry breaking is a rather counter-intuitive phenomenon (“equivariant methods can hurt even when the true function is equivariant”), so we strove to provide some intuition for the reader.

---

> > > > ### Author Response · Authors · 2025-11-25
> > > > **Thanks for the review**
> > > >
> > > > # W4: “Weak Validation as a Metric”
> > > > In new ablations, we confirmed that our method is not very sensitive to the choice of hyperparameters. We addressed the point about sensitivity to hyperparameter conditions in the overall response, and copy it below for the reviewer’s convenience before addressing the “sensitivity to actual equivariance changes” question.
> > > > * *Existing ablations in appendix*: 3 of 4 reviewers asked or commented on potential sensitivity of our metric to training hyperparameters, such as the architecture. This is a very reasonable point: in fact, in Table 7 of Appendix D.5 in our original submission, we already ran an ablation on the architecture size, and found very little sensitivity to transformer architecture parameters for QM9.
> > > > * *New ablations*: To more thoroughly explore potential sensitivities, we have now run much wider ablation studies on QM9 and MNIST, which can be found in Appendix D.5.1, Figures 33 and 34. We varied classifier model size, model architecture, size of the dataset (by randomly selecting smaller subsets), and learning rate. As shown, the test accuracies were remarkably stable to these hyperparameter variations -- with the exception of using too large a learning rate for the transformer architecture, a problem very easily detectable by checking loss curves (see next bullet point) -- and remained nearly constant for all but the smallest of dataset sizes. This shows the proposed metric does give a useful signal of symmetry breaking, relatively independent of the choice of parameters.
> > > > * *Our classifier metric is intuitive for ML practitioners*: One of the nice things about using the classifier metric is that it draws on very basic skills and intuitions that ML practitioners uniformly already have; namely, training simple networks (e.g. sanity checking loss curves, using a standard learning rate for an architecture, etc). Moreover, metrics that are not training-based have hyperparameters and design choices too (e.g. kernel MMD from Chiu and Bloem-Reddy 2023), but these might be even more opaque to select.
> > > > * *Learning theory roughly predicts the impact of model and dataset size on the metric*: We do not need to rely on ablations alone to predict the impact of model and dataset size on the classifier metric -- high-level ideas in learning theory apply to our metric. For completeness and clarity, we now articulate these results in Appendix B, and provide a very brief summary here.
> > > >    * First, (1) we cite existing results establishing the equivalence between the “ideal” classifier metric (if we had infinite data and perfect optimization) and other well-known quantities (the TV norm, integral probability metrics and $\phi$-divergences) under different choices of loss function and classifier function class.
> > > >    * Second, (2) we then note that the gap between the “ideal” classifier metric, and the “practical” classifier metric, is a classic term in learning theory (the “generalization gap”) and can be bounded using e.g. Rademacher complexity.
> > > >    * Overall, the choice of network architecture affects both (1) and (2): larger, more expressive networks make the “ideal” metric more powerful/discriminative, but they also increase the sample complexity of training them. In contrast, the dataset size only affects the generalization gap, (2). We stress that these are not novel theoretical results, but include them because we think they provide useful intuition and background for $m(p_X)$.

---

> > > > > ### Author Response · Authors · 2025-11-25
> > > > > **Thanks for the review**
> > > > >
> > > > > Returning now to other questions within W4 specific to this reviewer:
> > > > > * *Sensitivity to actual equivariance changes*: We do test this. In Figure 5, we randomly rotate a random X-fraction of the dataset (where X is on the X-axis), and evaluate our metric. We see exactly the behavior you would expect: a smooth decline from high accuracy on the original dataset, down to 50% accuracy on the fully rotated dataset. This is also repeated for MNIST in Figure 13. Similarly, in Figure 6 (left), we compute $m(p_X)$ on both fully rotated, and fully canonicalized, versions of each dataset as a sanity check. As expected, the accuracy for full canonicalization is nearly 100%, while under random rotation is 50%.
> > > > > * If we have misunderstood what the reviewer meant by “actual equivariance changes”, can they please clarify? To reiterate, we only study problems with equivariant labels, i.e. the function $f$ from input $x$ to output $y$ never breaks symmetry. This is motivated by data coming from the physical world (e.g. involving molecules), settings which are arguably where equivariant models have been the most successful. In these settings, the symmetry of the underlying function is a known, irrefutable fact. We find even this setting to be rich and full of complexity, as evidenced in our response to the first point (“Unclear Problem Definition”).
> > > > > * *Relationship to existing metrics/divergences*: To further contextualize the classifier metric, in Appendix B we have added a note of how it relates to integral probability metrics and $\phi$-divergences. However, again, the classifier distance existed already in the literature; what is new (in addition to both our theory and our extensive experiments) is our application of it to the problem of distributional symmetry breaking.
> > > > > * *On the possibility of “data-agnostic standards”*: we agree that it is probably not possible to provide data-agnostic standards: if the distributional symmetry-breaking is very subtle (i.e. if $p$ and $\tilde{p}$ are very close to each other), then we expect (A) a more expressive network class to be needed to distinguish between $p$ and $\tilde{p}$, and (B) more samples to be needed to fit this network (as noted in Appendix B, this is at least what classic learning theory would imply). Thus, the sample efficiency should vary by dataset. In practice, our ablations show that real benchmark datasets have strong enough distributional symmetry breaking, that even very small subsets of the dataset are sufficient to detect it with our metric (high values of $m(p_X)$).

---

> > > > > > ### Author Response · Authors · 2025-11-25
> > > > > > **Thanks for the review**
> > > > > >
> > > > > > # W5: “A Flaw in Motivation”
> > > > > > We appreciate the opportunity to clarify. We think there is a technical misunderstanding here, which we will try our best to clear up. We are defining a new group-averaged metric, $\bar{p}$ (“:=”). It is based on, but not equal to, $p$, although $\bar{p}$ still integrates to $1$ (which we will prove below). The only case when $p=\bar{p}$ is when $p$ was already invariant under the group, i.e. $p(x)=p(gx)$ $\forall x, \forall g$. The referenced line is an application of the Reynolds operator, which is a standard operator from invariant theory and is also sometimes called “group averaging” or “symmetrization” (see e.g. [1], [2], [3]). As noted in the text, $\bar{p}$ is the closest (in terms of L2 norm) invariant measure to the original measure $p$, which is not necessarily invariant due to distributional symmetry breaking. More specifically: the value of the metric $\bar{p}$ at $x$ is equal to the average value of the original metric $p$ along the entire orbit of $x$. Note that this integral is with respect to the [Haar measure](https://en.wikipedia.org/wiki/Haar_measure) associated with the compact group $G$ (basically, the uniform measure over the group). For a discrete group $G$, this means that the integral becomes a sum weighted by $\frac{1}{|G|}$. In general, the measure still integrates to 1. It is easy to show this: $$\int_x \int_g p(gx)dg dx = \int_g \int_x p(gx)dx dg = \int_g \int_y p(y)dy dg = \int_g 1 dg = 1$$ Here, we used a change of variable $y=gx$ for each fixed $g$, using that $g$ acts orthogonally (since $G$ is compact). We are moreover assuming that for any fixed $g$, $\mathcal{X} = g\mathcal{X}$, which is a basic condition that says that $\mathcal{X}$ only contains complete orbits (recall the orbit of $x$ is $\{gx:g\in G\}$. The reviewer may also find the preliminaries in section 3 of [4] illuminating. If this was the source of confusion, we are happy to state it explicitly in the text. If there is still some other confusion remaining, please let us know.
> > > > > >
> > > > > > [1] “Universal approximations of invariant maps by neural networks,” Dmitry Yarotsky, 2018, top of page 8
> > > > > >
> > > > > > [2] “Frame Averaging for Invariant and Equivariant Network Design,” Omri Puny et al, 2022, equation 1
> > > > > >
> > > > > > [3] “Equivariant Frames and the Impossibility of Continuous Canonicalization,” Nadav Dym et al, 2024, equation 1
> > > > > >
> > > > > > [4] “The Exact Sample Complexity Gain from Invariances for Kernel Regression,” Tahmesbi et al, 2023, Section 3

---

> > > > > > > ### Author Response · Authors · 2025-11-25
> > > > > > > **Thanks for the review**
> > > > > > >
> > > > > > > # Q1: Term clarification
> > > > > > > Thanks to the reviewer for pointing out this issue. We have added explicit clarification of the terms used in the introduction. In particular, we have added additional definitions of invariance/equivariance [L037-038], distributional symmetry breaking [L049-051], and isotropic [L081]; we also make a note that canonicalization is a case of strong distributional symmetry breaking [L083-084]. In this work, as our datasets have high $m(p_X)$ and are orientationally aligned, we say that they are canonicalized. Please let us know if there are additional questions regarding these terms.
> > > > > > >
> > > > > > > Again, we thank the reviewer for taking the time to engage with us, and remain available for further discussion.

---

### Official Review · Reviewer_RG36 · 2025-11-01

**Soundness:** 2
**Presentation:** 2
**Contribution:** 2
**Rating:** 6
**Confidence:** 2

**Summary:**

The paper explains that data augmentation and symmetrization can be harmful when invariant and non-invariant features are strongly correlated, and proposes a novel metric that measures the distributional symmetry breaking in a dataset.

**Strengths:**

The paper provides a principled analysis of when data augmentation benefits or harms generalization.

**Weaknesses:**

While the paper presents a compelling framework for selective data augmentation, it could be strengthened by a deeper exploration of its practical limitations and boundary conditions. For instance, the theoretical analysis assumes access to accurate estimates of distributional alignment and bias-variance components, but the paper provides limited discussion on how these quantities can be reliably approximated in high-dimensional, real-world settings.

**Questions:**

The paper’s framework relies on specific assumptions about the data distribution and the augmentation operator family. Could the authors clarify how restrictive these assumptions are in practice? For example, do they hold for common image augmentations such as Mixup or Cutout?

---

> ### Author Response · Authors · 2025-11-25
> **Thanks for the review**
>
> Thank you to the reviewer for their comments, and for noting the novelty of our distributional symmetry breaking metric.
> # Weakness 1: deeper exploration of practical limitations and boundary conditions
> * First, we’d like to clarify some points about the theoretical analysis. The theoretical analysis presents concrete proof that there are settings where, even when the ground truth function is invariant, data augmentation can be harmful. The purpose of the theory is conceptual and to provide intuition for where symmetrization helps or hurts, by studying a particular setting in which we know how to calculate the bias and variance. Theoretical results in machine learning necessarily rely on assumptions, which is why we complement them with extensive empirical evaluation.  In particular, accurate estimations of “distributional alignment” or the bias-variance components are not required to compute our metric $m(p_X)$. Rather, the metric is computed directly from the data using the classifier distance presented in the paper. We have also added Figure 3 strengthening the relation between the theory and the metric; see general response.
> * We assume that by boundary conditions, the reviewer is referring to the limiting cases of our metric and framework. We discuss and validate these cases in the paper:
>   * The dataset is already invariant with respect to a given group (e.g. it contains no distributional symmetry breaking). No network can distinguish between samples from the original dataset and a group-averaged version, so $m(p_X) \approx ½$ (chance level for the classifier).
>   * The dataset exhibits a single canonical orientation. In this case $m(p_X) \approx 1$.
> * Our metric interpolates between these two cases. We show that in the paper that it behaves as expected at these boundaries. For example, see Figure 4 where we show that for different datasets, $m(p_X)$ drops to 50\% when considering a fully randomly rotated version of the dataset. Could the reviewer clarify if they meant something different by boundary conditions?
> * We also will add more discussion of our method’s limitations. In particular, we note that in the case where $m(p_X)$ is high, but the task-dependent metric is low, it is not completely clear whether to use an equivariant method or not (and may depend on the domain of interest, see the flowchart in Appendix C.4, Figure 10). Nonetheless, one can use the task-independent metric $m(p_X)$ to determine whether there is distributional symmetry breaking present in their data, which is a valuable insight unto itself.
> # Question 1: Assumptions on data distribution/augmentation operator
> * Thank you for the opportunity to clarify our work. Our paper is meant to address augmentation specifically in the context of equivariant learning [e.g. 1], i.e. specifically “group augmentations”. These are extremely common, with AlphaFold3 a high-profile use case [2, 3, 4, 5]. Our assumptions on the data distribution are quite minimal beyond this, as we just assume that data points drawn from any distribution $p_X$ are acted on by a compact group $G$, in total alignment with the equivariance literature [1]. Therefore, our augmentations correspond to permutations, rotations, etc (actions of symmetry group elements). This is necessary for the interpretation of $m(p_X)$ as a measure of symmetry breaking.
> * Augmentations like Mixup or Cutout do not form a symmetry group, as they do not satisfy the axioms of an algebraic group. For example, Cutout is not invertible, since the information is simply lost. Thus, while one can always compute a classifier distinguishing between the original and augmented data as a measure of how “in-distribution” the augmentations are, this wouldn't provide the same notion of **symmetry** breaking, specifically.
> * This reviewer question highlights an interesting relation between generic “data augmentation” and our work. In particular, it is clear that augmentations like Mixup and Cutout induce a distribution shift, as they move images “off” the natural data manifold: an image that has been “mixed up” is clearly identifiable as a non-natural image. And yet, they still aid in task performance. This is analogous to the QM9 properties that are aided by augmentation despite the clear distribution shift (in this case, due to distributional symmetry breaking) that augmentation entails. That is, rotational data augmentations often improve performance on the test set, despite the fact that rotated versions of the molecules are out of distribution for the test set.
>
> [1] “Theoretical Aspects of Group Equivariant Neural Networks,” Esteves 2020.
>
> [2] “Accurate structure prediction of biomolecular interactions with AlphaFold3,” Abramson et al 2024
>
> [3] “Fine-Tuned Language Models Generate Stable Inorganic Materials as Text,” Gruver et al 2024
>
> [4] “Equivariance versus Augmentation for Spherical Images,” Gerken et al 2022
>
> [5] “Do we need equivariant models for molecule generation?” Nowara et al 2025

---

### Official Review · Reviewer_uapy · 2025-11-02

**Soundness:** 3
**Presentation:** 3
**Contribution:** 3
**Rating:** 4
**Confidence:** 3

**Summary:**

This paper investigates the often-overlooked issue of distributional symmetry breaking, situations where data transformations (e.g., rotations, permutations) are not equally likely under the data distribution, violating the common assumption behind data augmentation and equivariant architectures.
The authors propose a simple yet powerful metric m(pX) based on a two-sample neural classifier test to quantify how far a dataset deviates from perfect symmetry. They show how this measure can diagnose anisotropy in various datasets such as MNIST, ModelNet40, and QM9.
Through theoretical analysis on invariant ridge regression, they demonstrate that data augmentation can harm performance under asymmetric covariance when invariant and non-invariant features are correlated. Extensive experiments across 2D, 3D, and molecular datasets confirm the theory, revealing that many commonly used benchmarks are surprisingly canonicalized.
Overall, the paper argues that understanding when and why equivariance helps requires diagnosing symmetry biases in the data itself.

**Strengths:**

The paper introduces the notion of distributional symmetry breaking as a fundamental factor explaining inconsistencies in the effectiveness of equivariant or augmentation-based methods. This is an important conceptual contribution.

The proposed metric m(pX), implemented via a two-sample classifier, is elegant, interpretable, and easily applicable across data modalities (images, point clouds, molecules). It can serve as a general diagnostic tool for symmetry analysis.

The ridge regression analysis under asymmetric covariance provides deep insight into why augmentation may hurt, connecting symmetry assumptions to bias–variance trade-offs and data geometry.

**Weaknesses:**

While m(pX) quantifies symmetry breaking, the paper does not provide a formal correlation analysis between this metric and downstream model accuracy or robustness — the connection remains mostly qualitative.

Since m(pX) depends on a trained neural network, results may vary with architecture, capacity, and hyperparameters. The paper claims low sensitivity, but further ablation or calibration would strengthen this claim.

**Questions:**

Is there an empirical relationship between m(pX) and the observed gain/loss from equivariant methods? For example, could a threshold on m(pX) predict when augmentation is beneficial?

Can the proposed metric be used during training to adaptively modulate augmentation strength or enforce “symmetry regularization”? This could make the method more practical.

---

> ### Author Response · Authors · 2025-11-25
> **Thanks for the review**
>
> # Weakness 1/Question 1: formal analysis between metric and downstream model accuracy
> * A fully detailed, comprehensive relationship between the metric and downstream performance would be ideal. However, we show in our work that this is implausible. Even in the analytically tractable setting of Theorem 2, we find that while data augmentation can be harmful under some particular instances of distributional symmetry breaking, in others it is not clear. (We’ve taken the reviewers’ feedback and edited the section to emphasize this.)
> * We set up an interactive version of the theoretical predictions for Figure 3 https://www.desmos.com/calculator/kll2v9h5qd if the reviewer would like to explore this varied behavior more!
> * To follow up with the reviewer’s comment, we additionally relate this theory to our proposed classifier-based metric. We evaluate the metric in this theoretical setting, and find it correctly detects symmetry breaking, and is especially sensitive to the high-symmetry-breaking regime which in Figure 3 is shown to make data augmentation harmful.
> * Furthermore, we provide more explanation of the task-dependent metric (see the official comment to all reviewers). We add additional experiments in Section 5.2 (see the response to Reviewer oMVy) showing that the task-dependent metric is larger for tasks where equivariance does \emph{not} improve performance.
> * *Flowchart*: We also added a flowchart in Appendix C.4, Figure 10 showing how to connect our metrics to downstream model performance.
> * Thus, while a clean formal story may be out of reach, we believe our work provides a significant step in that direction.
> # Weakness 2: impact of architecture, capacity, hyperparameters
> We ran more ablations of our metric, confirming low sensitivity to architecture and hyperparameter choices.  We addressed this point in the main response, copied below.
> * *Existing ablations in appendix*: 3 of 4 reviewers asked or commented on potential sensitivity of our metric to training hyperparameters, such as the architecture. This is a very reasonable point: in fact, in Table 7 of Appendix D.5 in our original submission, we already ran an ablation on the architecture size, and found very little sensitivity to transformer architecture parameters for QM9.
> * *New ablations*: To more thoroughly explore potential sensitivities, we have now run much wider ablation studies on QM9 and MNIST, which can be found in Appendix D.5.1, Figures 33 and 34. We varied classifier model size, model architecture, size of the dataset (by randomly selecting smaller subsets), and learning rate. As shown, the test accuracies were remarkably stable to these hyperparameter variations -- with the exception of using too large a learning rate for the transformer architecture, a problem very easily detectable by checking loss curves (see next bullet point) -- and remained nearly constant for all but the smallest of dataset sizes. This shows the proposed metric does give a useful signal of symmetry breaking, relatively independent of the choice of parameters.
> * *Our classifier metric is intuitive for ML practitioners*: One of the nice things about using the classifier metric is that it draws on very basic skills and intuitions that ML practitioners uniformly already have; namely, training simple networks (e.g. sanity checking loss curves, using a standard learning rate for an architecture, etc). Moreover, metrics that are not training-based have hyperparameters and design choices too (e.g. kernel MMD from Chiu and Bloem-Reddy 2023), but these might be even more opaque to select.
> * *Learning theory roughly predicts the impact of model and dataset size on the metric*: We do not need to rely on ablations alone to predict the impact of model and dataset size on the classifier metric -- high-level ideas in learning theory apply to our metric. For completeness and clarity, we now articulate these results in Appendix B, and provide a very brief summary here.
>    * First, (1) we cite existing results establishing the equivalence between the “ideal” classifier metric (if we had infinite data and perfect optimization) and other well-known quantities (the TV norm, integral probability metrics and $\phi$-divergences) under different choices of loss function and classifier function class.
>    * Second, (2) we then note that the gap between the “ideal” classifier metric, and the “practical” classifier metric, is a classic term in learning theory (the “generalization gap”) and can be bounded using e.g. Rademacher complexity.
>    * Overall, the choice of network architecture affects both (1) and (2): larger, more expressive networks make the “ideal” metric more powerful/discriminative, but they also increase the sample complexity of training them. In contrast, the dataset size only affects the generalization gap, (2). We stress that these are not novel theoretical results, but include them because we think they provide useful intuition and background for $m(p_X)$.

---

> > ### Author Response · Authors · 2025-11-25
> > **Thanks for the review**
> >
> > # Question 2: Use of metric to adaptively change augmentation strength
> > * Using our metric to adaptively modulate augmentation is a great idea, and something we are certainly excited about as a future direction. In Appendix D.2.4 (which has all the details), we added a new preliminary experiment trying this out on MNIST. We threshold the probabilities output by the trained classifier to decide which rotation augmentations to sample from, only augmenting with rotations that the classifier thinks are from the original dataset. We experiment with three variants of MNIST: with the original dataset (“None''), a rotated version (“All''), and a partially rotated version with only certain digits rotated (“345''). To make the task harder and therefore benefitting from augmentation, we randomly select a subset of 1,000 images to use. As expected, ordinary full augmentation works best on “All”, when the whole dataset is rotated, but no augmentation works best on “345” and “None”, when most of the dataset remains aligned. **The selective augmentation method described above automatically interpolates between no augmentation and full augmentation, performing near the best of the two in each case.** These are promising first results, and can be investigated further in future work.
> > * This line of work is related to other works, like Augerino (Benton et al 2020), which learn how to augment from the data. However, unlike Augerino (which backpropagates through a differentiable augmentor), we do not need any labels. Rather than optimizing for a specific task, this approach essentially tries to identify which augmentations leave the original datapoint in-distribution. It does not address task-dependence, or when augmentations may still impart benefits despite inducing distribution shift -- these are good directions for future work. However, using our classifier to selectively augment can at least provide some improvement.

---

### Official Review · Reviewer_oMVy · 2025-11-04

**Soundness:** 3
**Presentation:** 3
**Contribution:** 2
**Rating:** 4
**Confidence:** 4

**Summary:**

This paper investigates the effect of an often overlooked issue in datasets of equivariant tasks, namely that many training data distributions are not actually invariant to the transformation that the trained equivariant model is. To this end, the authors first propose a simple metric to measure the distributional symmetry breaking of a given dataset. The metric uses a single classifier that tries to detect whether an input came from the original training distribution or from an augmented version of the same distribution. In a perfectly invariant data distribution, such a classifier can only do random predictions since the two distributions are identical. The authors propose to use the performance of such a classifier to define a quantitative measure of the degree of symmetry breaking. Along with the definition of the metric for measuring the distributional symmetry breaking, the authors use ridge regression to provide conditions under which data augmentation may actually hurt performance. Finally, they evaluate the proposed metric and theoretical insights across a large range of datasets, and show that many commonly used datasets for equivariant learning exhibit substantial distributional asymmetry.

**Strengths:**

- The paper's topic is quite timely, since there is an open discussion regarding the benefits of equivariants compared to the most commonly used data augmentations. The proposed metric provides a practical and easily implementable tool for testing the distributional invariant of a distribution. This can help researchers better understand when and why equivariant models or data augmentation are preferable.
- The theoretical analysis using ridge regression is clearly presented and provides formal examples that illustrate how augmentation can either hurt or improve performance. Although this analysis is performed in a simplified linear setting, it allows the readers to build intuition for the behavior of the more complex neural networks
- The authors perform an extensive evaluation, covering a large range of tasks and showcasing interesting behaviors (even when they might not explicitly align with the main hypothesis of this work).

**Weaknesses:**

- There is limited discussion and investigation about a task-dependent symmetry-breaking metric. This limitation is crucial since in a lot of different settings, even in cases when the input data distribution is invariant, the ground truth labels break the symmetry. Without being able to effectively model this task-specific symmetry breaking, the proposed metric can be less useful since it only shows a partial image of the problem.
- The results in the molecular datasets, such as QM9, contradict the main point of the paper. Although this is an important result that should be reported, it creates a more ambiguous interpretation regarding the utility of the proposed metric.
- There is limited discussion and ablation studies about how the expressivity of the classifier and also the availability of training data can affect the proposed metric.

**Questions:**

- How the number of training data and the parameter count of the classifier can affect the proposed metric. Is there any proposed heuristic for tuning such parameters, given a dataset we want to investigate?
- How can a practitioner decide whether to use the task-dependent or the task-independent metric to measure the symmetry breaking of a given distribution?

---

> ### Author Response · Authors · 2025-11-25
> **Thanks for the review**
>
> Thanks to the reviewer for their thoughtful evaluation of our work, and for noting both the timeliness and the significant breadth of our investigation. We respond to individual points below.
>
> # Weakness 1: limited discussion of task-dependent metric
> * *Clarifying what we mean by “task-dependent”:* Thanks to the reviewer for this question. There is possibly a miscommunication regarding what we meant by “task-dependent metric”. This paper is solely about tasks which are genuinely equivariant, i.e. the ground truth labels **never** break symmetry. This tends to be the case for molecular and physical regression datasets, where physical laws dictate that the ground truth labels obey certain invariances. What the reviewer describes is sometimes called “functional symmetry breaking” [Wang et al, 2024], and it is addressed in several other papers [Finzi et al., 2021; McNeela, 2023; Smidt et al., 2021; Urbano & Romero, 2024, Wang et al, 2024].
> * *Clarifying what we mean by “distributional symmetry breaking”:* Instead, the very surprising phenomenon we showcase in focus on in this paper is that, even when the labels themselves don’t break symmetry, the underlying (unlabeled) data distribution often does -- and this can sometimes mean one shouldn’t augment or use an equivariant model, even though the labels don’t break symmetry at all. This is “distributional symmetry breaking”.
> * *Our task-dependent metric:* Having clarified this, we defined a “task-dependent metric” to answer the question: when there is distributional symmetry breaking, is it relevant or useful to the specific task at hand? Even assuming the task is equivariant, the specific way in which data-level symmetry is broken can have implications for whether equivariant methods help or hinder learning.  For example, data may or may not be canonicalized in a helpful way (what we called “inherent” in the introduction and Figure 1). To illustrate the concept, consider the following example.
> * *QM7b Dipole Canonicalization:* This is a physics-based toy example where an equivariant task may benefit from task-relevant symmetry breaking. QM7b is a dataset of small molecules containing non-scalar material response properties. Consider predicting the dipole moment (a vector) from input molecular structures. An “easy” task-relevant canonicalization would be to align all dipole moments with a certain axis, and then align the molecules correspondingly. **This task-relevant symmetry breaking clearly makes it easier for a non-equivariant model to solve the task, whereas an equivariant model will be unable to use this orientation information.** Indeed, we verify empirically that no augmentation (FF) outperforms an equivariant model with this task-relevant canonicalization.

---

> > ### Author Response · Authors · 2025-11-25
> > **Thanks for the review**
> >
> > (cont'd)
> > * *New task-dependent metric experiments:* For our task dependent metric, we predict the label $f(x)$ from the orientation $c(x)$. To contextualize this number, we compare to the baseline of predicting $f(x)$ from $c(x)$ where the data is randomly rotated (so there should be no task-relevant information).  See the main response for the full definition of the task-dependent metric. In response to the reviewer’s interest, we also ran new experiments with our task-dependent metric, see the revised Section 5.2 L451-480. Below is a brief summary.
> >   * *QM7b Dipole (canonicalized).*
> >  This is an extreme, proof-of-concept case of task-relevant symmetry breaking. Since dipoles are all aligned, we expect—and observe—a large task-dependent signal.
> >   * *QM7b Dipole (original)*.
> > The task-dependent metric shows only a weak signal, aligning well with the fact that the equivariant model performs best on the unmodified dataset.
> > * *ModelNet.*
> >  Equivariance/data augmentation harm performance in this domain, and the metric correspondingly shows a large signal.
> > * *QM9.*
> > We include QM9 for comparison as a case where there is a smaller task-dependent signal, compared to the other datasets. This is consistent with equivariant architectures working well on QM9. However, we do note that for the property with the most significant task-dependent signal, train-time augmentation is harmful.
> > Here, we report the accuracy (ModelNet) or MAE (QM7b/QM9) of predicting $f(x)$ from $c(x)$, compared to the baseline of rotating the data and then predicting $f(x)$ from $c(x)$.
> > The relative signal quantifies how much $\mathcal{L}$ changes when random rotations are applied.
> > | Dataset                         | $\mathcal{L}$ | $\mathcal{L}_{\text{rot}}$ | Relative Improvement $(\mathcal{L}$ vs. $\mathcal{L}_{\text{rot}})$ |
> > |---------------------------------|---------------|-----------------------------|----------------------------------------------------------------------|
> > | QM7b Dipole $\mu$ ($\downarrow$)      | 0.12          | 0.45                        | **3.75**                                                             |
> > | QM7b Orig $\mu$ ($\downarrow$)        | 0.43          | 0.45                        | 1.04                                                                 |
> > | ModelNet ($\uparrow$)                 | 12            | 2                           | **6**                                                                 |
> > | QM9 $C_v$ ($\downarrow$)             | 3.02          | 3.19                        | 1.05                                                                 |
> > | QM9  $\| \vec{\mu} \|$ ($\downarrow$)      | 1.12          | 1.16                        | 1.04                                                                 |
> > | QM9 $G$ ($\downarrow$)               | 30.5          | 30.83                        | 1.01                                                                 |
> > Thus, our preliminary experiments show that the task-dependent metric is generally larger for tasks where equivariance does *not* improve performance. While this trend is not a universal decision rule, it may still provide useful hints to practitioners, as indicated by the new flowchart [Appendix C.4, Figure 10].
> > # Weakness 2: molecular datasets “contradict the main point of the paper”
> > * First, thanks to the reviewer for acknowledging that our QM9 finding is “an important result that should be reported.” We’d like to clarify that one of the several key points of our paper is that commonly used molecular benchmarking datasets have orientational alignment present in the first place, and that our metric detects these alignments. These contributions are unrelated to the QM9 result on downstream tasks.
> > * Second, we would like to note that the performance on QM9 is a subtle but important result. The result that equivariant models generally outperform non-equivariant models on QM9 property regression tasks generally aligns with the literature (for example, Equiformerv2 [Liao et al 2023]), but is surprising in light of our findings on QM9’s distributional symmetry breaking. We present alternative hypotheses for the success of equivariant models on molecular datasets, such as the prevalence of local atomic motifs (e.g. see Section 5.3, Locality Experiments) and task-dependence (Section 5.2).

---

> > > ### Author Response · Authors · 2025-11-25
> > > **Thanks for the review**
> > >
> > > # Weakness 3 / Question 1: is there sensitivity to hyperparameters, including amount of data?
> > > We addressed this point in the overall response, and copy it below for the reviewer’s convenience.
> > > * *Existing ablations in appendix*: 3 of 4 reviewers asked or commented on potential sensitivity of our metric to training hyperparameters, such as the architecture. This is a very reasonable point: in fact, in Table 7 of Appendix D.5 in our original submission, we already ran an ablation on the architecture size, and found very little sensitivity to transformer architecture parameters for QM9.
> > > * *New ablations*: To more thoroughly explore potential sensitivities, we have now run much wider ablation studies on QM9 and MNIST, which can be found in Appendix D.5.1, Figures 33 and 34. We varied classifier model size, model architecture, size of the dataset (by randomly selecting smaller subsets), and learning rate. As shown, the test accuracies were remarkably stable to these hyperparameter variations -- with the exception of using too large a learning rate for the transformer architecture, a problem very easily detectable by checking loss curves (see next bullet point) -- and remained nearly constant for all but the smallest of dataset sizes. This shows the proposed metric does give a useful signal of symmetry breaking, relatively independent of the choice of parameters.
> > > * *Our classifier metric is intuitive for ML practitioners*: One of the nice things about using the classifier metric is that it draws on very basic skills and intuitions that ML practitioners uniformly already have; namely, training simple networks (e.g. sanity checking loss curves, using a standard learning rate for an architecture, etc). Moreover, metrics that are not training-based have hyperparameters and design choices too (e.g. kernel MMD from Chiu and Bloem-Reddy 2023), but these might be even more opaque to select.
> > > * *Learning theory roughly predicts the impact of model and dataset size on the metric*: We do not need to rely on ablations alone to predict the impact of model and dataset size on the classifier metric -- high-level ideas in learning theory apply to our metric. For completeness and clarity, we now articulate these results in Appendix B, and provide a very brief summary here.
> > >    * First, (1) we cite existing results establishing the equivalence between the “ideal” classifier metric (if we had infinite data and perfect optimization) and other well-known quantities (the TV norm, integral probability metrics and $\phi$-divergences) under different choices of loss function and classifier function class.
> > >    * Second, (2) we then note that the gap between the “ideal” classifier metric, and the “practical” classifier metric, is a classic term in learning theory (the “generalization gap”) and can be bounded using e.g. Rademacher complexity.
> > >    * Overall, the choice of network architecture affects both (1) and (2): larger, more expressive networks make the “ideal” metric more powerful/discriminative, but they also increase the sample complexity of training them. In contrast, the dataset size only affects the generalization gap, (2). We stress that these are not novel theoretical results, but include them because we think they provide useful intuition and background for $m(p_X)$.
> > > # Question 2: deciding between task-independent and task-dependent metrics in practice
> > > * **TL;DR We added a practical flowchart showing when to use each metric in Appendix C.4, Figure 10, to answer this question.** We summarize it below.
> > > * In practice, we suggest starting with the task-independent metric, which was our primary focus in the paper. If this metric yields a higher-than-random (i.e. better than 50%) accuracy, it is a warning signal to the practitioner that their data has distributional symmetry-breaking: it is not oriented at random.
> > >   * If the practitioner is then curious whether or not the distributional symmetry-breaking has any correlation with a label of interest, they can subsequently try the task-dependent metric.
> > > * Ultimately, what our theory, and QM9 regression, $m(p_X)$, and task-dependent metric results show, is that it is hard to predict when augmentation will hurt 100% of the time. But, if $m(p_X)$ is around 50%, we can be reasonably confident that augmentation will help because it does not induce a change in distribution. (In other words, to good approximation, $m(p_X)$ is necessary but not sufficient for augmentation to harm learning.) Moreover, if the task-dependent metric is high, it is more likely that augmentation will hurt.
> > >
> > > Again, we thank the reviewer for taking the time to review our work, and remain available for further discussion.

---

### Author Response · Authors · 2025-11-25
**Official comment to all reviewers**

We thank all the reviewers for taking the time to carefully consider our work, and for their useful feedback. We make a few general comments here. All revisions to the PDF are shown in red.

# New experiment: ablations on network architecture, hyperparameters, and dataset size
* *Existing ablations in appendix*: 3 of 4 reviewers asked or commented on potential sensitivity of our metric to training hyperparameters, such as the architecture. This is a very reasonable point: in fact, in Table 7 of Appendix D.5 in our original submission, we already ran an ablation on the architecture size, and found very little sensitivity to transformer architecture parameters for QM9.
* *New ablations*: To more thoroughly explore potential sensitivities, we have now run much wider ablation studies on QM9 and MNIST, which can be found in Appendix D.5.1, Figures 33 and 34. We varied classifier model size, model architecture, size of the dataset (by randomly selecting smaller subsets), and learning rate. As shown, the test accuracies were remarkably stable to these hyperparameter variations -- with the exception of using too large a learning rate for the transformer architecture, a problem very easily detectable by checking loss curves (see next bullet point) -- and remained nearly constant for all but the smallest of dataset sizes. This shows the proposed metric does give a useful signal of symmetry breaking, relatively independent of the choice of parameters.
* *Our classifier metric is intuitive for ML practitioners*: One of the nice things about using the classifier metric is that it draws on very basic skills and intuitions that ML practitioners uniformly already have; namely, training simple networks (e.g. sanity checking loss curves, using a standard learning rate for an architecture, etc). Moreover, metrics that are not training-based have hyperparameters and design choices too (e.g. kernel MMD from Chiu and Bloem-Reddy 2023), but these might be even more opaque to select.
* *Learning theory roughly predicts the impact of model and dataset size on the metric*: We do not need to rely on ablations alone to predict the impact of model and dataset size on the classifier metric -- high-level ideas in learning theory apply to our metric. For completeness and clarity, we now articulate these results in Appendix B, and provide a very brief summary here.
   * First, (1) we cite existing results establishing the equivalence between the “ideal” classifier metric (if we had infinite data and perfect optimization) and other well-known quantities (the TV norm, integral probability metrics and $\phi$-divergences) under different choices of loss function and classifier function class.
   * Second, (2) we then note that the gap between the “ideal” classifier metric, and the “practical” classifier metric, is a classic term in learning theory (the “generalization gap”) and can be bounded using e.g. Rademacher complexity.
   * Overall, the choice of network architecture affects both (1) and (2): larger, more expressive networks make the “ideal” metric more powerful/discriminative, but they also increase the sample complexity of training them. In contrast, the dataset size only affects the generalization gap, (2). We stress that these are not novel theoretical results, but include them because we think they provide useful intuition and background for $m(p_X)$.

---

> ### Author Response · Authors · 2025-11-25
> **Official comment to all reviewers**
>
> # Highlighting the task-dependent metric
> * We further explain and develop the task-dependent metric in response to Reviewer oMVy. We would also like to highlight this contribution to other reviewers; see updated Section 5.2 [L451-480]. We had already defined this in the appendix in the original submission [Appendix C.1], but we did more extensive experiments in response to reviewer feedback.
> * *Flowchart*: We also added a flowchart for practitioners (showing when to use each metric) in Appendix C.4, Figure 10. We encourage the reviewers to take a look, as it clarifies our recommendations for practitioners.
> * *Intuition*: The task-dependent metric was intended to measure the dependence between the input orientation and the label. (The intuition is that, if the input orientation has some non-trivial dependence with the label, then any equivariance-enforcing methods like augmentation or canonicalization are discarding potentially useful predictive features.) Since the input orientation is not inherently defined, we use a small, untrained equivariant network $c(\cdot)$ to define it -- then, $c(x) \in G$ is interpreted as the input’s orientation.
> * *Definition*: To assess the dependence between $c(x)$ and $f(x)$, we directly predict $f(x)$ from the orientation $c(x)$. We then compare the test criterion $\mathcal{L}(c(x) \to f(x))$ to that obtained when the inputs are randomly transformed by elements of the given group,  $\mathcal{L}_{\text{rot}} = \mathcal{L}(c(gx) \to f(gx)), g \sim G$  which removes any task-relevant information in the orientations (where $\mathcal{L}$ is the performance on the test set, e.g. accuracy/MAE).
> * *Additional Experiments*: We provide additional experiments in Section 5.2 (see the response to Reviewer oMVy) showing that the task-dependent metric is larger for tasks where equivariance does \emph{not} improve performance. We note that if the task-dependent metric is better than on the randomly rotated dataset, it demonstrates that any non-equivariant model can learn some amount of information from the orientations present. This is a useful insight in its own right, since non-equivariant models might exploit specific orientations but fail to generalize to datasets containing other orientations.
> # New experiment: computing $m(p_X)$ on the theoretical construction
> * Please see the new figure, Figure 3, demonstrating the harm of data augmentation in the overparametrized ridge regression regime when there are fewer correlational modes (in the model from Section 4.2) than ambient dimensions (which induces distributional symmetry breaking). (Note: the left side of this figure was previously in the appendix, but we moved it into the main body to emphasize it.)
> * Left of figure: we demonstrate that at high levels of distributional symmetry-breaking (left side of x-axis), augmentation has higher excess risk. As the degree of distributional symmetry-breaking decreases, there is a crossover point and augmentation helps again.
> * Right of figure: to better connect our metric to our theoretical results, we computed $m(p_X)$ on this data. Indeed, on this dataset, $m(p_X)$ can pick up reasonably well on the distributional symmetry breaking that makes augmentation harmful, with test accuracy of 65% (compared to 50% for random guessing).
> # Noting contributions related to locality and LLM dataset:
> * We would like to note some perhaps overlooked contributions, which we believe provide additional value to our submission.
> * Locality: (Section 5.3, L480). Here, we present the hypothesis that part of the success of equivariant models on canonicalized molecular datasets can be explained by locally symmetric features. We use $m(p_X)$ to concretely determine that local atomic neighborhoods in QM9 are less canonicalized than entire molecules.
> * LLM dataset: There is growing interest in training LLMs on materials science data. A previous study (Gruver et. al 2024) noted that permutation augmentations on an LLM materials dataset hurt model performance (even though the task was permutation invariant). We postulated this was due to distributional symmetry breaking, and indeed, $m(p_X)$ had 95% accuracy. Conversion from molecules to text provides another area in which distributional symmetry-breaking is likely to be very relevant.

---

> > ### Author Response · Authors · 2025-11-25
> > **Official comment to all reviewers**
> >
> > # Relevance of observation:
> > * It has recently come to our attention that there is another submission to ICLR (https://openreview.net/pdf?id=zrCGvLOrTL) entitled “Take Note: Your Molecular Dataset is Probably Aligned”. This work similarly proposes a binary classifier for assessing the distributional symmetry-breaking (or “alignment”) present in a dataset, and applies it to molecular datasets, demonstrating their alignment.
> > * We highlight this work for two reasons. First, both the work itself and its enthusiastic reviews demonstrate the timeliness and general community interest of this line of investigation.
> > * Second, we believe that our work takes the question of molecular dataset alignment significantly further. Like this submission, we propose a classifier-based metric and apply it to conclude that molecular datasets are often aligned. Like this submission, we also note the implication for non-equivariant models–that if evaluated only on in-distribution validation data, non-equivariant models may appear accurate, yet fail to generalize under transformations. Our study extends beyond molecules to different domains, such as 3D shape data (ModelNet), showing that dataset alignment extends beyond the molecular domain. *However, we viewed the presence of canonicalization in benchmark datasets as the beginning, rather than the end, of the mystery.*
> > * *Key departures*: We then ran extensive experiments aiming to **connect distributional symmetry breaking/alignment to model behavior** – comparing equivariant models, different data augmentation settings, and group-averaged models across domains (Table 1). Rather than simply noting that the datasets contain distributional symmetry breaking, we evaluate what this means in each case for practitioners, and present preliminary hypotheses for how this is connected to equivariant model performance across domains (see Section 5.2: Task-Dependent Metric and Section 5.3: Locality). We also present nuanced theory showing that data augmentation can be harmful even in an invariant setting.
> > * We emphasize that the interplay between canonicalization and model performance is a complicated problem. We asked: why is this observation of interest? In particular, what does dataset alignment mean for practitioners training models on these datasets? To answer this question, we provided both theory and extensive experiments.
> >     * We chose to showcase interesting behaviors, such as when augmentation still helped for certain QM9 properties, “even when they might not explicitly align with the main hypothesis of this work” (-Reviewer oMVy)

---

### Author Response · Authors · 2025-12-03
**Message to New AC**

Dear AC,

We appreciate your effort to evaluate our paper under highly unusual circumstances. To start, here is a short summary of our work to orient you quickly:
- Most works in equivariant learning -- including theoretical works showing the benefit of equivariant methods -- assume that all group transformations of the datapoints are equally likely (for example, given a molecular dataset that is invariant under rotations, that all rotations of the input molecules are equally likely). Formally: given a group $G$, such as rotations or permutations, they assume that $p(x) = p(gx) \: \forall x, g$. When this assumption breaks, it is known as “distributional symmetry breaking” (DSB). DSB as a concept was defined in previous papers, but it had not been widely quantified. Our work first investigates whether this assumption holds in practice on benchmark point cloud datasets.
- **Key contribution 1**: We define a practical metric for rigorously quantifying distributional symmetry breaking. It is intuitive, and easy to implement: simply split your dataset in half, randomly transform one half, train a small classifier network to distinguish between the two halves, and compute its test accuracy. The higher it is, the more DSB is present.
- **Key contribution 2**: We then run this metric on many benchmark datasets, including QM9, MD17, OpenCatalyst, and ModelNet. Surprisingly, we found that they all have high degrees of DSB, i.e. the molecules/objects/digits take on extremely non-random orientations. In other words, the molecules in the datasets are, by and large, aligned! To our knowledge, this was not previously observed before our work.
This is a surprising observation in and of itself, but we then ask: what are the implications of this observation for learning?
- **Key contribution 3a**: For a theoretical ridge regression set-up in an asymptotic regime, we show that DSB can sometimes (but not always) render augmentation harmful, and provide some insight on what conditions make it so.
- **Key contribution 3b**: Complementing our theory, we run extensive experiments on the impact of equivariant augmentation and equivariant architectures under DSB. We find that, matching the theory, DSB can sometimes (but not always) render equivariance harmful on in-distribution data. As a practical heuristic, we suggest a more fine-grained “task-dependent metric” for measuring the relationship between the dataset bias implied by DSB, and the downstream task.
- Overall, we exhaustively investigate the phenomenon of distributional symmetry-breaking: when it occurs in practice, and what this means for learning. We expect it is of interest both for the equivariance community, and for the diverse communities that rely on these benchmarks.

## **Positive Reviewer Feedback**
Reviewers generally appreciated the following aspects of our paper:
- **The utility of the proposed metric**: it is “practical/easily implementable” (Reviewer oMVy), “elegant, interpretable, and easily applicable across data modalities” (Reviewer uapy). Reviewer oMVy also notes that “the paper’s topic is quite timely”
- **The theoretical analysis under ridge regression**: it provides “intuition and formal examples” (Reviewer oMVy), “deep insight into why augmentation might hurt” (Reviewer uapy)
- **Thorough experiments**: In general, our “extensive evaluation” covers “a large range of tasks and showcase interesting behaviors” (Review RG36) and provides a “comprehensive demonstration” (Review BM93).

---

> ### Author Response · Authors · 2025-12-03
> **Message to New AC**
>
> ## **Summary of Reviews/Rebuttal**
> The AC should of course make their own judgement, but here is our high-level summary of each review’s concerns + how we addressed them.
> ## **Reviewer oMVy**
> Overall: A thorough review, which was actually (in our opinion) generally quite positive. Their main concerns/questions preventing acceptance were:
> - **Sensitivity to hyperparameters**: We added wide ablation studies to Appendix D.5.1 (Figures 33/34), varying classifier model size and architecture, size of the dataset, and learning rate. The metric was remarkably stable to all of these settings.
> - **Task dependent metric**: We expanded on the task-dependent metric, meant to measure dependence between the input orientation and the label, in Section 5.2 [L451-480], moving discussion up from the appendix. For a few (but not all) QM9 properties, the task-dependent metric showed non-trivial leakage of information from the alignment to the task label. The metric is higher for tasks where equivariance does **not** help, improving practical interpretability. (see next point).
> - **Downstream utility of the metrics**: We added a flowchart for practitioners (showing when to use each metric and what to conclude) in Appendix C.4, Figure 10.
>
>     - TL;DR: If $m(p_X)$ is high and the task-dependent metric is also high (i.e. dataset orientations matter for the task), then use caution with equivariant methods: they may underperform in-distribution. (Of course, only the practitioner knows what distribution they want to generalize to. E.g., if they care about generalizing to all rotations, then equivariant methods are a good idea regardless of DSB.)
>      - We also argue that our observations about canonicalization in widely-used benchmark datasets (“Key contributions 1 and 2”) are important. As evidence of this, the concurrent submission discussed at the end of this note (https://openreview.net/forum?id=zrCGvLOrTL) received favorable initial reviews despite not addressing the implications for downstream learning.
> - **Why not symmetry breaking labels**: We clarified that this is a distinct phenomenon from “distributional symmetry breaking”, and not our focus in the paper. We focus on applications where labels are known to obey symmetry (e.g., molecules, materials, 3D physical systems), which is where equivariant methods have been most successful (for example, predicting molecular energies and forces [Equiformer v2 Liao et al 2023, Nequip Batzner et. al 2022]). We are quite confident that distributional symmetry breaking contains enough nuance and complexity to merit its own work.
>
> ## **Reviewer uapy**
> Overall: Also a detailed and generally positive review. Their concerns/questions aligned closely with oMVy’s 1-3 (see above), which we addressed.
> - **“Symmetry regularization”**: They also suggested using our classifier to selectively augment, i.e. to automatically interpolate between “no augmentation” and “full augmentation”. We think this is a great idea for future work, and would build naturally on existing work on learnable augmentation (e.g. “Learning Invariances in Neural Networks” by Benton et al 2020, or “Learning Probabilistic Symmetrization for Architecture Agnostic Equivariance” by Kim et al 2023). We implemented a proof-of-concept of this idea on MNIST in the revision, and it worked fairly well.
>
> ## **Reviewer RG36**
> Overall: Recommended for weak acceptance, albeit with a very short and possibly AI-generated review. Their main criticism was a claim about the lack of connection between theory and practically estimable quantities.
> - **Lack of connection between theory and practically estimable quantities**: We first clarified that the theory is supposed to be illustrative of a phenomenon (both its basic behaviors and its complexity), rather than directly practical -- the asymptotic regime and data distribution we study are tractable for analysis, but probably not fully representative of real learning setups (as is the case for most deep learning theory). We recommend our metrics as practical heuristics, as discussed above. To validate this recommendation, in the revision we computed our metric on the theoretical setting and showed that it correctly captured DSB (Figure 3).
> - **Application to non-group augmentations**: They asked if our work applies to Mixup and Cutout. We clarified that while these are image augmentations, they do not involve a symmetry group and are not in-scope for this work. In fact, our work is about asking if augmented versions of natural datapoints look “natural” themselves (formally, again, if $p(x)=p(gx)$). This is pretty clearly not the case for Mixup and Cutout -- e.g., natural images are not usually missing blocks of pixels.

---

> > ### Author Response · Authors · 2025-12-03
> > **Message to New AC**
> >
> > ## **Reviewer BM93**
> > Overall: The most critical review, with several concerns stemming from misunderstandings of the theory and the scope of the paper. We addressed their five concerns point by point.
> > - **Scope**: The reviewer questioned which of several goals (theory, empirical study, or practical metric) we aimed for. We disagree strongly that we lacked a focused goal, which was to understand the prevalence and impact of distributional symmetry breaking; we just used all the tools at our disposal to achieve it. We edited the paper to make this clearer, and clarified our contributions explicitly in the response.
> > - **Relevance of assumption**: They challenged whether the assumption that $p(x)=p(gx)$ is widespread beyond Elesedy et al. We listed four additional, seminal papers on the theory of equivariance that explicitly make this assumption. We also claim that it is often assumed implicitly in more empirical papers: they assume a problem with equivariant labels will automatically benefit from equivariant methods, but as we show theoretically and empirically, equivariant methods can hurt under DSB (even with equivariant labels).
> > - **Novelty of theory**: The lines the reviewer pointed to were background context provided to the reader (rather than our theoretical contribution). We clarified that the cited lines were intended as background context rather than novel (it did not have to do with DSB at all, in fact), and clarified which result was actually novel (Theorem 2). We renamed the statement in question (from “theorem” to “fact”) to clarify. See rebuttal for full details, including an interactive plot demonstrating the nuanced behavior of Theorem 2.
> > - **Sensitivity to hyperparameters**: Addressed with new ablations.
> > - **Alleged flaw in the Reynolds operator definition**: We explained why the written expression is correct. It is worth noting that this equation for group symmetrization, also known as the Reynolds operator, is a very standard and widespread equation in the equivariance literature, as well as invariant theory. It is also fundamental for understanding the entire approach. To us, this error + their lack of awareness of the equivariance papers we cited casts a bit of doubt on their confidence score of 4.
> > - They asked us to clarify our use of interchangeable symmetry-related terms. This was a fair criticism, and we added clear definitions in the revision.
> > - It’s worth noting that the reviewer seems to have focused almost entirely on the theory, and did not comment on the extensive empirical findings.

---

> > > ### Author Response · Authors · 2025-12-03
> > > **Message to New AC**
> > >
> > > ## **Emphasizing Additional Points from Rebuttal**
> > > There were a few additional relevant points from the rebuttal we’d like to summarize here:
> > >
> > > **Concurrent submission**:
> > > - It came to our attention that there is a well-received submission to ICLR (https://openreview.net/pdf?id=zrCGvLOrTL) entitled “Take Note: Your Molecular Dataset is Probably Aligned”. This work very similarly proposes a binary classifier for assessing the distributional symmetry-breaking (or “alignment”) present in a dataset and applies it to molecular datasets.
> > > - Both the work itself and its enthusiastic reviews demonstrate the timeliness and general community interest of this line of investigation.
> > > - While their work is a valuable addition to the community, we believe our work takes the question of molecular dataset alignment significantly further. We carefully probe what alignment means for downstream performance of equivariant methods (both augmentation and equivariant architectures), and provide the first theoretical analysis of the interplay between DSB and data augmentation. We reveal a surprisingly complex picture, in which equivariant methods sometimes hurt in-distribution (ModelNet), yet sometimes still help (QM9) in-distribution, in the presence of rotational alignment.
> > >
> > > **Locality experiment** (in the original submission, but not noted by reviewers):
> > > - (Section 5.3, L480). We show evidence that the success of equivariant models on canonicalized molecular datasets is due to locally symmetric features. We use $m(p_X)$ to determine that local atomic neighborhoods in QM9 are less canonicalized than entire molecules.
> > >
> > > **LLM dataset** (in the original submission, but possibly overlooked by reviewers):
> > > - (Section 5.1, L419). There is growing interest in training LLMs on materials science data, and prior work (Gruver et al. 2024) shows that permutation augmentation on an LLM materials dataset hurt model performance (even though the task was permutation invariant). Our metric yielded 95% accuracy, demonstrating that distributional symmetry breaking is quite prevalent in molecule-to-text representations and providing an explanation for this phenomenon.
> > >
> > > Disappointingly, the OpenReview leak occurred before we heard back from any of the reviewers (four days after we posted them, which overlapped with Thanksgiving in the USA). We are very confident that we addressed the reviewers’ comments, and of course encourage the AC to read the in-depth reviews and our responses to verify our summary for themselves.

---

### Meta-Review · Area_Chair_uDtb · 2026-01-06

**Summary:**

The submission investigates the phenomenon of distributional symmetry breaking that arises when data augmentations are used to approximately enforce equivariances/invariances. The core idea is that two elements of the same orbit may have different probability masses, leading to suboptimal models when augmentation approaches are used to synthesise unobserved data points. Many of the weaknesses raised by reviewers can be categorised as misunderstandings or presentation issues. However, a fairly crucial aspect is picked up on by three of the four reviewers: the proposed metric may be quite sensitive to the choice of hypothesis class and learning algorithm hyperparameters. A second weakness that I believe is less important but still warranted to bring up is that the proposed metric does not have a concrete connection to the downstream performance of models.

**Reviewer Concerns:**

I believe the primary concern regarding the sensitivity of the proposed metric to the choice of classifier hypothesis class and learning algorithm hyperparameters has been adequately addressed during the discussion phase. The authors have provided a substantially expanded set of experiments verifying that the metric is not overly sensitive to the choice of model class and hyperparameters. The second weakness, regarding the lack of quantitative connection between the proposed metric and downstream performance, is only partially addressed. However, I believe the impact of the proposed metric is much broader than implied by this criticism, so I do not view this as a justification to reject the paper. The authors have gone to great lengths to clear up many of the small misunderstandings and presentation issues picked up on by reviewers, and I think this has been successful for the most part. I therefore recommend the paper is accepted for a poster presentation at ICLR.

**Reviewer Scores:**

I think reviewers oMYv and uapy would have been very likely to increase their score to at least 6. Reviewer BM93 probably would have increased their score to 4, with a small chance they would even go up to 6. It is unclear to me whether reviewer RG36 would have increased their score, but I do not think they would have decreased it from 6.

---

### Decision · Program_Chairs · 2026-01-26

Accept (Poster)